# Efficient coding in biophysically realistic excitatory-inhibitory spiking networks

**Veronika Koren[1,2,3]\*, Simone Blanco Malerba[1], Tilo Schwalger[2,3], Stefano Panzeri[1]\***

[1]Institute of Neural Information Processing, Center for Molecular Neurobiology (ZMNH), University Medical Center Hamburg-Eppendorf, Hamburg, Germany; [2]Institute of Mathematics, Technische Universität Berlin, Berlin, Germany; [3]Bernstein Center for Computational Neuroscience Berlin, Berlin, Germany

## eLife Assessment

This study offers a **valuable** treatment of how the population of excitatory and inhibitory neurons integrates principles of energy efficiency in their coding strategies. The **convincing** analysis provides a comprehensive characterisation of the model, highlighting the structured connectivity between excitatory and inhibitory neurons. The role of the many free parameters are discussed and studied in depth.

**\*For correspondence:**
v.koren@uke.de (VK);
s.panzeri@uke.de (SP)

**Competing interest:** The authors declare that no competing interests exist.

**Abstract** The principle of efficient coding posits that sensory cortical networks are designed to encode maximal sensory information with minimal metabolic cost. Despite the major influence of efficient coding in neuroscience, it has remained unclear whether fundamental empirical properties of neural network activity can be explained solely based on this normative principle. Here, we derive the structural, coding, and biophysical properties of excitatory-inhibitory recurrent networks of spiking neurons that emerge directly from imposing that the network minimizes an instantaneous loss function and a time-averaged performance measure enacting efficient coding. We assumed that the network encodes a number of independent stimulus features varying with a time scale equal to the membrane time constant of excitatory and inhibitory neurons. The optimal network has biologically plausible biophysical features, including realistic integrate-and-fire spiking dynamics, spike-triggered adaptation, and a non-specific excitatory external input. The excitatory-inhibitory recurrent connectivity between neurons with similar stimulus tuning implements feature-specific competition, similar to that recently found in visual cortex. Networks with unstructured connectivity cannot reach comparable levels of coding efficiency. The optimal ratio of excitatory vs inhibitory neurons and the ratio of mean inhibitory-to-inhibitory vs excitatory-to-inhibitory connectivity are comparable to those of cortical sensory networks. The efficient network solution exhibits an instantaneous balance between excitation and inhibition. The network can perform efficient coding even when external stimuli vary over multiple time scales. Together, these results suggest that key properties of biological neural networks may be accounted for by efficient coding.

## Introduction

Information about the sensory world is represented in the brain through the dynamics of neural population activity (*Abbott et al., 2016*; *Thalmeier et al., 2016*). One prominent theory about the principles that may guide the design of neural computations for sensory function is efficient coding (*Barlow, 1961*; *Olshausen and Field, 1996*; *Deneve and Chalk, 2016a*). This theory posits that neural computations are optimized to maximize the information that neural systems encode about sensory stimuli while at the same time limiting the metabolic cost of neural activity. Efficient coding has been highly

**eLife digest** The networks of nerve cells that make up the brain are complex and versatile. They enable the information processing necessary for both simple and complex thought processes. But, the organization of nerve networks in the brain is a topic of great debate among scientists.

One idea is that nerve cell networks in the brain are organized to be as efficient as possible at transmitting information. Scientists supporting this idea say it allows the brain to send accurate information using as little energy as possible.

Scientists have developed mathematical models to explain how this efficient coding model of brain activity works. But how accurately these mathematical models capture complex brain tasks is up for debate. Some question how well these models explain how the brain makes sense of sensory information like sights or smells. Are they able to explain nerve cell organization or how nerve cells react to new information or experiences? Scientists also question how well the mathematical models capture biological and physical constraints on nerve cell activity.

To answer these questions, Koren et al. used mathematical models to systematically test whether the efficient coding model was consistent with what happens in realistic circumstances. The experiments show that mathematical models of efficient coding are consistent with actual brain cell behavior, organization and interconnections. The models also reflected the cells' biological and physical constraints.

The experiments support the idea that brain networks are designed for efficiency. But the models used in the study are too simple to assess the full range and complexity of information processing in the brain. More studies are needed to test more complex mathematical models that better recreate more advanced brain activities. Further study of the biological and physical constraints on nerve cells in the brain may shed more light on how they behave in brain networks.

influential as a normative theory of how networks are organized and designed to optimally process natural sensory stimuli in visual (*Atick, 1992*; *Olshausen and Field, 1997*; *Simoncelli and Olshausen, 2001*; *Vinje and Gallant, 2000*; *Olshausen and Field, 2004*; *Li, 2014*), auditory (*Lewicki, 2002*), and olfactory sensory pathways (*Koulakov and Rinberg, 2011*).

The first normative neural network models (*Olshausen and Field, 1996*; *Olshausen and Field, 2004*) designed with efficient coding principles had at least two major levels of abstractions. First, neural dynamics was greatly simplified, ignoring the spiking nature of neural activity. Instead, biological networks often encode information through millisecond-precise spike timing (*Bialek et al., 1991*; *Bialek and Rieke, 1992*; *Panzeri et al., 2001*; *Nemenman et al., 2008*; *Kayser et al., 2010*; *Ince et al., 2013*; *Panzeri et al., 2010*). Second, these earlier contributions mostly considered encoding of static sensory stimuli, whereas the sensory environment changes continuously at multiple timescales and the dynamics of neural networks encodes these temporal variations of the environment (*Fairhall et al., 2001*; *Wark et al., 2009*; *Mazzoni et al., 2008*; *Młynarski and Hermundstad, 2021*).

Recent years have witnessed a considerable effort and success in laying down the mathematical tools and methodology to understand how to formulate efficient coding theories of neural networks with more biological realism (*Koren et al., 2023*). This effort has established the incorporation of recurrent connectivity (*Lochmann et al., 2012*; *Zhu and Rozell, 2013*), of spiking neurons, and of time-varying stimulus inputs (*Boerlin et al., 2013*; *Bourdoukan et al., 2012*; *Moreno-Bote and Drugowitsch, 2015*; *Chalk et al., 2016*; *Deneve and Machens, 2016b*; *Gutierrez and Denève, 2019*; *Kadmon et al., 2020*; *Rullán Buxó and Pillow, 2020*). In these models, the efficient coding principle has been implemented by designing networks whose activity maximizes the encoding accuracy, by minimizing the error between a desired representation and a linear readout of network's activity, subject to a constraint on the metabolic cost of processing. This double objective is captured by a loss function that trades off encoding accuracy and metabolic cost. The minimization of the loss function is performed through a greedy approach, by assuming that a neuron will emit a spike only if this will decrease the loss. This, in turn, yields a set of leaky integrate-and-fire (LIF) neural equations (*Boerlin et al., 2013*; *Bourdoukan et al., 2012*), which can also include biologically plausible non-instantaneous synaptic delays (*Koren and Denève, 2017*; *Rullán Buxó and Pillow, 2020*; *Kadmon et al., 2020*). Although most initial implementations did not respect Dale's law, further studies analytically derived efficient networks of excitatory (E) and inhibitory (I) spiking neurons that respect Dale's

law (*Boerlin et al., 2013*; *Barrett et al., 2016*; *Chalk et al., 2016*; *Koren and Panzeri, 2022*) and included spike-triggered adaptation (*Koren and Panzeri, 2022*). These networks take the form of generalized leaky integrate-and-fire (gLIF) models neurons, which are realistic models of neuronal activity (*Brette and Gerstner, 2005*; *Mensi et al., 2012*; *Gerstner et al., 2014*) and capable of accurately predicting real neural spike times in vivo (*Jolivet et al., 2008*). Efficient spiking models thus have the potential to provide a normative theory of neural coding through spiking dynamics of E-I circuits (*Brendel et al., 2020*; *Koren and Panzeri, 2022*; *Podlaski and Machens, 2024*) with high biological plausibility.

However, despite the major progress described above, we still lack a thorough characterization of which structural, coding, biophysical and dynamical properties of excitatory- inhibitory recurrent spiking neural networks directly relate to efficient coding. Previous studies only rarely made predictions that could be quantitatively compared against experimentally measurable biological properties. As a consequence, we still do not know which, if any, fundamental properties of cortical networks emerge directly from efficient coding.

To address the above questions, we systematically analyze our biologically plausible efficient coding model of E and I neurons that respects Dale's law (*Koren and Panzeri, 2022*). We make concrete predictions about experimentally measurable structural, coding and dynamical features of neurons that arise from efficient coding. We systematically investigate how experimentally measurable emergent dynamical properties, including firing rates, trial-to-trial spiking variability of single neurons and E-I balance (*Vogels et al., 2011*), relate to network optimality. We further analyze how the organization of the connectivity arising by imposing efficient coding relates to the anatomical and effective connectivity recently reported in visual cortex, which suggests competition between excitatory neurons with similar stimulus tuning. We find that several key and robustly found empirical properties of cortical circuits match those of our efficient coding network. This lends support to the notion that efficient coding may be a design principle that has shaped the evolution of cortical circuits and that may be used to conceptually understand and interpret them.

## Results

### Assumptions and emergent structural properties of the efficient E-I network derived from first principles

We study the properties of a spiking neural network in which the dynamics and structure of the network are analytically derived starting from first principles of efficient coding of sensory stimuli. The model relies on a number of assumptions, described next.

The network responds to $M$ time-varying features of a sensory stimulus, $\vec{s}(t) = [s_1(t), \ldots, s_M(t)]^\top$ (e.g. for a visual stimulus, contrast, orientation, etc.) received as inputs from an earlier sensory area. We model each feature $s_k(t)$ as an independent Ornstein–Uhlenbeck (OU) processes (see Materials and methods). The network's objective is to compute a leaky integration of sensory features; the target representations of the network, $\vec{x}(t)$, is defined as

$$\dot{\vec{x}}(t) = -\frac{1}{\tau}\vec{x}(t) + \vec{s}(t), \tag{1}$$

with $\tau$ a characteristic integration time-scale (*Figure 1A(i)*). We assumed leaky integration of sensory features for consistency with previous theoretical models (*Barrett et al., 2016*; *Chalk et al., 2016*; *Gutierrez and Denève, 2019*). This assumption stems from the finding that, in many cases, integration of sensory evidence by neurons is well described by an exponential kernel (*Scott et al., 2017*; *Danskin et al., 2023*). Additionally, a leaky integration of neural activity with an exponential kernel implemented in models of neural activity readout often explains well the perceptual discrimination results (*Gold and Shadlen, 2001*; *Usher and McClelland, 2001*; *Chong et al., 2020*). This suggests that the assumption of leaky integration of sensory evidence, although possibly simplistic, captures relevant aspects of neural computations.

The network is composed of two neural populations of excitatory (E) and inhibitory (I) neurons, defined by their postsynaptic action which respects Dale's law. For each population, $y \in \{E, I\}$, we define a population readout of each feature, $\hat{\vec{x}}^y(t)$, as a filtered weighted sum of spiking activity of neurons in the population,

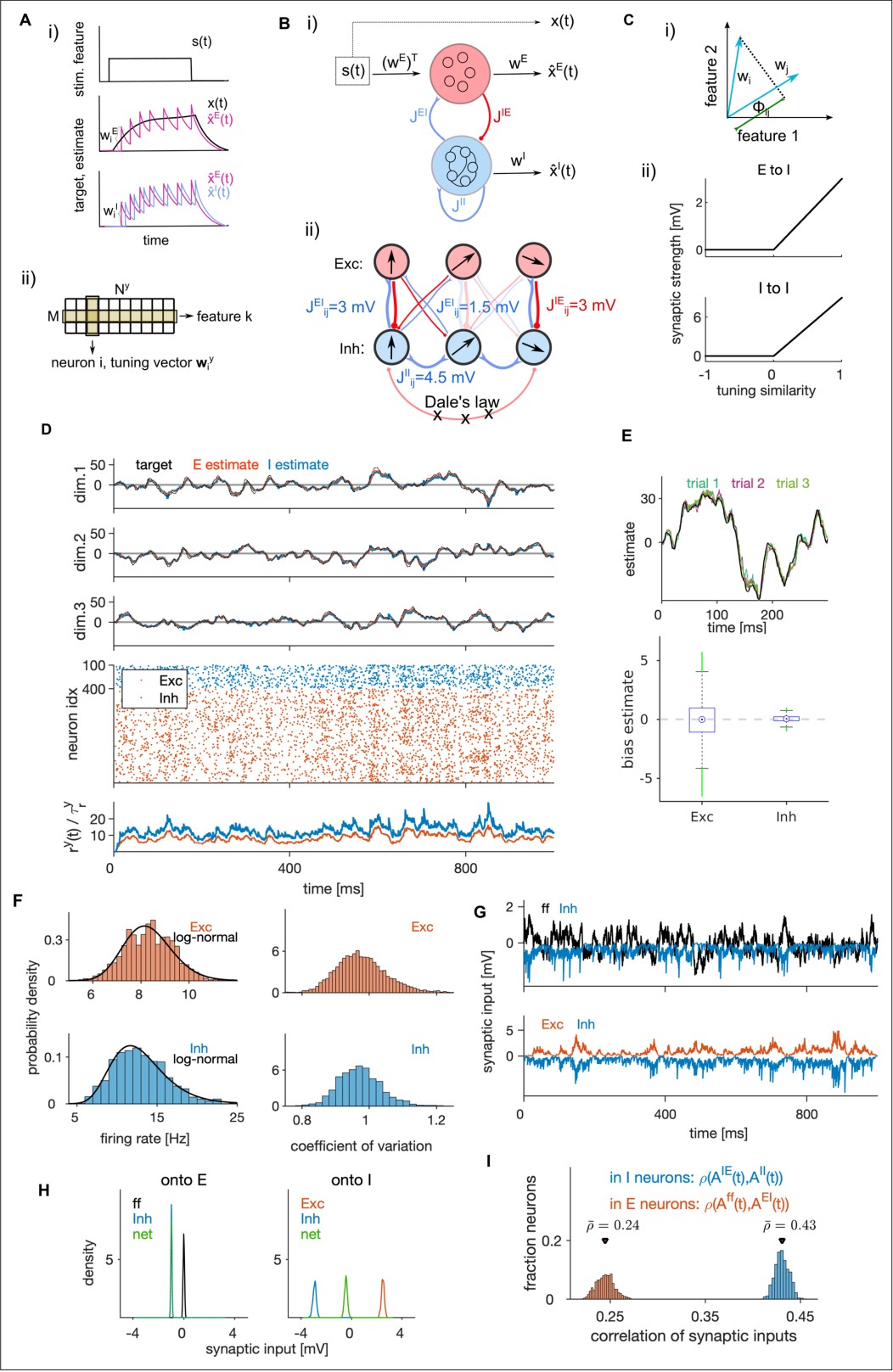

**Figure 1.** Structural and dynamical properties of the efficient E-I spiking network. (**A, i**) Encoding of a target signal representing the evolution of a stimulus feature (top) with one E (middle) and one I spiking neuron (bottom). The target signal $x(t)$ integrates the input signal $s(t)$. The readout of the E neuron tracks the target signal and the readout of the I neuron tracks the readout of the E neuron. Neurons spike to bring the readout of their activity

*Figure 1 continued on next page*

*Figure 1 continued*

closer to their respective target. Each spike causes a jump of the readout, with the sign and the amplitude of the jump being determined by neuron's tuning parameters. (ii) Schematic of the matrix of tuning parameters. Every neuron is selective to all stimulus features (columns of the matrix), and all neurons participate in encoding of every feature (rows). (**B, i**) Schematic of the network with E (red) and I (blue) cell type. E neurons are driven by the stimulus features while I neurons are driven by the activity of E neurons. E and I neurons are connected through recurrent connectivity matrices. (ii) Schematic of E (red) and I (blue) synaptic interactions. Arrows represent the direction of the tuning vector of each neuron. Only neurons with similar tuning are connected and the connection strength is proportional to the tuning similarity. (**C, i**) Schematic of similarity of tuning vectors (tuning similarity) in a 2-dimensional space of stimulus features. (ii) Synaptic strength as a function of tuning similarity. (**D**) Coding and dynamics in a simulation trial. Top three rows show the signal (black), the E estimate (red) and the I estimate (blue) for each of the three stimulus features. Below are the spike trains. In the bottom row, we show the average instantaneous firing rate (in Hz). (**E**) Top: Example of the target signal (black) and the E estimate in three simulation trials (colors) for one stimulus feature. Bottom: Distribution (across time) of the time-dependent bias of estimates in E and I cell type. (**F**) Left: Distribution of time-averaged firing rates in E (top) and I neurons (bottom). Black traces are fits with log-normal distribution. Right: Distribution of coefficients of variation of interspike intervals for E and I neurons. (**G**) Distribution (across neurons) of time-averaged synaptic inputs to E (left) and I neurons (right). In E neurons, the mean of distributions of inhibitory and of net synaptic inputs are very close. (**H**) Sum of synaptic inputs over time in a single E (top) and I neuron (bottom) in a simulation trial. (**I**) Distribution (across neurons) of Pearson's correlation coefficients measuring the correlation of synaptic inputs $A^{Ey}$ and $A^{Iy}$ (as defined in Materials and methods, *Equation 43*) in single E (red) and I (blue) neurons. All statistical results (**E–F, H–I**) were computed on 10 simulation trials of 10 s duration. For model parameters, see *Table 1*.

The online version of this article includes the following figure supplement(s) for figure 1:

**Figure supplement 1.** Efficient spiking model with one cell type and the encoding bias of the E-I network.

$$\dot{\vec{x}}^y(t) = -\frac{1}{\tau}\vec{x}^y(t) + \sum_{i=1}^{N^y} \vec{w}_i^y f_i^y(t), \tag{2}$$

where $f_i^y(t)$ is the spike train of neuron $i$ of type $y$ and $\vec{w}_i^y = [w_{1i}^y, \ldots, w_{Mi}^y]^\top$ is the vector of decoding weights of the neuron for features $k = 1, \ldots, M$ (*Figure 1A(ii)*). We assume that every neuron encodes multiple ($M > 1$) stimulus features and that the encoding of every stimulus is distributed among neurons. As a result of the optimization, the decoding weights of the neurons are equivalent to the neuron's stimulus tuning parameters (see Materials and methods and *Brendel et al., 2020*). We sampled tuning parameters uniformly from a $M$-dimensional hypersphere with unit radius, giving tuning vectors with unit length to all neurons (see Materials and methods). To control the amount of inhibition in the network, we then multiplied the tuning vectors of I neurons with a factor $d > 1$, homogeneously across all I neurons. Normalization of decoding vectors preserves the heterogeneity of decoding weights across neurons, which may benefit coding efficiency (*Zeldenrust et al., 2021*).

Following previous work (*Boerlin et al., 2013*; *Barrett et al., 2016*; *Chalk et al., 2016*), we impose that E and I neurons have distinct normative objectives and we define specific loss functions relative to each neuron type. To implement at the same time, as requested by efficient coding, the constraints of faithful stimulus representation with limited computational resources (*Tavoni et al., 2019*), we define the loss functions of the population $y \in \{E, I\}$ as a weighted sum of a time-dependent encoding error and time-dependent metabolic cost:

$$L^y(t) = \epsilon^y(t) + \beta \kappa^y(t), \qquad y \in \{E, I\}. \tag{3}$$

We refer to $\beta$, the parameter controlling the relative importance of the metabolic cost over the encoding error, as the metabolic constant of the network. We hypothesize that population readouts of E neurons, $\vec{x}^E(t)$, track the target representations, $\vec{x}(t)$, and the population readouts of I neurons, $\vec{x}^I(t)$, track the population readouts of E neurons, $\vec{x}^E(t)$, by minimizing the squared error between these quantities (*Koren and Panzeri, 2022*) (see also (*Boerlin et al., 2013*; *Denève et al., 2017*) for related approaches). Furthermore, we hypothesize the metabolic cost to be proportional to the instantaneous estimate of network's firing frequency. We thus define the variables of loss functions in *Equation 3* as

$$\varepsilon^E(t) = \left\| \vec{x}(t) - \vec{x}^E(t) \right\|^2, \qquad \kappa^E(t) = \sum_{i=1}^{N^E} \left[ r_i^E(t) \right]^2,$$

$$\varepsilon^I(t) = \left\| \vec{x}^E(t) - \vec{x}^I(t) \right\|^2, \qquad \kappa^I(t) = \sum_{i=1}^{N^I} \left[ r_i^I(t) \right]^2,$$

(4)

where $r_i^y$, $y \in \{E, I\}$, is the low-pass filtered spike train of neuron $i$ (single neuron readout) with time constant $\tau_r^y$, proportional to the instantaneous firing rate of the neuron: $z_i^y(t) = (\tau_r^y)^{-1} r_i^y(t)$. We then impose the following condition for spiking: a neuron emits a spike at time $t$ only if this decreases the loss function of its population (*Equation 3*) in the immediate future. The condition for spiking also includes a noise term (Materials and methods) accounting for sources of stochasticity in spike generation (*Faisal et al., 2008*) which include the effect of non-specific inputs from the rest of the brain.

We derived the dynamics and network structure of a spiking network that instantiates efficient coding (*Figure 1B*, see Materials and methods). The derived dynamics of the subthreshold membrane potential $V_i^E(t)$ and $V_i^I(t)$ obey the equations of the generalized leaky integrate and fire (gLIF) neuron

$$\tau \dot{V}_i^y(t) = -\left( V_i^y(t) - V_{\text{rest}}^y \right) + R_m \left( I_i^{\text{syn},y}(t) - I_i^{\text{ad},y}(t) + I_i^{\text{ext},y}(t) \right), \qquad y \in \{E, I\},$$

(5)

where $I_i^{\text{syn},y}$, $I_i^{\text{ad},y}$, and $I_i^{\text{ext},y}$ are synaptic current, spike-triggered adaptation current and non-specific external current, respectively, $R_m$ is the membrane resistance and $V_{\text{rest}}^y$ is the resting potential. This dynamics is complemented with a fire-and-reset rule: when the membrane potential reaches the firing threshold $\vartheta^y$, a spike is fired and $V_i^y(t)$ is set to the reset potential $V_i^{\text{reset},y}$. The analytical solution in *Equation 5* holds for any number of neurons (with at least 1 neuron in each population) and predicts an optimal spike pattern to encode the presented external stimulus. Following previous work (*Boerlin et al., 2013*) in which physical units were assigned to derived mathematical expressions to interpret them as biophysical variables, we express computational variables (target stimuli in *Equation 1*, population readouts in *Equation 2* and the metabolic constant in *Equation 3*) with physical units in such a way that all terms of the biophysical model (*Equation 5*) have realistic physical units.

The synaptic currents in E neurons, $I_i^{\text{syn},E}$, consist of feedforward currents, obtained as stimulus features $\vec{s}(t)$ weighted by the tuning weights of the neuron, and of recurrent inhibitory currents (*Figure 1B*). Synaptic currents in I neurons, $I_i^{\text{syn},I}$, consist of recurrent excitatory and inhibitory currents. Note that there are no recurrent connections between E neurons, a consequence of our assumption of no across-feature interaction in the leaky integration of stimulus features (*Equation 1*). This assumption is likely to be simplistic even for early sensory cortices (*Emanuel et al., 2021*). However, in other studies, we found that many properties of efficient networks implementing leaky integration hold also when input features are linearly mixed during integration (*Koren and Panzeri, 2022*; *Koren et al., 2023*).

The optimization of the loss function yielded structured recurrent connectivity (*Figure 1B(ii)– C*). Synaptic strength between two neurons is proportional to their tuning similarity, forming like-to-like connectivity, if the tuning similarity is positive; otherwise the synaptic weight is set to zero (*Figure 1C(ii)*) to ensure that Dale's law is respected. A connectivity structure in which the synaptic weight is proportional to pairwise tuning similarity is consistent with some empirical observations in visual cortex (*Znamenskiy et al., 2024*) and has been suggested by previous models (*Boerlin et al., 2013*; *Sadeh and Clopath, 2020*). Such connectivity organization is also suggested by across-neuron influence measured with optogenetic perturbations of visual cortex (*Chettih and Harvey, 2019*; *Oldenburg et al., 2024*). While such connectivity structure is the result of optimization, the rectification of the connectivity that enforces Dale's law does not emerge from imposing efficient coding, but from constraining the space of solutions to biologically plausible networks. Rectification also sets the overall connection probability to 0.5, which is consistent with empirically observed connection probability from pyramidal (E) neurons to parvalbumin-positive (I) neurons (*Pala and Petersen, 2015*; *Campagnola et al., 2022*), but likely overestimates the connection probability from parvalbumin-positive neurons to pyramidal neurons, which tends to be lower (*Campagnola et al., 2022*). (For a study of how efficient coding would be implemented if the above Dale's law constraint were removed and each neuron were free to have either an inhibitory or excitatory effect depending on the postsynaptic target, see Appendix 1 and *Figure 1—figure supplement 1A–E*).

**Table 1.** Table of default model parameters for the efficient E-I network.
Parameters above the double horizontal line are the minimal set of parameters needed to simulate model equations (*Equation 29a-29h* in Materials and methods). Parameters below the double horizontal line are biophysical parameters, derived from the same model equations and from model parameters listed above the horizontal line. Parameters $N^E$, $M$, $\tau$ and $\sigma_w^E$ were chosen for their biological plausibility and computational simplicity. Parameters $N^I$, $\tau_r^E$, $\tau_r^I$, $\sigma$, ratio of mean E-I to I-I synaptic connectivity and $\beta$ are parameters that maximize network efficiency (see the section 'Criterion for determining model parameters' in Materials and methods). The metabolic constant $\beta$ and the noise strength $\sigma$ are interpreted as global network parameters and are for this reason assumed to be the same across the E and I population, e.g., $\beta^E = \beta^I = \beta$ and $\sigma^E = \sigma^I = \sigma$ (see *Equation 3*). The connection probability of $p^{xy} = 0.5$ is the consequence of rectification of the connectivity (see *Equation 24* in Materials and methods).

| Parameter | Notation | Value |
|---|---|---|
| Number of E neurons | $N^E$ | 400 |
| Ratio of E to I neuron numbers | $N^E : N^I$ | 4:1 |
| Number of the input features | $M$ | 3 |
| Time constant of the population readout (E and I) | $\tau$ | 10ms |
| Time constant of the single neuron readout | $\tau_r^E = \tau_r^I$ | 10ms |
| Noise strength (non-specific current) | $\sigma$ | 5.0 mV |
| Heterogeneity factor of tuning parameters in E | $\sigma_w^E$ | 1.0 (mV)$^{1/2}$ |
| Ratio of mean I-I to E-I synaptic connectivity | mean I-I: mean E-I | 3:1 |
| Metabolic constant | $\beta$ | 14 mV |
| Threshold constant | $c/2$ | 18 mV |
| Distance threshold to reset potential (E neurons) | $\lvert\vartheta^E - V_{\text{rest}}^E\rvert$ | 19 mV |
| Distance threshold to reset potential (I neurons) | $\lvert\vartheta^I - V_{\text{rest}}^I\rvert$ | 21 mV |
| Connection probability (recurrent synapses) | $p^{IE} = p^{II} = p^{EI}$ | 0.5 |
| Mean E-I synaptic weight (EPSP to I at max) | $\langle J_{ij}^{IE} \rangle$ | 0.75 mV |
| Mean I-E synaptic weight (IPSP to E at max) | $\langle J_{ij}^{EI} \rangle$ | 0.75 mV |
| Mean I-I synaptic weight (IPSP at max) | $\langle J_{ij}^{II} \rangle$ | 2.25 mV |

The spike-triggered adaptation current of neuron $i$ in population $y$, $I_i^{\text{ad},y}$, is proportional to its low-pass filtered spike train. This current realizes spike-frequency adaptation or facilitation depending on the difference between the time constants of population and single neuron readout (see Results subsection 'Weak or no spike-triggered adaptation optimizes network efficiency').

Finally, non-specific external currents $I_i^{\text{ext},y}(t)$ have a constant mean that depends on the parameter $\beta$, and fluctuations that arise from the noise with strength $\sigma$ in the condition for spiking. The relative weight of the metabolic cost over the encoding error, $\beta$, controls how the network responds to feed-forward stimuli, by modulating the mean of the non-specific synaptic currents incoming to all neurons. Together with the noise strength $\sigma$, these two parameters set the non-specific synaptic currents to single neurons that are homogeneous across the network and akin to the background synaptic input discussed in *Destexhe et al., 2003*. By allowing a large part of the distance between the resting potential and the threshold to be taken by the non-specific current, we found a biologically plausible set of optimally efficient model parameters (*Table 1*) including the firing threshold at about 20 mV from the resting potential, which is within the experimental ballpark (*Constantinople and Bruno,*

*2013*), and average synaptic strengths of 0.75 mV (E-I and I-E synapses) and 2.25 mV (I-I synapses), which are consistent with measurements in sensory cortex (*Campagnola et al., 2022*). An optimal network without non-specific currents can be derived (see Materials and methods, *Equation 25*), but its parameters are not consistent with biology (see Appendix 2). The non-specific currents can be interpreted as synaptic currents that are modulated by larger-scale variables, such as brain states (see subsection 'Non-specific currents regulate network coding properties').

To summarize, the analytical derivation of an optimally efficient network includes gLIF neurons (*Burkitt, 2006*; *Jolivet et al., 2008*; *Gerstner et al., 2014*; *Schwalger et al., 2017*; *Harkin et al., 2021*), a distributed code with linear mixed selectivity to the input stimuli (*Chang and Tsao, 2017*; *Kaufman et al., 2014*), spike-triggered adaptation, structured synaptic connectivity, and a non-specific external current akin to background synaptic input.

## Encoding performance and neural dynamics in an optimally efficient E-I network

The equations for the E-I network of gLIF neurons in *Equation 5* optimize the loss functions at any given time and for any set of parameters. In particular, the network equations have the same analytical form for any positive value of the metabolic constant $\beta$. To find a set of parameters that optimizes the overall performance, we minimized the loss function averaged over time and trials. We then optimized the parameters by setting the metabolic constant $\beta$ such that the encoding error weights 70% and the metabolic error weights 30% of the average loss, and by choosing all other parameters such as to minimize numerically the average loss (see Materials and methods). The numerical optimization was performed by simulating a model of 400 E and 100 I units, a network size relevant for computations within one layer of a cortical microcolumn (*Lefort et al., 2009*). The set of model parameters that optimized network efficiency is detailed in *Table 1*. Unless otherwise stated, we will use the optimal parameters of *Table 1* in all simulations and only vary parameters detailed in the figure axes.

With optimally efficient parameters, population readouts closely tracked the target signals (*Figure 1D*, M=3, $R^2 = [0.95, 0.97]$ for E and I neurons, respectively). When stimulated by our three-dimensional time-varying feedforward input, the optimal E-I network provided a precise estimator of target signals (*Figure 1E*, top). The average estimation bias ($B^E$ and $B^I$, see Materials and methods) of the network minimizing the encoding error was close to zero ($B^E = 0.02$ and $B^I = 0.03$) while the bias of the network minimizing the average loss (and optimizing efficiency) was slightly larger and negative ($B^E = -0.15$ and $B^I = -0.34$), but still small compared to the stimulus amplitude (*Figure 1E*, bottom, *Figure 1—figure supplement 1F*). Time- and trial-averaged encoding error (RMSE) and metabolic cost (MC, see Materials and methods) were comparable in magnitude ($RMSE = [3.5, 2.4]$, $MC = [4.4, 2.8]$ for E and I), but with smaller error and lower cost in I, leading to a better performance in I (average loss of 2.5) compared to E neurons (average loss of 3.7). We report both the encoding error and the metabolic cost throughout the paper, so that readers can evaluate how these performance measures may generalize when weighting differently the error and the metabolic cost.

Next, we examined the emergent dynamical properties of an optimally efficient E-I network. I neurons had higher average firing rates compared to E neurons, consistently with observations in cortex (*Neske et al., 2015*). The distribution of firing rates was well described by a log-normal distribution (*Figure 1F*, left), consistent with distributions of cortical firing observed empirically (*Buzsáki and Mizuseki, 2014*). Neurons fired irregularly, with mean coefficient of variation (CV) slightly smaller than 1 (*Figure 1F*, right; CV = [0.97, 0.95] for E and I neurons, respectively), compatible with cortical firing (*Softky and Koch, 1993*). We assessed E-I balance in single neurons through two complementary measures. First, we calculated the *average* (global) balance of E-I currents by taking the time-average of the net sum of synaptic inputs (shortened to net synaptic input, see *Ahmadian and Miller, 2021*). Second, we computed the *instantaneous* (*Okun and Lampl, 2008*; also termed detailed in *Vogels et al., 2011*) E-I balance as the Pearson correlation ($\rho$) over time of E and I currents received by each neuron (see Materials and methods).

We observed an excess inhibition in both E and I neurons, with a negative net synaptic input in both E and I cells (*Figure 1H*), indicating an inhibition-dominated network according to the criterion of average balance (*Ahmadian and Miller, 2021*). In E neurons, net synaptic current is the sum of the feedforward current and recurrent inhibition and the mean of the net current is close to the mean of the inhibitory current, because feedforward inputs have vanishing mean. Furthermore, we found a

**Table 2.** Table of parameter ranges for Monte-Carlo sampling.
Minimum and maximum of the uniform distributions from which we randomly drew parameters during Monte-Carlo random sampling.

| Parameter | $\tau_r^E$ | $\tau_r^I$ | $\beta$ | $\sigma$ | $N^E : N^I$ | mean I-I: mean E-I |
|---|---|---|---|---|---|---|
| minimum | 5 ms | 5 ms | 2 mV | 1 mV | 1 | 1 |
| maximum | 50 ms | 50 ms | 29 mV | 10 mV | 8 | 8 |

moderate instantaneous balance (*Xue et al., 2014*), stronger in I compared to E cell type (*Figure 1G, I*, $\rho = [0.44, 0.25]$, for I and E neurons, respectively), similar to levels measured empirically in rat visual cortex (*Tan et al., 2013*).

We determined optimal model parameters by optimizing one parameter at a time. To independently validate the so obtained optimal parameter set (reported in *Table 1*), we varied all six model parameters explored in the paper with Monte-Carlo random joint sampling (10,000 random samples), uniformly within a biologically plausible parameter range for each parameter (*Table 2*). We did not find any parameter configuration with lower average loss than the setting in *Table 1* (*Figure 2A–B*) when using the weighting of the encoding error with metabolic cost between $0.4 < g_L < 0.81$ (*Figure 2C*). The three parameter settings that came the closest to our configuration on *Table 1* had stronger noise but also stronger metabolic constant than our configuration (*Table 3*). The second, third and fourth configurations had longer time constants of both E and I single neurons. Ratios of E-I neuron numbers and of I-I to E-I connectivity in the second, third, and fourth best configuration were either jointly increased or decreased with respect to our optimal configuration. This suggests that joint covariations in parameters may influence the network's optimality. While our finite Monte-Carlo random sampling does not fully prove the global optimality of the configuration in *Table 1*, it shows that it is highly efficient.

## Competition across neurons with similar stimulus tuning emerging in efficient spiking networks

We next explored coding properties emerging from recurrent synaptic interactions between E and I populations in the optimally efficient networks.

An approach that has recently provided empirical insight into local recurrent interactions is measuring effective connectivity with cellular resolution. Recent effective connectivity experiments photostimulated single E neurons in primary visual cortex and measured its effect on neighbouring neurons, finding that the photostimulation of an E neuron led to a decrease in firing rate of similarly tuned close-by neurons (*Chettih and Harvey, 2019*). This effective lateral inhibition (*Lochmann*

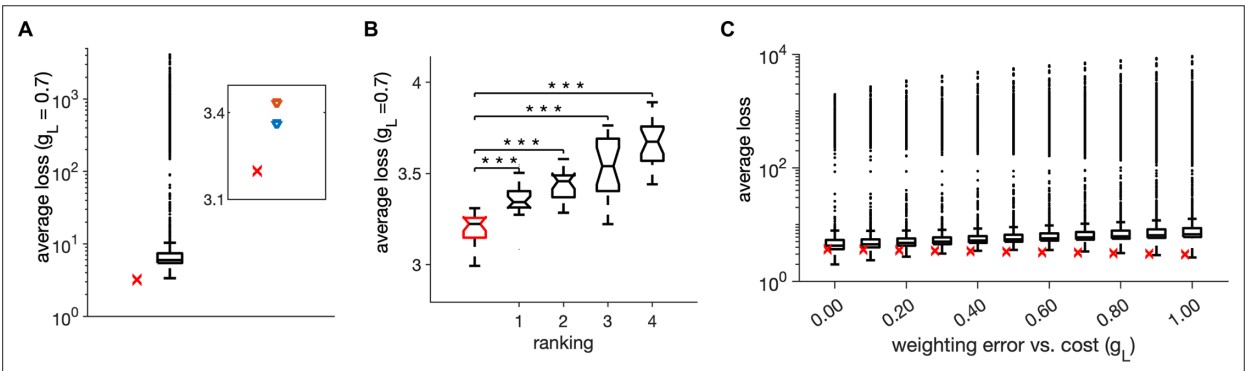

**Figure 2.** Monte-Carlo joint random sampling on six model parameters. (**A**) Distribution of the trial-averaged loss, with weighting $g_L = 0.7$, from 10,000 random simulations and using 20 simulation trials of duration of 1 s for each parameter configuration. The red cross marks the average loss of the parameter setting in *Table 1*. Inset: The average loss of the parameter setting in *Table 1* (red cross) and of the first- and second-best parameter settings from the random search. (**B**) Distribution of the average loss across 20 simulation trials for the parameter setting in *Table 1* (red) and for the first four ranked points according to the trial-averaged loss in A. Stars indicate a significant two-tailed t-test against the distribution in red (*** indicate $p < 0.001$). (**C**) Same as in A, for different values of weighting of the error with the cost $g_L$. Parameters for all plots are in *Table 1*.

**Table 3.** Table of best four parameter settings from Monte-Carlo sampling.
The performance was evaluated using trial- and time-averaged loss. Each parameter setting was evaluated on 20 trials, with each trial using an independent realization of tuning parameters, noise in the non-specific current and initial conditions for the integration of the membrane potentials. We tested 10,000 parameter settings.

| Parameter | $\tau_r^E$ | $\tau_r^I$ | $\beta$ | $\sigma$ | $N^E : N^I$ | mean I-I: mean E-I |
|---|---|---|---|---|---|---|
| First | 12.6 | 11.1 | 2.1 | 4.7 | 5.4 | 3.0 |
| Second | 11.4 | 10.0 | 2.9 | 6.2 | 6.1 | 3.3 |
| Third | 10.0 | 10.7 | 10.1 | 3.0 | 3.2 | 2.5 |
| Fourth | 12.5 | 13.5 | 2.9 | 5.4 | 4.9 | 3.5 |

*et al., 2012*) between E neurons with similar tuning to the stimulus implements competition between neurons for the representation of stimulus features (*Chettih and Harvey, 2019*). Since our model instantiates efficient coding by design and because we removed connections between neurons with different selectivity, we expected that our network implements lateral inhibition and would thus give comparable effective connectivity results in simulated photostimulation experiments.

To test this prediction, we simulated photostimulation experiments in our optimally efficient network. We first performed experiments in the absence of the feedforward input to ensure all effects are only due to the recurrent processing. We stimulated a randomly selected single target E neuron and measured the change in the instantaneous firing rate from the baseline firing rate, $\Delta z_i(t)$, in all the other I and E neurons (*Figure 3A*, left). The photostimulation was modeled as an application of a constant depolarising current with a strength parameter, $a_p$, proportional to the distance between the resting potential and the firing threshold ($a_p = 0$ means no stimulation, while $a_p = 1$ indicates photostimulation at the firing threshold). We quantified the effect of the simulated photostimulation of a target E neuron on other E and I neurons, distinguishing neurons with either similar or different tuning with respect to the target neuron (*Figure 3A*, right; *Figure 3—figure supplement 1A–D*).

The photostimulation of the target E neuron increased the instantaneous firing rate of similarly-tuned I neurons and reduced that of other similarly-tuned E neurons (*Figure 3B*). We quantified the effective connectivity as the difference between the time-averaged firing rate of the recorded cell in presence or absence of the photostimulation of the targeted cell, measured during perturbation and up to 50 ms after. We found positive effective connectivity on I and negative effective connectivity on E neurons with similar tuning to the target neuron, with a positive correlation between tuning similarity and effective connectivity on I neurons and a negative correlation on E neurons (*Figure 3C*). We confirmed these effects of photostimulation in presence of a weak feedforward input (*Figure 3—figure supplement 1E*), similar to the experiments of *Chettih and Harvey, 2019* in which photostimulation was applied during the presentation of visual stimuli with weak contrast. Thus, the optimal network replicates the preponderance of negative effective connectivity between E neurons and the dependence of its strength on tuning similarity found in *Chettih and Harvey, 2019*.

In summary, lateral excitation of I neurons and lateral inhibition of E neurons with similar tuning is an emerging coding property of the efficient E-I network, which recapitulates competition between neurons with similar stimulus tuning found in visual cortex (*Chettih and Harvey, 2019*; *Oldenburg et al., 2024*). An intuition of why this competition implements efficient coding is that the E neuron that fires first activates I neurons with similar tuning. In turn, these I neurons inhibit all similarly tuned E neurons (*Figure 3A*, right), preventing them to generate redundant spikes to encode the sensory information that has already been encoded by the first spike. Suppression of redundant spiking reduces metabolic cost without reducing encoded information (*Boerlin et al., 2013*; *Koren and Denève, 2017*).

While perturbing the activity of E neurons in our model qualitatively reproduces empirically observed lateral inhibition among E neurons (*Chettih and Harvey, 2019*; *Oldenburg et al., 2024*), these experiments have also reported positive effective connectivity between E neurons with very similar stimulus tuning. Our intuition is that our simple model cannot reproduce this finding because it lacks E-E connectivity.

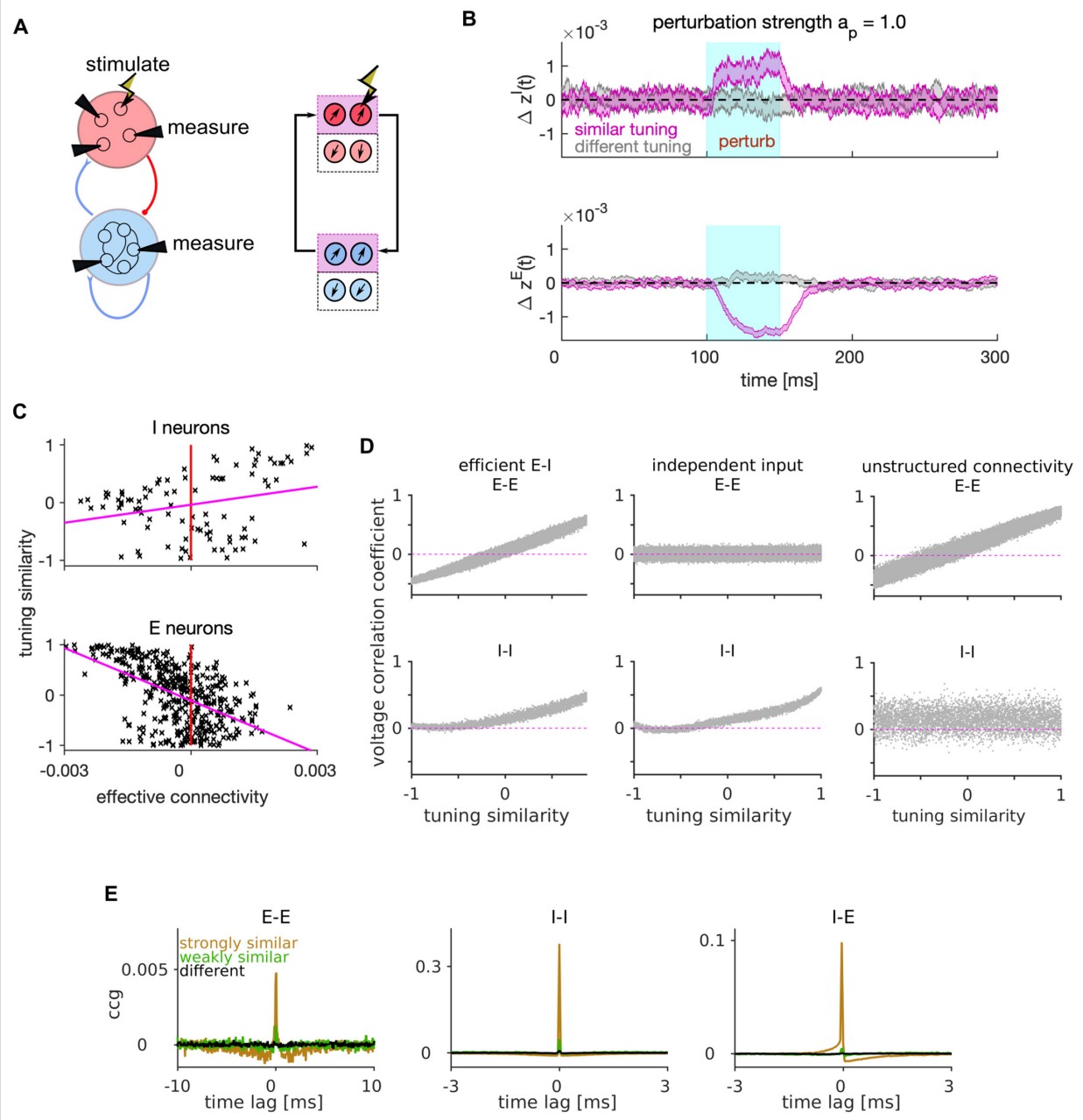

**Figure 3.** Mechanism of lateral excitation/inhibition in the efficient spiking network. (**A**) Left: Schematic of the E-I network and of the stimulation and measurement in a perturbation experiment. Right: Schematic of the propagation of the neural activity between E and I neurons with similar tuning. (**B**) Trial and neuron-averaged deviation of the firing rate from the baseline, for the population of I (top) and E (bottom) neurons with similar (magenta) and different tuning (gray) to the target neuron. Traces show the mean ± standard error of the mean, with the standard error of the mean on the variability across neurons and across trials. The stimulation strength corresponded to an increase in the firing rate of the stimulated neuron by 28.0 Hz. (**C**) Scatter plot of the tuning similarity vs. effective connectivity to the target neuron. Red line marks zero effective connectivity and magenta line is the least-squares line. Stimulation strength was $a_p = 1$. (**D**) Correlation of membrane potentials vs. the tuning similarity in E (top) and I cell type (bottom), for the efficient E-I network (left), for the network where each E neuron receives independent instead of shared stimulus features (middle), and for the network with unstructured connectivity (right). In the model with unstructured connectivity, elements of each connectivity matrix were randomly shuffled. We quantified voltage correlation using the (zero-lag) Pearson's correlation coefficient, denoted as $\rho(V_i^y(t), V_j^y(t))$, for each pair of neurons. (**E**) Average cross-correlogram (CCG) of spike timing with strongly similar (orange), weakly similar (green) and different tuning (black). Statistical results (**B–E**) were computed on 100 simulation trials. The duration of the trial in D-E was 1 s. Parameters for all plots are in **Table 1**.

The online version of this article includes the following figure supplement(s) for figure 3:

**Figure supplement 1.** Tuning similarity and its relation to lateral excitation/inhibition.

To explore further the consequences of E-I interactions for stimulus encoding, we next investigated the dynamics of lateral inhibition in the optimal network driven by the feedforward sensory input but without perturbing the neural activity. Previous work has established that efficient spiking neurons may present strong correlations in the membrane potentials, but only weak correlations in the spiking output, because redundant spikes are prevented by lateral inhibition (*Boerlin et al., 2013*; *Deneve and Machens, 2016b*). We investigated voltage correlations in pairs of neurons within our network as a function of their tuning similarity. Because the feedforward inputs are shared across E neurons and weighted by their tuning parameters, they cause strong positive voltage correlations between E-E neuronal pairs with very similar tuning and strong negative correlations between pairs with very different (opposite) tuning (*Figure 3D*, top-left). Voltage correlations between E-E pairs vanished regardless of tuning similarity when we made the feedforward inputs independent across neurons (*Figure 3D*, top-middle), showing that the dependence of voltage correlations on tuning similarity occurs because of shared feedforward inputs. In contrast to E neurons, I neurons do not receive feedforward inputs and are driven only by similarly tuned E neurons (*Figure 3A*, right). This causes positive voltage correlations in I-I neuronal pairs with similar tuning and vanishing correlations in neurons with different tuning (*Figure 3D*, bottom-left). Such dependence of voltage correlations on tuning similarity disappears when removing the structure from the E-I synaptic connectivity (*Figure 3D*, bottom-right).

In contrast to voltage correlations, and as expected by previous studies (*Boerlin et al., 2013*; *Deneve and Machens, 2016b*), the coordination of spike timing of pairs of E neurons (measured with cross-correlograms or CCGs) was very weak (*Figure 3E*). For I-I and E-I neuronal pairs, the peaks of CCGs were stronger than those observed in E-E pairs, but they were present only at very short lags (lags < 1 ms). This confirms that recurrent interactions of the efficient E-I network wipe away the effect of membrane potential correlations at the spiking output level, and shows information processing with millisecond precision in these networks (*Boerlin et al., 2013*; *Deneve and Machens, 2016b*; *Koren and Denève, 2017*).

## The effect of structured connectivity on coding efficiency and neural dynamics

The analytical solution of the optimally efficient E-I network predicts that recurrent synaptic weights are proportional to the tuning similarity between neurons. We next investigated the role of such connectivity structure by comparing the behavior of an efficient network with an unstructured E-I network, similar to the type studied in previous works (*Brunel, 2000*; *Renart et al., 2010*; *Mazzoni et al., 2008*). We removed the connectivity structure by randomly permuting synaptic weights across neuronal pairs (see Materials and methods). Such shuffling destroys the relationship between tuning similarity and synaptic strength (as shown in *Figure 1C(ii)*) while it preserves Dale's law and the overall distribution of connectivity weights.

We found that shuffling the connectivity structure significantly altered the efficiency of the network (*Figure 4A, B*), neural dynamics (*Figure 4C, D, F–H*) and lateral inhibition (*Figure 4I*). In particular, structured networks differ from unstructured ones by showing better encoding performance (*Figure 4A*), lower metabolic cost (*Figure 4B*), weaker variance of the membrane potential over time (*Figure 4C*), lower firing rates (*Figure 4D*) and weaker average (*Figure 4F*) and instantaneous balance (*Figure 4G*) of synaptic inputs. However, we found only a small difference in the variability of spiking between structured and unstructured networks (*Figure 4E*). While these results are difficult to test experimentally due to the difficulty of manipulating synaptic connectivity structures in vivo, they highlight the importance of the connectivity structure for cortical computations.

We also compared structured and unstructured networks about their relation between pairwise voltage correlations and tuning similarity, by randomizing connections within a single connectivity type (E-I, I-I or I-E) or within all these three connectivity types at once ('all'). We found the structure of E-I connectivity to be crucial for the linear relation between voltage correlations and tuning similarity in pairs of I neurons (*Figure 4H*, magenta).

Finally, we analyzed how the structure in recurrent connectivity influences lateral inhibition that we observed in efficient networks. We found that the dependence of lateral inhibition on tuning similarity vanishes when the connectivity structure is fully removed (*Figure 4I*, 'all' on the right plot), thus showing that connectivity structure is necessary for lateral inhibition. While networks with unstructured E-I and I-E connectivity still show inhibition in E neurons upon single neuron photostimulation (because of the

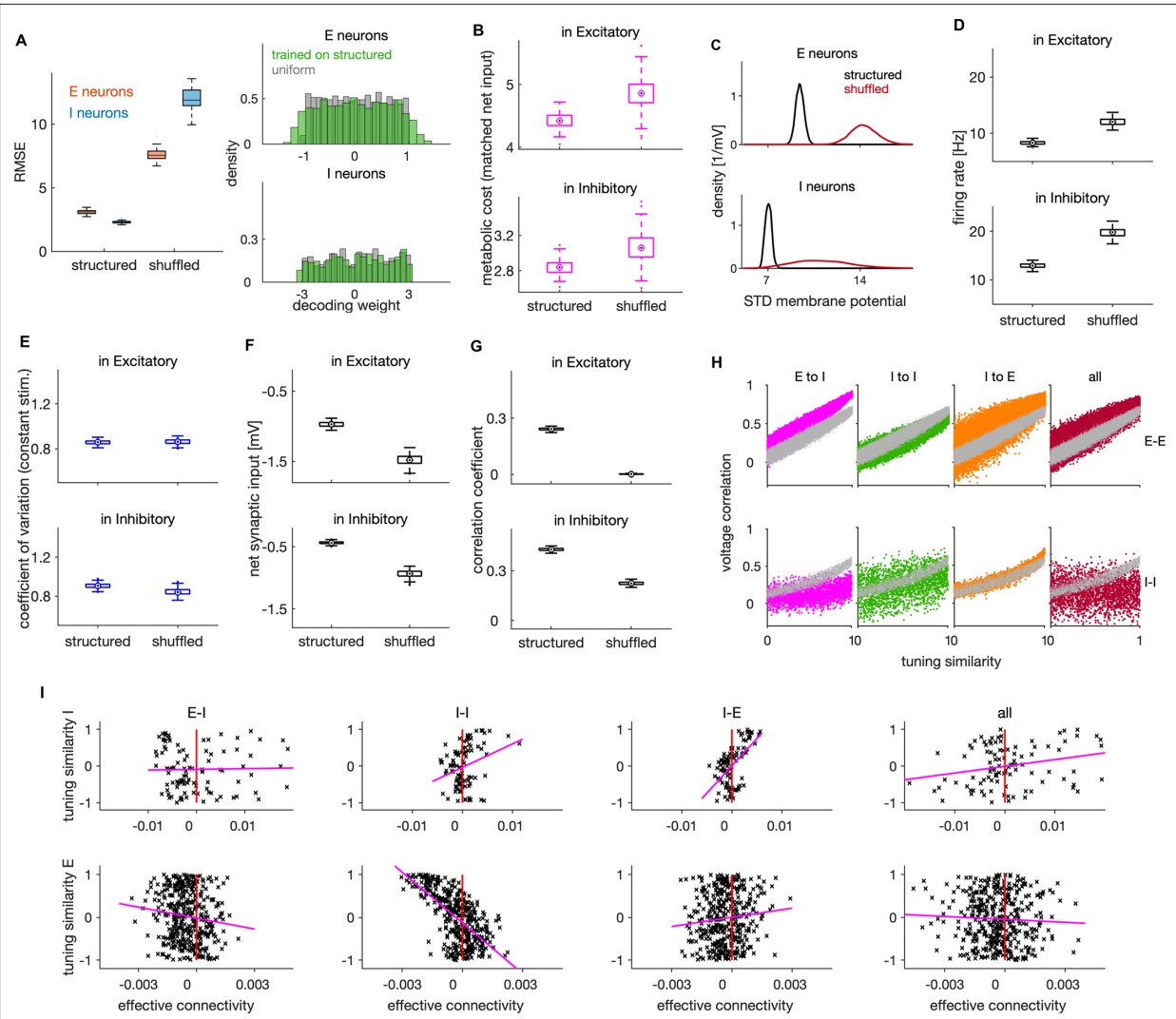

**Figure 4.** Effects of connectivity structure on coding efficiency, neural dynamics and lateral inhibition. (**A**) Left: Root mean squared error (RMSE) in networks with structured and randomly shuffled recurrent connectivity. Random shuffling consisted of a random permutation of the elements within each of the three (**E–I, I–I, I–E**) connectivity matrices. Right: Distribution of decoding weights after training the decoder on neural activity from the structured network (green), and a sample from uniform distribution as typically used in the optimal network. (**B**) Metabolic cost in structured and shuffled networks with matched average balance. The average balance of the shuffled network was matched with the one of the structured network by changing the following parameters: $\tilde{\beta} = 16.3$, $\tilde{\sigma} = 3$ and by decreasing the amplitude of the OU stimulus by factor of 0.88. (**C**) Standard deviation of the membrane potential (in mV) for networks with structured and unstructured connectivity. Distributions are across neurons. (**D**) Average firing rate of E (top) and I neurons (bottom) in networks with structured and unstructured connectivity. (**E**) Same as in D, showing the coefficient of variation of spiking activity in a network responding to a constant stimulus. (**F**) Same as in D, showing the average net synaptic input, a measure of average imbalance. (**G**) Same as in D, showing the time-dependent correlation of synaptic inputs, a measure of instantaneous balance. (**H**) Voltage correlation in E-E (top) and I-I neuronal pairs (bottom) for the four cases of unstructured connectivity (colored dots) and the equivalent result in the structured network (grey dots). We show the results for pairs with similar tuning. (**I**) Scatter plot of effective connectivity versus tuning similarity to the photostimulated E neuron in shuffled networks. The title of each plot indicates the connectivity matrix that has been shuffled. The magenta line is the least-squares regression line and the photostimulation is at threshold ($a_p = 1.0$). Results were computed using 200 (**A–G**) and 100 (**H–I**) simulation trials of 1 s duration. Parameters for all plots are in *Table 1*.

The online version of this article includes the following figure supplement(s) for figure 4:

**Figure supplement 1.** Effect of removal of connectivity structure and of jittering of synaptic weights.

net inhibitory effect of recurrent connectivity; *Figure 4—figure supplement 1F*), this inhibition was largely unspecific to tuning similarity (*Figure 4I*, 'E-I' and 'I-E'). Unstructured connectivity decreased the correlation between tuning similarity and effective connectivity from $r = [0.31, -0.54]$ in I and E neurons in a structured network to $r = [0.02, -0.13]$ and $r = [0.57, 0.11]$ in networks with unstructured E-I and I-E connectivity, respectively. Removing the structure in I-I connectivity, in contrast, increased the correlation between effective connectivity and tuning similarity in E neurons ($r = [0.30, -0.65]$, *Figure 4I*, second from the left), showing that lateral inhibition takes place irrespectively of the I-I connectivity structure.

Previous empirical (*Znamenskiy et al., 2024*) and theoretical work has established the necessity of strong E-I-E synaptic connectivity for lateral inhibition (*Sadeh and Clopath, 2020*; *Mackwood et al., 2021*). To refine this understanding, we asked what is the minimal connectivity structure necessary to qualitatively replicate empirically observed lateral inhibition. We did so by considering a simpler connectivity rule than the one obtained from first principles. We assumed neurons to be connected (with random synaptic efficacy) if their tuning vectors are similar ($J_{ij}^{xy} > 0$ if $\phi_{ij}^{xy} > 0$) and unconnected otherwise ($J_{ij}^{xy} = 0$ if $\phi_{ij}^{xy} \leq 0$), relaxing the precise proportionality relationship between tuning similarity and synaptic weights (as on *Figure 1C(ii)*). We found that networks with such simpler connectivity respond to activity perturbation in a qualitatively similar way as the optimal network (*Figure 3—figure supplement 1F*) and still replicate experimentally observed activity profiles in *Chettih and Harvey, 2019*.

While optimally structured connectivity predicted by efficient coding is biologically plausible, it may be difficult to realise it exactly on a synapse-by-synapse basis in biological networks. Following *Calaim et al., 2022*, we verified the robustness of the model to small deviations from the optimal synaptic weights by adding a random jitter, proportional to the synaptic strength, to all synaptic connections (see Materials and methods). The encoding performance and neural activity were barely affected by weak and moderate levels of such perturbation (*Figure 4—figure supplement 1G, H*), demonstrating that the network is robust against random jittering of the optimal synaptic weights.

In summary, we found that some aspects of recurrent connectivity structure, such as the like-to-like organization, are crucial to achieve efficient coding. Instead, for other aspects there is considerable flexibility; the proportionality between tuning similarity and synaptic weights is not crucial for efficiency and small random jitter of optimal weights has only minor effects. Structured E-I and I-E, but not I-I connectivity, is necessary for implementing experimentally observed pattern of lateral inhibition whose strength is modulated by tuning similarity.

## Weak or no spike-triggered adaptation optimizes network efficiency

We next investigated the role of within-neuron feedback triggered by each spike, $I_i^{\mathrm{ad},y}$, that emerges from the optimally efficient solution (*Equation 5*). A previous study (*Gutierrez and Denève, 2019*) showed that spike-triggered adaptation, together with structured connectivity, redistributes the activity from highly excitable neurons to less excitable neurons, leaving the population readout invariant. Here, we address model efficiency in presence of adapting or facilitating feedback as well as differential effects of adaptation in E and I neurons.

The spike-triggered within-neuron feedback $I_i^{\mathrm{ad},y}$ has a time constant equal to that of the single neuron readout $\tau_r^E$ (E neurons) and $\tau_r^I$ (I neurons). The strength of the current is proportional to

**Table 4.** Relation of time constants of single-neuron and population readout set an adaptation or a facilitation current.

The population readout that evolves on a faster (slower) time scale than the single neuron readout determines a spike-triggered adaptation (facilitation) in its own cell type.

| Relative speed | Relation of time constants | Current |
|---|---|---|
| $\hat{x}^E$ faster than $r^E$ | $\tau < \tau_r^E$ | adaptation in E |
| $\hat{x}^E$ slower than $r^E$ | $\tau > \tau_r^E$ | facilitation in E |
| $\hat{x}^I$ faster than $r^I$ | $\tau < \tau_r^I$ | adaptation in I |
| $\hat{x}^I$ slower than $r^I$ | $\tau > \tau_r^I$ | facilitation in I |

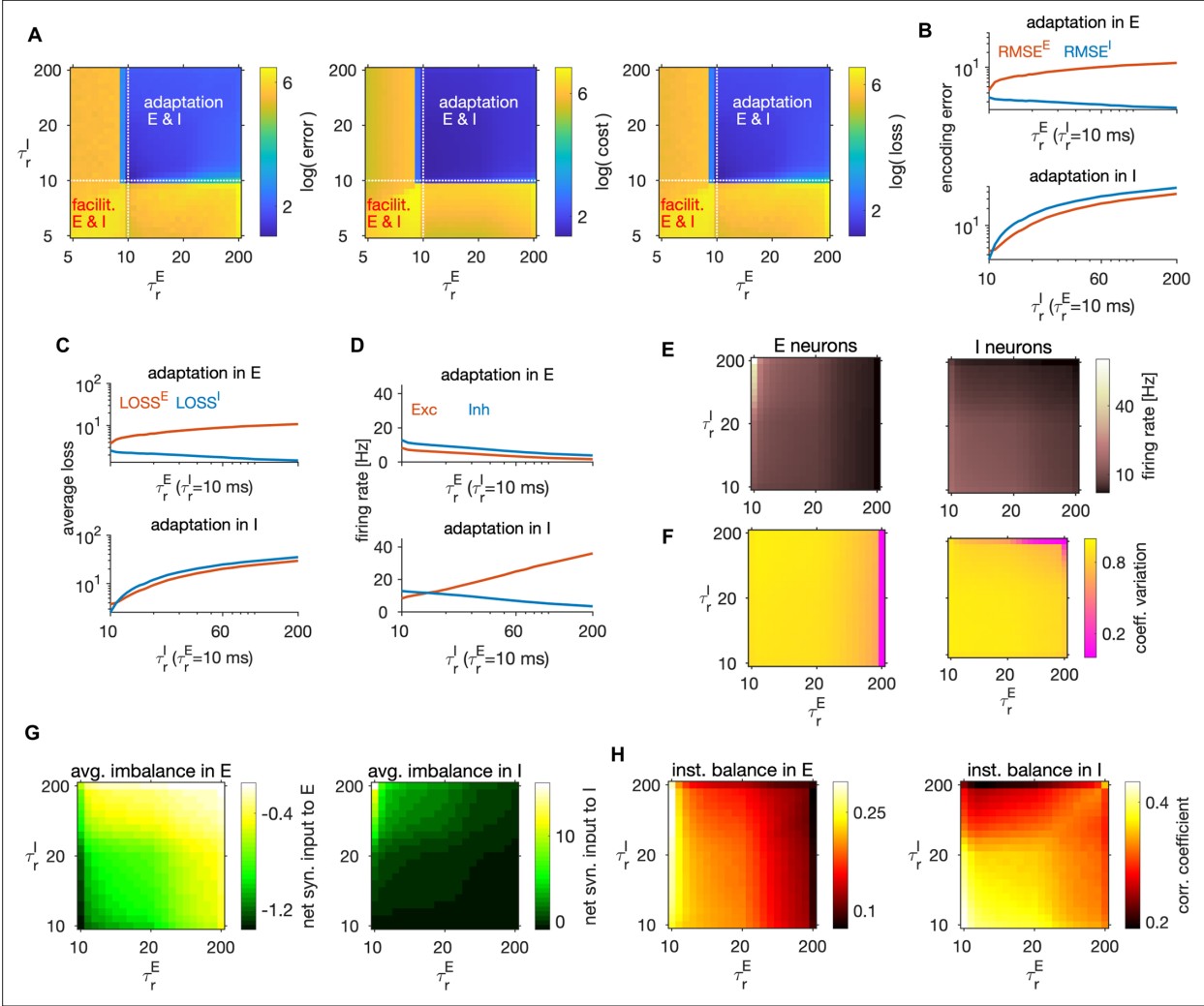

**Figure 5.** Adaptation, network coding efficiency and excitation-inhibition balance. (**A**) The encoding error (left), metabolic cost (middle) and average loss (right) as a function of single neuron time constants $\tau_r^E$ (E neurons) and $\tau_r^I$ (I neurons), in units of ms. These parameters set the sign, the strength, as well as the time constant of the feedback current in E and I neurons. Best performance (lowest average loss) is obtained in the top right quadrant, where the feedback current is spike-triggered adaptation in both E and I neurons. The performance measures are computed as a weighted sum of the respective measures across the E and I populations with equal weighting for E and I. All measures are plotted on the scale of the natural logarithm for better visibility. (**B**) Top: Log-log plot of the RMSE of the E (red) and the I (blue) estimates as a function of the time constant of the single neuron readout of E neurons, $\tau_r^E$, in the regime with spike-triggered adaptation. Feedback current in I neurons is set to 0. Bottom: Same as on top, as a function of $\tau_r^I$ while the feedback current in E neurons is set to 0. (**C**) Same as in B, showing the average loss. (**D**) Same as in B, showing the firing rate. (**E**) Firing rate in E (left) and I neurons (right), as a function of time constants $\tau_r^E$ and $\tau_r^I$. (**F**) Same as in E, showing the coefficient of variation. (**G**) Same as E, showing the average net synaptic input, a measure of average imbalance. (**H**) Same as E, showing the average net synaptic input, a measure of instantaneous balance. All statistical results were computed on 100 simulation trials of 1 s duration. For other parameters, see **Table 1**.

the difference in inverse time constants of single neuron and population readouts, $1/\tau - 1/\tau_r^y$. This spike-triggered current is negative, giving spike-triggered adaptation (**Mensi et al., 2012**), if the single-neuron readout has longer time constant than the population readout ($\tau_r^y > \tau$), or positive, giving spike-triggered facilitation, if the opposite is true ($\tau_r^y < \tau$; **Table 4**). We expected that network efficiency would benefit from spike-triggered adaptation, because accurate encoding requires fast temporal dynamics of the population readouts, to capture fast fluctuations in the target signal, while we expect a slower dynamics in the readout of single neuron's firing frequency, $r_i^y(t)$, a process that could be related to homeostatic regulation of single neuron's firing rate (**Abbott and Nelson, 2000**; **Turrigiano and Nelson, 2004**). In our optimal E-I network we indeed found that optimal coding efficiency is achieved in absence of within-neuron feedback or with weak adaptation in both cell types

(*Figure 5A*). The optimal set of time constants $[\tau_r^E, \tau_r^I]$ only weakly depended on the weighting of the encoding error with the metabolic cost $g_L$ (*Figure 7—figure supplement 1A*). We note that adaptation in E neurons promotes efficient coding because it enforces every spike to be error-correcting, while a spike-triggered facilitation in E neurons would lead to additional spikes that might be redundant and reduce network efficiency. Contrary to previously proposed models of adaptation in LIF neurons (*Brette and Gerstner, 2005*; *Schwalger and Lindner, 2013*), the strength and the time constant of adaptation in our model are not independent, but they both depend on $\tau_r^y$, with larger $\tau_r^y$ yielding both longer and stronger adaptation.

To gain insights on the differential effect of adaptation in E vs I neurons, we set the adaptation in one cell type to 0 and varied the strength of adaptation in the other cell type by varying the time constant of the single neuron readout. With adaptation in E neurons (and no adaptation in I), we observed a slow increase of the encoding error in E neurons, while the encoding error increased faster with adaptation in I neurons (*Figure 5B*). Similarly, network efficiency increased slowly with adaptation in E and faster with adaptation in I neurons (*Figure 5C*), thus showing that adaptation in E neurons decreases less the performance compared to the adaptation in I neurons. With increasing adaptation in E neurons, the firing rate in E neurons decreased (*Figure 5D*), leading to E estimates with smaller amplitude. Because E estimates are target signals for I neurons and because weaker E signals imply weaker drive to I neurons, average loss of the I population decreased by increasing adaptation in E neurons (*Figure 5C* top, blue trace).

Firing rates and variability of spiking were sensitive to the strength of adaptation. As expected, adaptation in E neurons caused a decrease in the firing levels in both cell types (*Figure 5D, E*). In contrast, adaptation in I neurons decreased the firing rate in I neurons, but increased the firing rate in E neurons, due to a decrease in the level of inhibition. Furthermore, adaptation decreased the variability of spiking, in particular in the cell type with strong adaptation (*Figure 5F*), a well-known effect of spike-triggered adaptation in single neurons (*Schwalger and Lindner, 2013*).

In regimes with adaptation, time constants of single neuron readout $\tau_r^y$ influenced the average balance (*Figure 5G*) as well as the instantaneous balance (*Figure 5H*) in E and I cell type. To gain a better understanding of the relationship between adaptation, E-I interactions and network optimality, we measured the instantaneous and time-averaged E-I balance while varying the adaptation parameters and studied their relation with the loss. By increasing adaptation in E neurons, the average imbalance got weaker in E neurons (*Figure 5G*, left), but stronger in I neurons (*Figure 5G*, right). Regimes with precise average balance in both cell types were suboptimal (compare *Figure 5A*, right and G), while regimes with precise instantaneous balance were highly efficient (compare *Figure 5A*, right and H).

To test how well the average balance and the instantaneous balance of synaptic inputs predict network efficiency, we concatenated the column-vectors of the measured average loss and of the average imbalance in each cell type and computed the Pearson correlation between these quantities. The correlation between the average imbalance and the average loss was weak in the E cell type ($r = 0.16$) and close to zero in the I cell type ($r = 0.02$), suggesting almost no relation between efficiency and average imbalance. In contrast, the average loss was negatively correlated with the instantaneous balance in both E ($r = -0.35$) and in I cell type ($r = -0.45$), showing that instantaneous balance of synaptic inputs is positively correlated with network efficiency. When measured for varying levels of spike-triggered adaptation, unlike the average balance of synaptic inputs, the instantaneous balance is thus mildly predictive of network efficiency.

In sum, our results show that the optimally efficient solution does not include within-neuron feedback, while a model with weak and short-lasting spike-triggered adaptation is slightly suboptimal, although still highly efficient. Our results predict that information coding would be more efficient with adaptation than with facilitation. Assuming that our I neurons describe parvalbumin-positive interneurons, our results suggest that the weaker adaptation in I compared to E neurons, reported empirically (*Pala and Petersen, 2015*), may be beneficial for the network's encoding efficiency.

Spike-triggered adaptation in our model captures adaptive processes in single neurons that occur on time scales lasting from a couple of milliseconds to tens of milliseconds after each spike. However, spiking in biological neurons triggers adaptation on multiple time scales, including much slower time scales on the order of seconds or tens of seconds (*Pozzorini et al., 2013*). Our model does not capture adaptive processes on these longer time scales (but see *Gutierrez and Denève, 2019*).

## Non-specific currents regulate network coding properties

In our derivation of the optimal network, we obtained a non-specific external current (in the following, non-specific current) $I_i^{\mathrm{ext},y}(t)$. Non-specific current captures all synaptic currents that are unrelated and unspecific to the stimulus features. This non-specific term collates effects of synaptic currents from neurons untuned to the stimulus (*Levy et al., 2020*; *Zylberberg, 2018*), as well as synaptic currents from other brain areas. It can be conceptualized as the background synaptic activity that provides a large fraction of all synaptic inputs to both E and I neurons in cortical networks (*Destexhe and Paré, 1999*), and which may modulate feedforward-driven responses by controlling the distance between the membrane potential and the firing threshold (*Destexhe et al., 2003*). Likewise, in our model, the non-specific current does not directly convey information about the feedforward input features, but influences the network dynamics.

Non-specific current comprises mean and fluctuations (see Materials and methods). The mean is proportional to the metabolic constant $\beta$ and its fluctuations reflect the noise that we included in the condition for spiking. Since $\beta$ governs the trade-off between encoding error and metabolic cost (*Equation 3*), higher values of $\beta$ imply that more importance is assigned to the metabolic efficiency than to coding accuracy, yielding a reduction in firing rates. In the expression for the non-specific current, we found that the mean of the current is negatively proportional to the metabolic constant $\beta$ (see Materials and methods). Because the non-specific current is typically depolarizing, this means that increasing $\beta$ yields a weaker non-specific current and increases the distance between the mean membrane potential and the firing threshold. Thus, an increase of the metabolic constant is expected to make the network less responsive to the feedforward signal.

We found the metabolic constant $\beta$ to significantly influence the spiking dynamics (*Figure 6A*). The optimal efficiency was achieved for non-zero levels of the metabolic constant (*Figure 6B*), with the mean of the non-specific current spanning more than half of the distance between the resting potential and the threshold (*Table 1*). Stronger weighting of the loss of I compared to E neurons and stronger weighting of the error compared to the cost yielded weaker optimal metabolic constant (*Figure 7—figure supplement 1B*). Metabolic constant modulated the firing rate as expected, with the firing rate in E and I neurons decreasing with the increasing of the metabolic constant (*Figure 6C*, top). It also modulated the variability of spiking, as increasing the metabolic constant decreased the variability of spiking in both cell types (*Figure 6C*, bottom). Furthermore, it modulated the average balance and the instantaneous balance in opposite ways: larger values of $\beta$ led to regimes that had stronger average balance, but weaker instantaneous balance (*Figure 6D*). We note that, even with suboptimal values of the metabolic constant, the neural dynamics remained within biologically relevant ranges.

The fluctuation part of the non-specific current, modulated by the noise strength $\sigma$ that we added in the definition of spiking rule for biological plausibility (see Materials and methods), strongly affected the neural dynamics as well (*Figure 6E*). The optimal performance was achieved with non-vanishing noise levels (*Figure 6F*), similarly to previous work showing that the noise prevents excessive network synchronization that would harm performance (*Chalk et al., 2016*; *Koren and Denève, 2017*; *Timcheck et al., 2022*). The optimal noise strength depended on the weighting of the error with the cost, with strong weighting of the error predicting stronger noise (*Figure 7—figure supplement 1C*).

The average firing rate of both cell types, as well as the variability of spiking in E neurons, increased with noise strength (*Figure 6G*), and some level of noise in the non-specific inputs was necessary to establish the optimal level of spiking variability. Nevertheless, we measured significant levels of spiking variability already in the absence of noise, with a coefficient of variation of about 0.8 in E and 0.9 in I neurons (*Figure 6G*, bottom). This indicates that the recurrent network dynamics generates substantial variability even in absence of an external source of noise. The average and instantaneous balance of synaptic currents exhibited a non-linear behavior as a function of noise strength (*Figure 6H*). Due to decorrelation of membrane potentials by the noise, instantaneous balance in I neurons decreased with increasing noise strength (*Figure 6H*, bottom).

Next, we investigated the joint impact of the metabolic constant and the noise strength on network optimality. We expect these two parameters to be related, because larger noise strength requires stronger metabolic constant to prevent the activity of the network to be dominated by noise. We thus performed a two-dimensional parameter search (*Figure 6I*). As expected, the optima of the metabolic constant and the noise strength were positively correlated. A weaker noise required lower metabolic

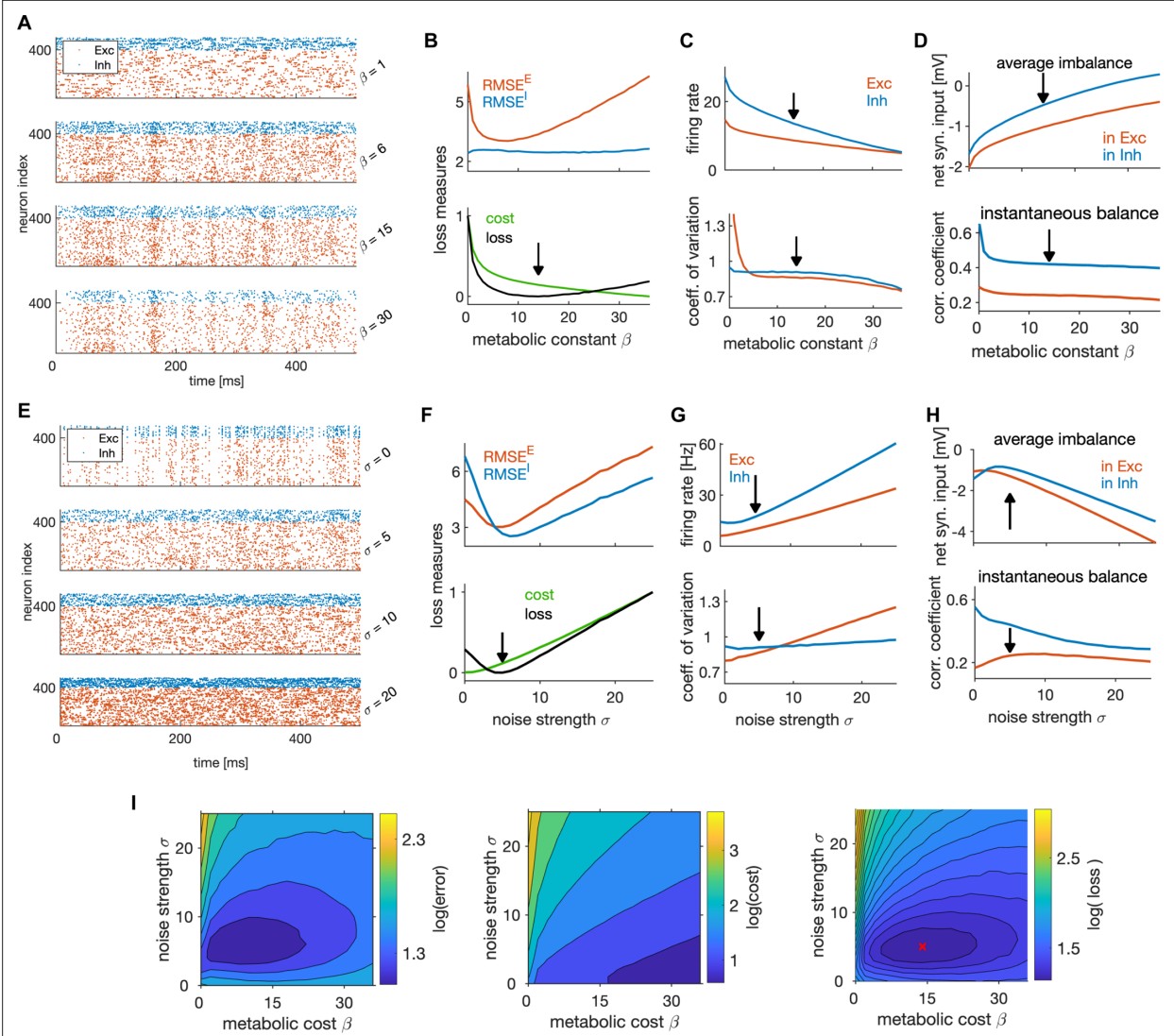

**Figure 6.** State-dependent coding and dynamics are controlled by non-specific currents. (**A**) Spike trains of the efficient E-I network in one simulation trial, with different values of the metabolic constant $\beta$. The network received identical stimulus across trials. (**B**) Top: RMSE of E (red) and I (blue) estimates as a function of the metabolic constant. Bottom: Normalized average metabolic cost and average loss as a function of the metabolic constant. Black arrow indicates the minimum loss and therefore the optimal metabolic constant. (**C**) Average firing rate (top) and the coefficient of variation of the spiking activity (bottom), as a function of the metabolic constant. Black arrow marks the metabolic constant leading to optimal network efficiency in B. (**D**) Average imbalance (top) and instantaneous balance (bottom) balance as a function of the metabolic constant. (**E**) Same as in A, for different values of the noise strength $\sigma$. (**F**) Same as in B, as a function of the noise strength. The noise is a Gaussian random process, independent over time and across neurons. (**G**) Same as C, as a function of the noise strength. (**H**) Top: Same as in D, as a function of the noise strength. (**I**) The encoding error measured as RMSE (left), the metabolic cost (middle) and the average loss (right) as a function of the metabolic constant $\beta$ and the noise strength $\sigma$. Metabolic constant and noise strength that are optimal for the single parameter search (in B and F) are marked with a red cross in the figure on the right. For plots in B-D and F-I, we computed and averaged results over 100 simulation trials with 1 second duration. For other parameters, see **Table 1**.

constant, and-vice-versa. While achieving maximal efficiency at non-zero levels of the metabolic cost and noise (see **Figure 6I**) might seem counterintuitive, we speculate that such setting is optimal because some noise in the non-specific current prevents over-synchronization and over-regularity of firing that would harm efficiency, similarly to what was shown in previous works (**Chalk et al., 2016**; **Koren and Denève, 2017**; **Timcheck et al., 2022**). In the presence of noise, a non-zero metabolic constant is needed to suppress inefficient spikes purely induced by noise that do not contribute to coding and increase the error. This gives rise to a form of stochastic resonance, where an optimal level of noise is helpful to detect the signal coming from the feedforward currents.

In summary, non-specific external currents derived in our optimal solution have a major effect on coding efficiency and on neural dynamics. In qualitative agreement with empirical measurements (*Destexhe and Paré, 1999*; *Destexhe et al., 2003*), our model predicts that more than half of the average distance between the resting potential and firing threshold is accounted for by non-specific synaptic currents. Similarly to previous theoretical work (*Chalk et al., 2016*; *Koren and Denève, 2017*), we find that some level of external noise, in the form of a random fluctuation of the non-specific synaptic current, is beneficial for network efficiency. This remains a prediction for experiments.

## Optimal ratio of E-I neuron numbers and of the mean I-I to E-I synaptic efficacy coincide with biophysical measurements

Next, we investigated how coding efficiency and neural dynamics depend on the ratio of the number of E and I neurons ($N^E : N^I$ or E-I ratio) and on the relative synaptic strengths between E-I and I-I connections.

Efficiency objectives (*Equation 3*) are based on population, rather than single-neuron activity. Our efficient E-I network thus realizes a computation of the target representation that is distributed across multiple neurons (*Figure 7A*). Following previous reports (*Barrett et al., 2016*), we predict that, if the number of neurons within the population decreases, neurons have to fire more spikes to achieve an optimal population readout because the task of tracking the target signal is distributed among fewer neurons. To test this prediction, we varied the number of I neurons while keeping the number of E neurons constant. As predicted, a decrease of the number of I neurons (and thus an increase in the ratio of the number of E to I neurons) caused a linear increase in the firing rate of I neurons, while the firing rate of E neurons stayed constant (*Figure 7B*, top). However, the variability of spiking and the average synaptic inputs remained relatively constant in both cell types as we varied the E-I ratio (*Figure 7B*, bottom, C), indicating a compensation for the change in the ratio of E-I neuron numbers through adjustment in the firing rates. These results are consistent with the observation in neuronal cultures of a linear change in the rate of postsynaptic events but unchanged postsynaptic current in either E and I neurons for variations in the E-I neuron number ratio (*Sukenik et al., 2021*).

The ratio of the number of E to I neurons had a significant influence on coding efficiency. We found a unique minimum of the encoding error of each cell type, while the metabolic cost increased linearly with the ratio of the number of E and I neurons (*Figure 7D*). Using the usual weighting $g_L = 0.7$, we found the optimal ratio of E to I neuron numbers to be in range observed experimentally in cortical circuits (*Figure 7D*, bottom, black arrow, $N^E : N^I = 3.75 : 1$; *Markram et al., 2004*). The optimal ratio depended on the weighting of the error with the cost, decreasing when increasing the cost of firing (*Figure 7E*, bottom). Also the encoding error (RMSE) alone, without considering the metabolic cost, predicted optimal ratio of the number of E to I neurons within a plausible physiological range, $N^E : N^I = [3.75 : 1, 5.25 : 1]$, with stronger weightings of the encoding error by I neurons predicting higher ratios (*Figure 7E*, top).

Next, we investigated the impact of the strength of E and I synaptic efficacy (EPSPs and IPSPs). As evident from the expression for the population readouts (*Equation 2*), the magnitude of tuning parameters (which are also decoding weights) determines the amplitude of jumps of the population readout caused by spikes (*Figure 7F*). The larger these weights are, the larger is the impact of spikes on the population signals.

E and I synaptic efficacies depend on the tuning parameters. We parametrized the distribution of tuning parameters as uniform distributions centered at zero, but allowed the spread of distributions in E and I neurons ($\sigma_w^E$ and $\sigma_w^I$) to vary across E and I cell type (Materials and methods). In the optimally efficient network, as found analytically (Materials and methods section 'Dynamic equations for the membrane potentials'), the E-I connectivity is the transpose of the of the I-E connectivity, which implies that these connectivities are exactly balanced and have the same mean. We also showed analytically that by parametrizing tuning parameters with uniform distributions, the scaling of synaptic connectivity of E-I (equal to I-E) and I-I connectivity is controlled by the variance of tuning parameters of the pre and postsynaptic population as follows: $\langle J^{xy} \rangle \propto \sigma_w^x \sigma_w^y$. Using these insights, we were able to analytically evaluate the mean E-I and I-I synaptic efficacy (see Materials and methods section 'Parametrization of synaptic connectivity').

We next searched for the optimal ratio of the mean I-I to E-I efficacy as the parameter that maximizes network efficiency. Network efficiency was maximized when such ratio was about 3–1 (*Figure 7G*). Our

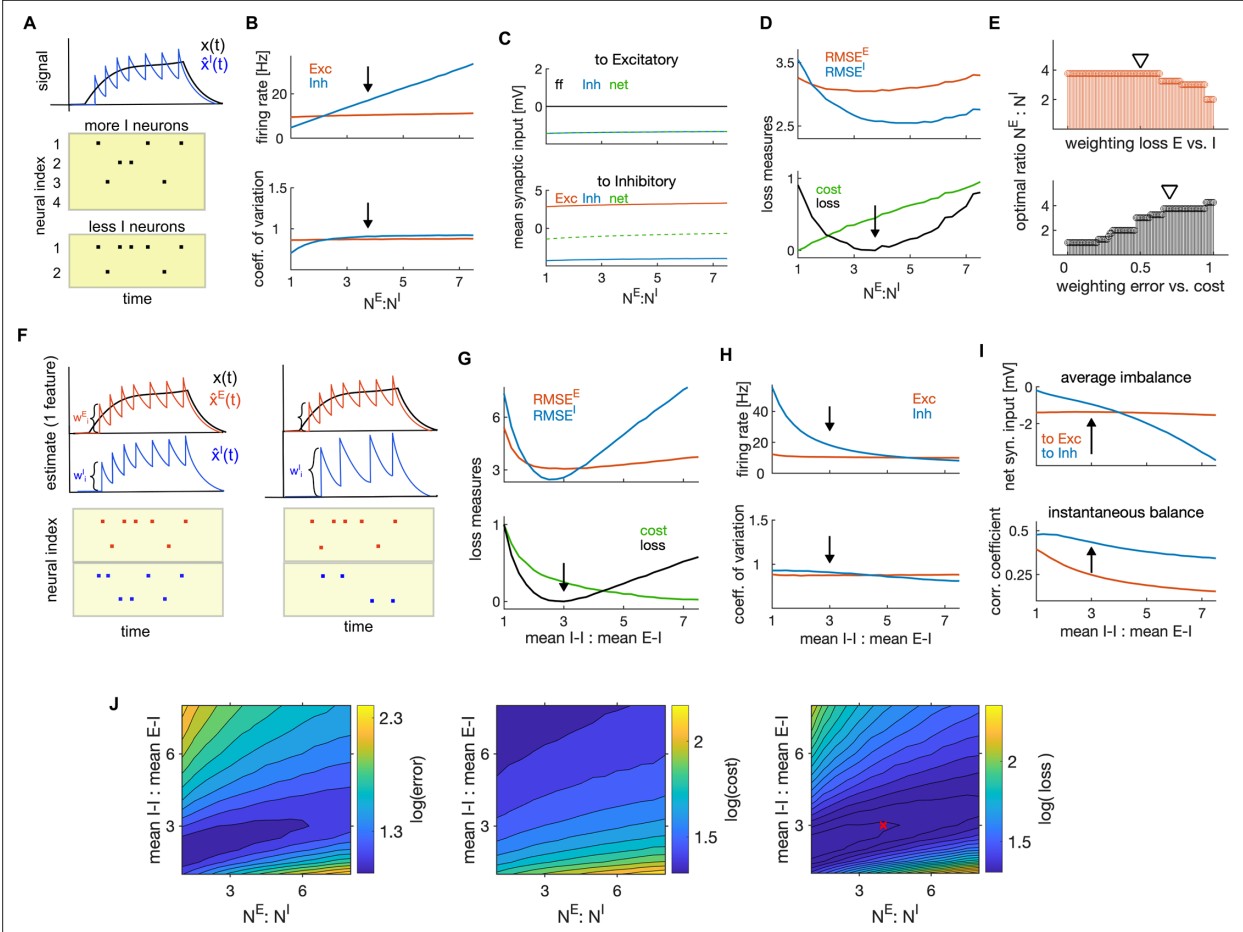

**Figure 7.** Optimal ratios of E-I neuron numbers and of mean I-I to E-I efficacy. (**A**) Schematic of the effect of changing the number of I neurons on firing rates of I neurons. As encoding of the stimulus is distributed among more I neurons, the number of spikes per I neuron decreases. (**B**) Average firing rate as a function of the ratio of the number of E to I neurons. Black arrow marks the optimal ratio. (**C**) Average net synaptic input in E neurons (top) and in I neurons (bottom). (**D**) Top: Encoding error (RMSE) of the E (red) and I (blue) estimates, as a function of the ratio of E-I neuron numbers. Bottom: Same as on top, showing the cost and the average loss. Black arrow shows the minimum of the loss, indicating the optimal parameter. (**E**) Top: Optimal ratio of the number of E to I neurons as a function of the weighting of the average loss of E and I cell type (using the weighting of the error and cost of 0.7 and 0.3, respectively). Bottom: Same as on top, measured as a function of the weighting of the error and the cost when computing the loss. (The weighting of the losses of E and I neurons is 0.5.) Black triangles mark weightings that we typically used. (**F**) Schematic of the readout of the spiking activity of E (red) and I population (blue) with equal amplitude of decoding weights (left) and with stronger decoding weight in I neuron (right). Stronger decoding weight in I neurons results in a stronger effect of spikes on the readout, leading to less spikes by the I population. (**G–H**) Same as in D and B, as a function of the ratio of mean I-I to E-I efficacy. (**I**) Average imbalance (top) and instantaneous balance (bottom) balance, as a function of the ratio of mean I-I to E-I efficacy. (**J**) The encoding error (RMSE; left) the metabolic cost (middle) and the average loss (right) as a function of the ratio of E-I neuron numbers and the ratio of mean I-I to E-I connectivity. The optimal ratios obtained with single parameter search (in D and G) are marked with a red cross. All statistical results were computed on 100 simulation trials of 1 second duration. For other parameters, see *Table 1*.

The online version of this article includes the following figure supplement(s) for figure 7:

**Figure supplement 1.** Dependence of optimal parameters on weighting of the encoding error and the metabolic cost and analysis of mean ratio of I-I to E-I connectivity by varying the number of E neurons.

results suggest the optimal E-I and I-E synaptic efficacy, averaged across neuronal pairs, of 0.75 mV, and the optimal I-I efficacy of 2.25 mV, values that are consistent with empirical measurements in the primary sensory cortex (*Cossell et al., 2015*; *Pala and Petersen, 2015*; *Campagnola et al., 2022*). The optimal ratio of mean I-I to E-I connectivity decreased when the error was weighted more with respect to the metabolic cost (*Figure 7—figure supplement 1D*).

Similarly to the ratio of E-I neuron numbers, a change in the ratio of mean E-I to I-E synaptic efficacy was compensated for by a change in firing rates, with stronger I-I synapses leading to a decrease in the firing rate of I neurons (*Figure 7H*, top). Conversely, weakening the E-I (and I-E) synapses resulted

in an increase in the firing rate in E neurons (*Figure 7—figure supplement 1E, F*). This is easily understood by considering that weakening the E-I and I-E synapses activates less strongly the lateral inhibition in E neurons (*Figure 3*) and thus leads to an increase in the firing rate of E neurons. We also found that single neuron variability remained almost unchanged when varying the ratio of mean I-I to E-I efficacy (*Figure 7H*, bottom) and the optimal ratio yielded optimal levels of average and instantaneous balance of synaptic inputs, as found previously (*Figure 7I*). The instantaneous balance monotonically decreased with increasing ratio of I-I to E-I efficacy (*Figure 7I*, bottom, *Figure 7—figure supplement 1G*).

Further, we tested the co-dependency of network optimality on the above two ratios with a 2-dimensional parameter search. We expected a positive correlation of network performance as a function of these two parameters, because both of them regulate the level of instantaneous E-I balance in the network. We found that the lower ratio of E-I neuron numbers indeed predicts a lower ratio of the mean I-I to E-I connectivity (*Figure 7J*). This is because fewer E neurons bring less excitation in the network, thus requiring less inhibition to achieve optimal levels of instantaneous balance. The co-dependency of the two parameters in affecting network optimality might be informative as to why E-I neuron number ratios may vary across species (for example, it is reported to be 2:1 in human cortex [*Fang et al., 2022*] and 4:1 in mouse cortex). Our model predicts that lower E-I neuron number ratios require weaker mean I-I to E-I connectivity.

In summary, our analysis suggests that optimal coding efficiency is achieved with more E neurons than I neurons and with mean I-I synaptic efficacy stronger than the E-I and I-E efficacy, and that these two parameters are positively correlated. Optimal ratios of E to I neurons and of connection strengths are broadly consistent with empirical measurements of these parameters in biological networks. The optimal network has less I than E neurons, but the impact of spikes of I neurons on the population readout is stronger, also suggesting that spikes of I neurons convey more information.

## Dependence of efficient coding and neural dynamics on the stimulus statistics

We further investigated how the network's behavior depends on the timescales of the input stimulus features. We manipulated the stimulus timescales by changing the time constants of $M = 3$ OU processes. The network efficiently encoded stimulus features when their time constants varied between 1 and 200 ms, with stable encoding error, metabolic cost (*Figure 8A*) and neural dynamics (*Figure 8—figure supplement 1A, B*). To examine if the network can efficiently encode also stimuli that evolve on different timescales, we tested its performance in response to $M = 3$ input variables, each with a different timescale. We kept the timescale of the first variable constant at $\tau_1^s = 10$ ms, while we varied the time constants of the other two keeping the time constant of the third twice as long as that of the second. We found excellent performance of the network in response to such stimuli that was stable across timescales (*Figure 8B*). The prediction that the network can encode information effectively over a wide range of time scales can be tested experimentally, by measuring the sensory information encoded by the activity of a set of neurons while varying the sensory stimulus timescales over a wide range.

We next examined network performance while varying the timescale of targets $\tau_x$ (see *Equation 1*). Because we assumed that the target time constants equal the membrane time constant of E and I neurons ($\tau_x = \tau^E = \tau^I = \tau$), it is not surprising that the best performance was achieved when these time constants were similar (*Figure 8—figure supplement 1C*). Firing rates, firing variability and the average and instantaneous balance did not change appreciably with this time constant (*Figure 8—figure supplement 1D–E*).

Next, we tested how the network's behavior changed when we varied the number of stimulus features $M$. Because all other parameters were optimized using $M = 3$, the encoding error of E (RMSE$^E$) and I neurons (RMSE$^I$) achieved a minimum around this value (*Figure 8C*, top). The metabolic cost increased monotonically with $M$ (*Figure 8C*, bottom). The number of features that optimized network efficiency (and minimized the average loss) depended on $g_L$, with stronger penalty of firing yielding a smaller optimal number of features. Increasing $M$ beyond the optimal number resulted in a gentle monotonic increase in firing rates for both E and I neurons, and it increased the average E-I balance and weakened the instantaneous balance (*Figure 8—figure supplement 1F, G*).

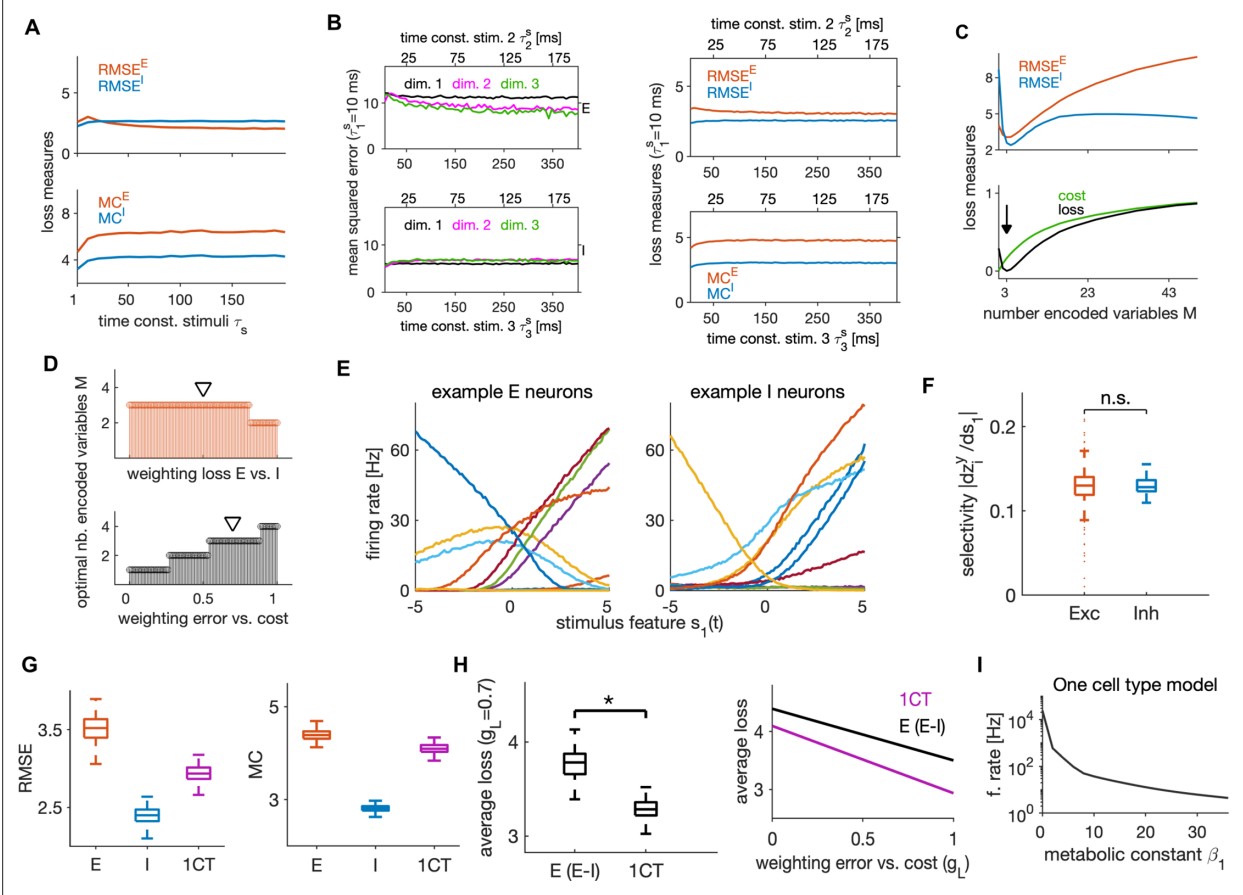

**Figure 8.** Dependence of efficient coding and neural dynamics on stimulus parameters and comparison of E-I versus one cell type model architecture. (**A**) Top: Root mean squared error (RMSE) of E estimates (red) and I estimates (blue), as a function of the time constant (in ms) of stimulus features. The time constant $\tau_s$ is the same for all stimulus features. Bottom: Same as on top, showing the metabolic cost (MC) of E and I cell type. (**B**) Left: Mean squared error between the targets and their estimates for every stimulus feature (marked as dimensions), as a function of time constants of OU stimuli in E population (top) and in I population (bottom). In the first dimension, the stimulus feature has a time constant fixed at 10 ms, while the second and third feature increase their time constants from left to right. The time constant of the third stimulus feature (x-axis on the bottom) is the double of the time constant of the second stimulus feature (x-axis on top). Right: Same as on the left, showing the RMSE that was averaged across stimulus features (top), and the metabolic cost (bottom) in E (red) and I (blue) populations. (**C**) Top: Same as in A top, measured as a function of the number of stimulus features $M$. Bottom: Normalized cost and the average loss as a function of the number of stimulus features. Black arrow marks the minimum loss and the optimal parameter $M$. (**D**) Top: Optimal number of encoded variables (stimulus features) as a function of weighting of the losses of E and I population. The weighting of the error with the cost is 0.7. Bottom: Same as on top, as a function of the weighting of the error with the cost and with equal weighting of losses of E and I populations. (**E**) Tuning curves of 10 example E (left) and I neurons (right). We computed tuning curves using = M3 stimulus features that were constant over time. We varied the amplitude of the first stimulus feature $s_1$, while two other stimulus features were kept fixed. (**F**) Distribution of the selectivity index across E (red) and I neurons (blue). (**G**) Root mean squared error (left) and metabolic cost (right) in E and I populations in the E-I model and in the 1CT model. The distribution is across 100 simulation trials. (**H**) Left: Average loss in the E population of the E-I model and of the 1CT model. The distribution is across 100 simulation trials. Right: Average loss in the E population of the E-I models and in the 1CT model as a function of the weighting $g_L$, averaged across trials. (**I**) Firing rate in the 1CT model as a function of the metabolic constant. All statistical results were computed on 100 simulation trials of 1 second duration. For other parameters of the E-I model see **Table 1**, and for the 1CT model see Appendix 2.

The online version of this article includes the following figure supplement(s) for figure 8:

**Figure supplement 1.** Effect of stimulus properties on efficient neural coding and dynamics.

We next characterized the tuning and the stimulus selectivity of E and I neurons. E neurons receive a feedforward current, which is expected to make them stimulus-selective, while I neurons receive synaptic inputs from E neurons through dense E-I connectivity. We measured stimulus tuning by computing tuning curves for each neuron in response to M=3 constant stimulus features (see Materials and methods). Similarly to previous work (**Barrett et al., 2016**), tuning curves of both E and I neurons were strongly heterogeneous (**Figure 8E**). We tested if the selectivity differs across E and I

cell types. We computed a selectivity index for each neuron as the stimulus-response gain (average change in the firing rate in response to a small change in the stimulus divided by the stimulus change size, see Materials and methods), and found that E and I neurons had similar mean stimulus selectivity ($p = 0.418$, two-tailed t-test; *Figure 8F*). Thus, I neurons, despite not receiving direct feedforward inputs and acquiring stimulus selectivity only through structured E-I connections, are tuned to the input stimuli as strongly as the E neurons.

## Comparison of E-I and one cell type model architecture for coding efficiency and robustness

Neurons in the brain are either excitatory or inhibitory. To understand how differentiating E and I neurons benefits efficient coding, we compared the properties of our efficient E-I network with an efficient network with a single cell type (1CT). The 1CT model can be seen as a simplification of the E-I model (see Appendix 1) and has been derived and analyzed in previous studies (*Bourdoukan et al., 2012*; *Boerlin et al., 2013*; *Koren and Denève, 2017*; *Gutierrez and Denève, 2019*; *Alemi et al., 2018*; *Brendel et al., 2020*). We compared the average encoding error (RMSE), the average metabolic cost (MC), and the average loss (see Appendix 3) of the E-I model against the one cell type (1CT) model. Compared to the 1CT model, the E-I model exhibited a higher encoding error and metabolic cost in the E population, but a lower encoding error and metabolic cost in the I population (*Figure 8G*). The 1CT model can perform similar computations as the E-I network. Instead of an E neuron directly providing lateral inhibition to its neighbor (*Figure 1—figure supplement 1A–C*), it goes through an interneuron in the E-I model (*Figure 1A(i) and B*). Because the E population of the E-I model and the 1CT model perform a similar computation, we compared the efficiency of the E population of the E-I model with the 1CT model. We found that the 1CT model is slightly more efficient than the E population of the E-I model, consistently for different weightings of the error with the cost (*Figure 8H*).

We further compared the robustness of firing rates to changes in the metabolic constant of the two models. Consistently with previous studies (*Koren and Denève, 2017*; *Rullán Buxó and Pillow, 2020*), firing rates in the 1CT model were highly sensitive to variations in the metabolic constant (*Figure 8I*, note the logarithmic scale on the y-axis), with a superexponential growth of the firing rate with the inverse of the metabolic constant in regimes with metabolic cost lower than optimal. This is in contrast to the E-I model, whose firing rates exhibited lower sensitivity to the metabolic constant, and never exceeded physiological limits (*Figure 7C*, top). Because our E-I model does not incorporate a saturating input-output function that constrains the range of firing as in *Kadmon et al., 2020*, the ability of the E-I model to maintain firing rates within biologically plausible limits emerges as a highly desirable dynamic property. One reason for higher stability of our E-I model compared to the 1CT model is that the delay of the lateral inhibition in the E-I model is twice that of the 1CT model (because in the E-I model, the lateral inhibition travels through an additional synapse). A second reason is that the recurrent connectivity of the 1CT model has exactly the same amount of average excitation and inhibition, while the E-I model is inhibition-dominated, which makes the E-I model more stable.

In summary, although the optimal E-I model is slightly less efficient than the optimal 1CT model, it does not enter into states of physiologically unrealistic firing rates when the metabolic constant is lower than the optimal one.

## Discussion

We analyzed the structural, dynamical and coding properties that emerge in networks of spiking neurons that implement efficient coding. We demonstrated that efficient recurrent E-I networks form highly accurate representations of stimulus features with biologically plausible parameters, biologically plausible neural dynamics, instantaneous E-I balance and like-to-like connectivity structure leading to lateral inhibition. The network can implement efficient coding with stimulus features varying over a wide range of timescales and when encoding even multiple such features. Here we discuss the implications of these findings.

By a systematic study of the model, we determined the model parameters that optimize network efficiency. The optimal parameters (including the ratio between the number of E and I neurons, the ratio of I-I to E-I synaptic efficacy and parameters of non-specific currents) were consistent with parameters

measured empirically in cortical circuits, and generated plausible spiking dynamics. This result lends credibility to the hypothesis that cortical networks might be designed for efficient coding and may operate close to optimal efficiency, as well as provides a solid intuition about what specific parameter ranges (e.g. higher numbers of E than I neurons) may be good for. With moderate deviations from the optimal parameters, efficient networks still exhibited realistic dynamics and close-to-efficient coding, suggesting that the optimal operational point of such networks is relatively robust. We also found that optimally efficient analytical solution derives generalized LIF (gLIF) equations for neuron models (*Koren and Panzeri, 2022*). While gLIF (*Schwalger and Lindner, 2013*; *Gerstner et al., 2014*) and LIF (*Brunel, 2000*; *Renart et al., 2010*) models are reasonably biologically plausible and are widely used to model and study spiking neural network dynamics, it was unclear how their parameters affect network-level information coding. Our study provides a principled way to determine uniquely the parameter values of gLIF networks that are optimal for efficient information encoding. Studying the dynamics of gLIF networks with such optimal parameters thus provides a direct link between optimal coding and neural dynamics. Moreover, our formalism provides a framework for the optimization of neural parameters that can in principle be used not only for neural network models that study brain function but also for the design of artificial neuromorphic circuits that perform information coding computations (*Roy et al., 2019*; *Schuman et al., 2022*).

Our model generates a number of insights about the role of structured connectivity in efficient information processing. A first insight is that I neurons develop stimulus feature selectivity because of the structured recurrent connectivity. While in visual cortex of naive animals, I neurons are typically reported to be less tuned than E neurons (*Hofer et al., 2011*; *Hu et al., 2014*) (but see *Runyan et al., 2010*), recent studies in association areas of well task trained animals consistently find I neurons as strongly tuned as E neurons (*Najafi et al., 2020*; *Kuan et al., 2024*). A second insight is that a network with structured connectivity shows stronger average and instantaneous E-I balance, as well as significantly lower variance of membrane potentials compared to an equivalent network with randomly organized connections. This implies that the connectivity structure is not only crucial for coding efficiency, but also influences the dynamical regime of the network. A third insight is that the structured network exhibits both lower encoding error and lower firing rates compared to unstructured networks, thus achieving higher efficiency. Our analysis of the effective connectivity created by the efficient connectivity structure shows that this structure sharpens stimulus representations, reduces redundancy and increases metabolic efficiency by implementing feature-specific competition, that is a negative effective connectivity between E neurons with similar stimulus tuning, as proposed by recent theories (*Moreno-Bote and Drugowitsch, 2015*) and experiments (*Chettih and Harvey, 2019*; *Oldenburg et al., 2024*) of computations in visual cortex.

Our model gives insights on what would be minimal requirements for a biological network to implement efficient coding. The network has to have structured E-I and I-E connectivity and weak and short-lasting or no spike-triggered adaptation. Further, at least half of the distance between the resting potential and the threshold should be provided by a stochastic external current that is unrelated to the feedforward stimuli. Finally, the network should have a ratio of E to I neuron numbers in the range of about 2:1 to 4:1 and the ratio of average I-I to E-I connectivity in the range of about 2:1 to 3:1, with smaller E-I neuron number ratios implying smaller average I-I to E-I connectivity ratios.

Our study gives insights into how structured connectivity between E and I neurons affects the dynamics of E-I balancing and how this relates to information coding. Previous work (*Deneve and Machens, 2016b*) proposed that the E-I balance in efficient spiking networks operates on a finer time scale than in classical balanced E-I networks with random connectivity (*Renart et al., 2010*). However, theoretical attempts to determine the levels of instantaneous E-I balance that are optimal for coding are rare (*Engelken and Goedeke, 2022*). Consistent with the general idea put forth in *Deneve and Machens, 2016b*; *Chalk et al., 2016*; *Denève et al., 2017*, we here showed that moderate levels of E-I balance are optimal for coding, and that too strong levels of instantaneous E-I balance are detrimental to coding efficiency. Our results predict that structured E-I-E connectivity is necessary for optimal levels of instantaneous E-I balance. Finally, the E-I-E structured connectivity that we derived supports optimal levels of instantaneous E-I balance and causes desynchronization of the spiking output. Such intrinsically generated desynchronization is a desirable network property that in previously proposed models could only be achieved by the less plausible addition of strong noise to each neuron (*Chalk et al., 2016*; *Rullán Buxó and Pillow, 2020*).

Our result that network efficiency depends gently on the number of neurons is consistent with previous findings that demonstrated robustness of efficient networks to neuronal loss (*Barrett et al., 2016*) and robustness of efficient spiking to the number of neurons (*Calaim et al., 2022*). Building on these studies, we additionally documented how the optimal ratio of the number of E to I neurons relates to the optimal ratio of average I-I to E-I connectivity. In particular, our analysis predicts that the optima of these two ratios are positively correlated. This might give insights into the diversity of ratios of E-I neuron number ratios observed across species (*Fang et al., 2022*).

We found that our efficient network, optimizing the representation of a leaky integration of stimulus features, does not require recurrent E-E connections. This is compatible with the relatively sparse levels of recurrent E-E connections in primary visual cortex (*Seeman et al., 2018*), with the majority of E-E synapses suggested to be long-range (*Stepanyants et al., 2009*). Nevertheless, a limitation of our study is that it did not investigate the computations that could be made by E-E connections. Future studies could address the role of recurrent excitatory synapses that implement efficient coding computations beyond leaky integration, such as linear (*Koren and Panzeri, 2022*) or non-linear mixing of stimulus features (*Alemi et al., 2018*). Investigating such networks would also allow addressing whether biologically plausible efficient networks exhibit criticality, as suggested by *Safavi et al., 2024*.

A more realistic mapping of efficient coding onto biological networks would also entail including multiple types of inhibitory neurons (*Wilson et al., 2012*), which could provide additional insights into how interneuron diversity serves information coding. Further limitations of our study to be addressed in future work include a more realistic implementation of the feedforward current. In our implementation, the feedforward current is simply a sum of uncorrelated stimulus features. However, in biological circuits, the feedforward input is a series of complex synaptic inputs from upstream circuits. A more detailed implementation of feedforward inputs, coupled with recurrent E-E synapses, might influence the levels of the instantaneous balance, in particular in E neurons, and have an impact on network efficiency. Moreover, we here did not explore cases where the same stimulus feature has multiple time scales. Finally, we note that efficient encoding might be the primary normative objective in sensory areas, while areas supporting high-level cognitive tasks might include other computational objectives, such as efficient transmission of information downstream to generate reliable behavioral outputs (*Valente et al., 2021*; *Panzeri et al., 2022*; *Manning et al., 2024*; *Koren et al., 2023*; *Blanco Malerba et al., 2024*). It would thus be important to understand how networks could simultaneously optimize or trade off different objectives.

## Materials and methods

### Overview of the current approach and of differences with previous approaches

In the following, we present a detailed derivation of the E-I spiking network implementing the efficient coding principle. The analytical derivation is based on previous works on efficient coding with spikes (*Boerlin et al., 2013*; *Koren and Denève, 2017*), and in particular on our recent work (*Koren and Panzeri, 2022*). While these previous works analytically derived feedforward and recurrent transmembrane currents in leaky integrate-and fire neuron models, they did not contain any synaptic currents unrelated to feedforward and recurrent processing. Non-specific synaptic currents were suggested to be important for an accurate description of coding and dynamics in cortical networks (*Destexhe et al., 2003*). In the model derivation that follows, we also derived non-specific external current from efficiency objectives.

Moreover, we here revisited the derivation of physical units in efficient spiking networks. We built on a previous work *Boerlin et al., 2013* that assigned physical units to mathematical expressions that correspond to membrane potentials, firing thresholds, etc. Here, we instead assigned physical units to the computational variables such as the target signals and the population readouts, and then derived units of the membrane potentials and firing thresholds.

With this model, we aim to describe neural dynamics and computation in early sensory cortices such as the primary visual cortex in rodents, even though many principles of the model developed here could be relevant throughout the brain.

## Introducing variables of the model

We consider two types of neurons, excitatory neurons $E$ and inhibitory neurons $I$. We denote as $N^E$ and $N^I$ the number of $E$-cells and $I$-cells, respectively. The spike train of neuron $i$ of type $y \in \{E, I\}$, $i = 1, 2, ..., N^y$, is defined as a sum of Dirac delta functions,

$$f_i^y(t) = \sum_\alpha \delta(t - t_i^{y, \alpha}),$$ (6)

where $t_i^{y, \alpha}$ is the time of the $\alpha$-th spike of that neuron, defined as a time point at which the membrane potential of neuron $i$ crosses the firing threshold.

We define the readout of the spiking activity of neuron $i$ of type $y$ (in the following, 'ingle neuron readout') as a leaky integration of its spike train,

$$\dot{r}_i^y(t) = -\lambda_r^y r_i^y(t) + f_i^y(t), \qquad y \in \{E, I\},$$ (7)

with $\lambda_r^y$ denoting the inverse time constant. This way, the quantity $z_i^y(t) = \lambda_r^y r_i^y(t)$ represents an estimate of the instantaneous firing rate of neuron $i$.

We denote as $\vec{s}(t) := [s_1(t), \dots, s_M(t)]^\top$ the set of $M$ dynamical features of the external stimulus (in the following, stimulus features) which are transmitted to the network through a feedforward sensory pathway. The stimulus features have the unit of the square root of millivolt, $(mV)^{1/2}$. The target signal is then obtained through a leaky integration of the feedforward variable, $\vec{s}(t)$ (**Bourdoukan et al., 2012**), with inverse time constant $\lambda$, as

$$\dot{\vec{x}}(t) = -\lambda \vec{x}(t) + \vec{s}(t),$$ (8)

with $\vec{x}(t) := [x_1(t), \dots, x_M(t)]^\top$ the vector of $M$ target signals. Furthermore, we define a linear population readout of the spiking activity of E and I neurons

$$\dot{\vec{\hat{x}}}^y(t) = -\lambda \vec{\hat{x}}^y(t) + \sum_{i=1}^{N^y} \vec{w}_i^y f_i^y(t), \qquad y \in \{E, I\},$$ (9)

with $\vec{\hat{x}}^y(t) := [\hat{x}_1^y(t), \dots, \hat{x}_M^y(t)]^\top$ the vector of estimates of cell type $y$ and $\vec{w}_i^y$ in units of $(mV)^{1/2}$. Here, each neuron $i$ of type $y$ is associated with a vector $\vec{w}_i^y := [w_{1i}^y, \dots, w_{Mi}^y]^\top$ of $M$ tuning parameters representing the decoding weight of neuron $i$ with respect to the $M$ population readouts in **Equation 9**. These decoding vectors can be combined in the $M \times N^y$ matrix $W^y = [w_{ki}^y]$. The rows of this matrix define the patterns of decoding weights $\vec{w}_k^y := [w_{k1}^y, \dots, w_{kN^y}^y]^\top$ for each signal dimension $k = 1, \dots, M$.

## Loss functions

We assume that the activity of a population $y \in \{E, I\}$ is set so as to minimize a time-dependent encoding error and a time-dependent metabolic cost:

$$L^y(t) = \epsilon^y(t) + \beta^y \kappa^y(t),$$ (10)

with $\beta^y > 0$ in units of mV the Lagrange multiplier which controls the weight of the metabolic cost relative to the encoding error. The time-dependent encoding error is defined as the squared distance between the targets and their estimates, and the role of estimates is assigned to the population readouts $\vec{\hat{x}}^y(t)$. In E neurons, the targets are defined as the target signals $\vec{x}(t)$, and their estimators are the population readouts of the spiking activity of E neurons, $\vec{\hat{x}}^E(t)$. In I neurons, the targets are defined as the population readouts of E neurons $\vec{\hat{x}}^E(t)$ and their estimators are the population readouts of I neurons $\vec{\hat{x}}^I(t)$. Furthermore, the time-dependent metabolic cost is proportional to the squared estimate of the instantaneous firing rate, summed across neurons from the same population. Following these assumptions, we define the variables of loss functions in **Equation 10** as

$$\varepsilon^E(t) = \left\| \vec{x}(t) - \vec{\tilde{x}}^E(t) \right\|^2, \qquad \kappa^E(t) = \sum_{i=1}^{N^E} \left[ r_i^E(t) \right]^2,$$

$$\varepsilon^I(t) = \left\| \vec{\tilde{x}}^E(t) - \vec{\tilde{x}}^I(t) \right\|^2, \qquad \kappa^I(t) = \sum_{i=1}^{N^I} \left[ r_i^I(t) \right]^2. \tag{11}$$

We use a quadratic metabolic cost because it promotes the distribution of spiking across neurons (**Boerlin et al., 2013**). In particular, the loss function of I neurons, $L^I(t)$ implies the relevance of the approximation: $\vec{\tilde{x}}^E(t) \approx \vec{\tilde{x}}^I(t)$ (see $\epsilon^I$ in the **Equation 11**), which will be used in what follows.

## When shall a neuron spike?

We minimize the loss function by positing that neuron $i$ of type $y \in \{E, I\}$ emits a spike as soon as its spike decreases the loss function of its population $y$ in the immediate future (**Koren and Panzeri, 2022**). We also define $t^-$ and $t^+$ as the left- and right-sided limits of a spike time $t = t_i^{y,\alpha}$, respectively. Thus, at the spike time, the following jump condition must hold:

$$L^y(t^+) \leq L^y(t^-) + \xi_i^y(t^-), \qquad y \in \{E, I\}, \tag{12}$$

with $\xi_i^y$ in units of mV. Here, the arguments $t^-$ and $t^+$ denote the left- and right-sided limits of the respected functions at time $t$. Furthermore, we added a noise term on the right-hand side of the **Equation 12** in order to consider the stochastic nature of spike generation in biological networks (**Faisal et al., 2008**). A convenient choice for the noise $\xi_i^y(t)$ is the Ornstein-Uhlenbeck process obeying

$$\dot{\xi}_i^y(t) = -\lambda \xi_i^y(t) + \sqrt{2\lambda} \sigma_\xi^y \eta_i^y(t), \tag{13}$$

where $\eta_i^y$ is a Gaussian white noise with auto-covariance function $\langle \eta_i(t)\eta_j(t') \rangle = \delta_{ij}\delta(t - t')$. The process $\xi_i^y(t)$ has zero mean and auto-covariance function $\langle \xi_i(t)\xi_j(t') \rangle = (\sigma_\xi^y)^2 \delta_{ij} e^{-\lambda |t - t'|}$, with $(\sigma_\xi^y)^2$ the variance of the noise.

By applying the condition for spiking in **Equation 12** using $y = E$ and $y = I$, respectively, we get

$$\left[ \left\| \vec{x}(t^+) - \vec{\tilde{x}}^E(t^+) \right\|^2 + \beta^E \sum_{j=1}^{N^E} \left( r_j^E(t^+) \right)^2 \right] - \left[ \left\| \vec{x}(t^-) - \vec{\tilde{x}}^E(t^-) \right\|^2 + \beta^E \sum_{j=1}^{N^E} \left( r_j^E(t^-) \right)^2 \right] \leq \xi_i^E(t^-),$$

$$\left[ \left\| \vec{\tilde{x}}^E(t^+) - \vec{\tilde{x}}^I(t^+) \right\|^2 + \beta^I \sum_{j=1}^{N^I} \left( r_j^I(t^+) \right)^2 \right] - \left[ \left\| \vec{\tilde{x}}^E(t^-) - \vec{\tilde{x}}^I(t^-) \right\|^2 + \beta^I \sum_{j=1}^{N^I} \left( r_j^I(t^-) \right)^2 \right] \leq \xi_i^I(t^-). \tag{14}$$

According to the definitions in **Equation 7** and **Equation 9**, if neuron $i$ fires a spike at time $t = t_i^{y,\alpha}$, it causes a jump of its own filtered spike train (but not of other neurons $j \neq i$), as well as of the population readout of the population it belongs to. Therefore, when neuron $i$ fires a spike, we have for a given neuron $j$ and a given population readout $k$:

$$r_j^y(t^+) = r_j^y(t^-) + \delta_{ij}, \tag{15a}$$

$$\hat{x}_k^y(t^+) = \hat{x}_k^y(t^-) + w_{ki}^y. \tag{15b}$$

By inserting **Equation 15a**, **Equation 15b** in **Equation 12**, we find that neuron $i$ of type $y$ should fire a spike if the following condition holds:

$$(\vec{w}_i^E)^\top \left( \vec{x}(t) - \vec{\tilde{x}}^E(t) \right) - \beta^E r_i^E(t) \geq \frac{1}{2} \left( \left\| \vec{w}_i^E \right\|^2 + \beta^E - \xi_i^E(t) \right),$$

$$(\vec{w}_i^I)^\top \left( \vec{\tilde{x}}^E(t) - \vec{\tilde{x}}^I(t) \right) - \beta^I r_i^I(t) \geq \frac{1}{2} \left( \left\| \vec{w}_i^I \right\|^2 + \beta^I - \xi_i^I(t) \right) \tag{16a}$$

with $\|\vec{w}_i^y\|^2 := \sum_{k=1}^M (w_{ki}^y)^2$ the squared length of the tuning vector of neuron $i$ of type $y$. These equations tell us when the neuron $i$ of type $E$ and $I$, respectively, emits a spike, and are similar to the ones derived in previous works (**Koren and Panzeri, 2022**; **Boerlin et al., 2013**). In addition to what has

been found in these previous works, we here also find that each term on the left- and right-hand side in the *Equation 16a* has the physical units of millivolts.

We note that the expression derived from the minimization of the loss function of E neurons in the top row of *Equation 16a* is independent of the activity of I neurons, and would thus lead to the E population being unconnected with the I population. In order to derive a recurrently connected E-I network, the activity of E neurons must depend on the activity of I neurons. We impose this property by using the approximation of estimates that holds under the assumption of efficient coding in I neurons (see $\epsilon^I$ in the *Equation 11*), $\vec{\tilde{x}}^I(t) \approx \vec{\tilde{x}}^E(t)$. This yields the following conditions:

$$
\begin{aligned}
(\vec{w}_i^E)^\top \left( \vec{x}(t) - \vec{\tilde{x}}^E(t) \right) - \beta^E r_i^E(t) &\geq \frac{1}{2} \left( \left\| \vec{w}_i^E \right\|^2 + \beta^E - \xi_i^E(t) \right), \\
(\vec{w}_i^I)^\top \left( \vec{\tilde{x}}^E(t) - \vec{\tilde{x}}^I(t) \right) - \beta^I r_i^I(t) &\geq \frac{1}{2} \left( \left\| \vec{w}_i^I \right\|^2 + \beta^I - \xi_i^I(t) \right).
\end{aligned}
\tag{16b}
$$

We now define new variables $u_i^y(t)$ and $\theta_i^y$ as proportional to the left- and the right-hand side of these expressions,

$$
\begin{aligned}
u_i^E(t) &:= (\vec{w}_i^E)^\top \left( \vec{x}(t) - \vec{\tilde{x}}^I(t) \right) - \beta^E r_i^E(t), \\
u_i^I(t) &:= (\vec{w}_i^E)^\top \left( \vec{\tilde{x}}^E(t) - \vec{\tilde{x}}^I(t) \right) - \beta^I r_i^I(t), \\
\theta_i^y &:= \frac{1}{2} \left( \left\| \vec{w}_i^y \right\|^2 + \beta^y \right) - \frac{1}{2} \xi_i^y(t), \quad y \in \{E, I\}.
\end{aligned}
\tag{17}
$$

The variables $u_i^y(t)$ and $\theta_i^y$ are interpreted as the membrane potential and the firing threshold of neuron $i$ of cell type $y \in \{E, I\}$.

## Dynamic equations for the membrane potentials

In this section, we develop the exact dynamic equations of the membrane potentials $\dot{u}_i^y(t)$ for $y \in \{E, I\}$ according to the efficient coding assumption. We rewrite *Equation 9* in vector notation as

$$
\begin{aligned}
\dot{\vec{\tilde{x}}}^E(t) &= -\lambda \vec{\tilde{x}}^E(t) + W^E \vec{f}^E(t), \\
\dot{\vec{\tilde{x}}}^I(t) &= -\lambda \vec{\tilde{x}}^I(t) + W^I \vec{f}^I(t),
\end{aligned}
\tag{18}
$$

with $\vec{f}^y(t) := [f_1^y(t), \dots, f_{N^y}^y(t)]^\top$ the vector of spike trains for $N^y$ neurons of cell type $y \in \{E, I\}$.

In the case of E neurons, the time-derivative of the membrane potential $\dot{u}_i^E(t)$ in *Equation 17*, is obtained as

$$
\dot{u}_i^E(t) = (\vec{w}_i^E)^\top \left( \dot{\vec{x}} - \dot{\vec{\tilde{x}}}^I(t) \right) - \beta^E \dot{r}_i^E(t).
\tag{19}
$$

By inserting the dynamic equations of the target signal $\vec{x}(t)$, its estimate $\vec{\tilde{x}}^I(t)$ (*Equation 18*) and of the single neuron readout $\dot{r}_i^E(t)$ (*Equation 7* in the case $y = E$), we get

$$
\begin{aligned}
\dot{u}_i^E(t) =& (\vec{w}_i^E)^\top \left[ -\lambda \vec{x}(t) + \vec{s}(t) + \lambda \vec{\tilde{x}}^I(t) - W^I \vec{f}^I \right] - \beta^E \left[ -\lambda_r^E r_i^E(t) + f_i^E(t) \right], \\
=& -\lambda \left[ (\vec{w}_i^E)^\top \left( \vec{x}(t) - \vec{\tilde{x}}^I(t) \right) - \beta^E r_i^E(t) \right] + (\vec{w}_i^E)^\top \vec{s}(t) - (\vec{w}_i^E)^\top W^I \vec{f}^I \\
& - \beta^E (\lambda - \lambda_r^E) r_i^E(t) - \beta^E f_i^E(t), \\
=& -\lambda u_i^E(t) + (\vec{w}_i^E)^\top \vec{s}(t) - (\vec{w}_i^E)^\top W^I \vec{f}^I - \beta^E (\lambda - \lambda_r^E) r_i^E(t) - \beta^E f_i^E(t),
\end{aligned}
\tag{20}
$$

where in the last line we used the definition of $u_i^E(t)$ from the *Equation 17*.

In the case of I neurons, the time derivative of the membrane potential $\dot{u}_i^I(t)$ in *Equation 17* is

$$
\dot{u}_i^E(t) = (\vec{w}_i^E)^\top \left( \dot{\vec{\tilde{x}}}^E(t) - \dot{\vec{\tilde{x}}}^I(t) \right) - \beta^I \dot{r}_i^I(t).
\tag{21}
$$

By inserting the dynamic equations of the population readouts of E neurons $\vec{\hat{x}}^E(t)$ and of the I neurons $\vec{\hat{x}}^I(t)$ (**Equation 18**) and of the single neuron readout $r_i^I(t)$ (**Equation 7** in the case $y = I$), we get

$$
\begin{aligned}
\dot{u}_i^I(t) =& (\vec{w}_i^I)^\top \left[ -\lambda \vec{\hat{x}}^E(t) + W^E \vec{f}^E(t) + \lambda \vec{\hat{x}}^I(t) - W^I \vec{f}^I(t) \right] - \beta^I \left[ -\lambda_r^I r_i^I(t) + f_i^I(t) \right], \\
=& -\lambda \left[ (\vec{w}_i^I)^\top \left( \vec{\hat{x}}^E(t) - \vec{\hat{x}}^I(t) \right) - \beta^I r_i^I(t) \right] + (\vec{w}_i^I)^\top W^E \vec{f}^E(t) - (\vec{w}_i^I)^\top W^I \vec{f}^I(t) \\
& - \beta^I (\lambda - \lambda_r^I) r^I I_i(t) - \beta^I f_i^{EI}(t), \\
=& -\lambda u_i^I(t) + (\vec{w}_i^I)^\top W^E \vec{f}^E(t) - (\vec{w}_i^I)^\top W^I \vec{f}^I(t) - \beta^I (\lambda - \lambda_r^I) r_i^I(t) - \beta^I f_i^I(t).
\end{aligned}
\tag{22}
$$

where in the last line we used the definition of $u_i^I(t)$ from **Equation 17**.

## Leaky integrate-and-fire neurons

The terms on the right-hand-side in **Equation 20** and **Equation 22** can be interpreted as transmembrane currents. The last term in these equations, $-\beta^y f_i^y(t), y \in \{E, I\}$, can be interpreted as a current instantaneously resetting the membrane potential upon reaching the firing threshold (**Boerlin et al., 2013**). Indeed, when the membrane potential reaches the threshold, it triggers a spike and causes a jump of the membrane potential by an amount $-\beta^y$; this realizes resetting of the membrane potential which is equivalent to the resetting rule of integrate-and-fire neurons (**Burkitt, 2006**; **Harkin et al., 2021**). Thus, by taking into account the resetting mechanism and defining the time constants of population and single neuron readout $\tau := \lambda^{-1}$ and $\tau_r^y := (\lambda_r^y)^{-1}$, we can rewrite **Equation 20** and **Equation 22** as a leaky integrate-and-fire neuron model,

$$
\begin{aligned}
\dot{u}_i^E(t) =& -\frac{1}{\tau} u_i^E(t) + (\vec{w}_i^E)^\top \vec{s}(t) - \sum_{j=1}^{N^I} (\vec{w}_i^E)^\top \vec{w}_j^I f_j^I(t) - \beta^E \left( \frac{1}{\tau} - \frac{1}{\tau_r^E} \right) r_i^E(t), \\
\dot{u}_i^E(t) =& -\frac{1}{\tau} u_i^I(t) + \sum_{j=1}^{N^E} (\vec{w}_i^I)^\top \vec{w}_j^E f_j^E(t) - \sum_{\substack{j=1 \\ i \neq j}}^{N^I} (\vec{w}_i^I)^\top \vec{w}_j^I f_j^I(t) - \beta^I \left( \frac{1}{\tau} - \frac{1}{\tau_r^I} \right) r_i^I(t), \\
& \text{if } u_i^y(t^-) \geq \theta_i^y \to u_i^y(t^+) = u_i^{y,\text{reset}}, \\
& \theta_i^y = \frac{1}{2} \left( \| \vec{w}_i^y \|^2 + \beta^y - \xi_i^y(t) \right), \\
& u_i^{E,\text{reset}} = \theta_i^E - \beta^E, \\
& u_i^{I,\text{reset}} = \theta_i^I - \beta^I - \left\| \vec{w}_i^I \right\|^2.
\end{aligned}
\tag{23}
$$

In the **Equation 23**, we wrote explicitly the terms $(\vec{w}_i^y)^\top W^x \vec{f}^x(t) = \sum_{j=1}^{N^x} (\vec{w}_i^y)^\top \vec{w}_j^x f_j^x(t)$, which correspond to the synaptic projections of $N^x$ presynaptic neurons of type $x$ to the postsynaptic neuron $i$ of type $y$, with the quantity $(\vec{w}_i^y)^\top \vec{w}_j^x$ denoting the synaptic weight. We note that, in the case of I neurons, the element with $j = i$ describes an autapse, that is a projection of a neuron with itself; this term is equal to $- \left( \vec{w}_i^I \right)^\top \vec{w}_i^I f_i^I(t) = -\| \vec{w}_i^I \|^2 f_i^I(t)$, and thus contributes to the resetting of the neuron $i$.

## Imposing Dale's principle on synaptic connectivity

We now examine the synaptic terms in **Equation 23**. As a first remark, we see that synaptic weights depend on tuning parameters $w_{ki}^y$. For the sake of generality we drew tuning parameters $w_{ki}^y$ from a normal distribution with vanishing mean, which yielded both positive and negative values of $w_{ki}^y$. This has the desirable consequence that a spike of a neuron with a positive tuning parameter in signal dimension $k$, $w_{ki}^y > 0$ pulls the estimate, $\hat{x}_k^y(t)$, up, while a spike of a neuron with $w_{ki}^y < 0$ pulls the estimate down, allowing population readouts to track both positive and negative fluctuations of the target signal on a fast time scale.

Another consequence of synaptic connectivity in the **Equation 23** is that the synaptic weight between a presynaptic neuron $j$ of type $x$ and a postsynaptic neuron $i$ of type $y$ is symmetric and depends on

the similarity of tuning vectors of the presynaptic and the postsynaptic neuron: $(\vec{w}_i^y)^\top \vec{w}_j^x = \sum_{k=1}^M w_{ki}^y w_{kj}^x$. The sign of this scalar product is positive between neurons with similar tuning and negative between neurons with different tuning (and zero when the two tuning vectors are orthogonal). Thus, for a presynaptic neuron $j$ of type $x$, the synaptic weights of its outgoing connections can be both positive and negative, because some of its postsynaptic neurons have similar tuning to the neuron $j$ while others have different tuning. This is inconsistent with Dale's principle (**Whittaker, 1983**), which postulates that a particular neuron can only have one type of effect on postsynaptic neurons (excitatory or inhibitory), but never both. To impose this constraint in our model, we set synaptic weights between neurons with different tuning (i.e. $(\vec{w}_i^y)^\top \vec{w}_j^x < 0$) to zero. To this end, we define the rectified connectivity matrices,

$$J_{ij}^{yx} = \left[ (\vec{w}_i^y)^\top \vec{w}_j^x \right]_+,$$ 
(24)

with $(x,y) \in \{(E,I),(I,I),(I,E)\}$ and $[a]_+ \equiv \max(0,a)$ a rectified linear function. Note that there are no direct synaptic connections between E neurons. Since the elements of the matrix $J^{yx}$ are all non-negative, it is the sign in front of the synaptic term in the **Equation 23** that determines the sign of the synaptic current between neurons $i$ and $j$. The synaptic current is excitatory if the sign is positive, and inhibitory if the sign is negative.

It is also interesting to note that rectification affects the rank of connectivity matrices. Without rectification, the product in **Equation 24** yields a connectivity matrix with rank smaller or equal to the number of input features to the network, $M$, similarly as in previous works (**Bourdoukan et al., 2012**; **Barrett et al., 2016**; **Alemi et al., 2018**). Since typically the number of input features is much smaller than the number of neurons, that is $M << N^y$, this would give a low-rank connectivity matrix. However, rectification in **Equation 24**, necessary to ensure Dale's principle in presence of positive and negative tuning parameters, typically results in a substantial increase of the rank of the connectivity matrix.

Using the synaptic connectivity defined in **Equation 24**, we rewrite the network dynamics from **Equation 23** as:

$$\dot{u}_i^E(t) = -\frac{1}{\tau} u_i^E(t) + (\vec{w}_i^E)^\top \vec{s}(t) - \sum_{j=1}^{N^I} J_{ij}^{EI} f_j^I(t) - \beta^E \left( \frac{1}{\tau} - \frac{1}{\tau_r^E} \right) r_i^E(t),$$

$$\dot{u}_i^E(t) = -\frac{1}{\tau} u_i^I(t) + \sum_{j=1}^{N^E} J_{ij}^{IE} f_j^E(t) - \sum_{\substack{j=1 \\ i \neq j}}^{N^I} J_{ij}^{II} f_j^I(t) - \beta^I \left( \frac{1}{\tau} - \frac{1}{\tau_r^I} \right) r_i^I(t),$$

$$\text{if } u_i^y(t^-) \geq \theta_i^y \rightarrow u_i^y(t^+) = u_i^{y,\text{reset}},$$
(25)

$$\theta_i^y = \frac{1}{2} \left( \| \vec{w}_i^y \|^2 + \beta^y - \xi_i^y(t) \right),$$

$$u_i^{E,\text{reset}} = \theta_i^E - \beta^E,$$

$$u_i^{I,\text{reset}} = \theta_i^I - \beta^I - \left\| \vec{w}_i^I \right\|^2.$$

These equations express the neural dynamics which minimizes the loss functions (**Equation 10**) in terms of a generalized leaky integrate-and-fire model with E and I cell types, and are consistent with Dale's principle.

In principle, it is possible to use the same strategy as for the E-I network to enforce Dale's principle in model with one cell type (introduced by **Boerlin et al., 2013**). To do so, we constrained the recurrent connectivity of the model with a single cell type from **Koren and Denève, 2017** by keeping only connections between neurons with similar tuning vectors and setting other connections to 0 (see Appendix 1). This led to a network of only inhibitory neurons, a type of network model which is less relevant for the description of biological networks.

## Model with resting potential and an external current

In the model given by the **Equation 25** the resting potential is equal to zero. In order to account for biophysical values of the resting potential and to introduce an implementation of the metabolic constant that is consistent with neurobiology, we add a constant value to the dynamical equations

of the membrane potentials $\dot{u}_i^y$, the firing thresholds $\theta_i^y$ and the reset potentials $u_i^{y,\text{reset}}$. This does not change the spiking dynamics of the model, as what matters to correctly infer the efficient spiking times of neurons is the distance between the membrane potential and the threshold.

Furthermore, in the same equations, the role of the metabolic constant $\beta^y$ as a biophysical quantity is questionable. The metabolic constant $\beta^y$ is an important parameter that weights the metabolic cost over the encoding error in the objective functions (*Equation 10*). On the level of computational objectives, the metabolic constant naturally controls firing rates, as it allows the network to fire more or less spikes to correct for a certain encoding error. A flexible control of the firing rates is a desirable property, as gives the possibility to potentially capture different dynamical regimes of efficient spiking networks (*Koren and Denève, 2017*). In the spiking model we developed thus far (*Equation 25*), similarly to previous efficient spiking models (*Koren and Denève, 2017*; *Gutierrez and Denève, 2019*), the metabolic constant $\beta^y$ controls the firing threshold. In neurobiology, however, strong changes to the firing threshold that would reflect metabolic constraints of the network are not plausible. We thus searched for an implementation of the metabolic constant $\beta^y$ that is consistent with neurobiology.

The condition for threshold crossing of the neuron $i$ can be written by *Equation 25* as

$$u_i^y(t) + V_{\text{rest}}^y + \frac{1}{2}\left(c - \beta^y + \xi_i^y(t)\right) \geq \frac{1}{2}\left(\left\|\vec{w}_i^y\right\|^2 + c\right) + V_{\text{rest}}^y, \tag{26}$$

with $c$ an arbitrary constant in units of millivolts. In *Equation 26* we added a constant $c/2$ and a resting potential $V_{\text{rest}}^y$ on the left- and right-hand side of the firing rule. Moreover, we shifted the noise and the dependency on the parameter $\beta$ from the firing threshold to the membrane potential. Thus, we assumed that the firing threshold is independent of the metabolic constant and the noise, and we instead assumed the dependence on the metabolic constant and noise in the membrane potentials.

We now define new variables for $y \in \{E, I\}$:

$$\begin{aligned} V_i^y(t) &\equiv u_i^y(t) + V_{\text{rest}}^y + \frac{1}{2}\left(c - \beta^y + \xi_i^y(t)\right), \qquad V_{\text{rest}}^y < 0, \\ \vartheta_i^y &\equiv V_{\text{rest}}^y + \frac{1}{2}\left(\left\|\vec{w}_i^y\right\|^2 + c\right), \end{aligned} \tag{27}$$

and rewrite the model in *Equation 25* in these new variables

$$\begin{aligned} \tau\dot{V}_i^E(t) = &-\left(V_i^E(t) - V_{\text{rest}}^E\right) + \tau\left(\vec{w}_i^E\right)^\top \vec{s}(t) - \tau\sum_{j=1}^{N^I} J_{ij}^{EI} f_j^I(t) - \beta^E(1 - \frac{\tau}{\tau_r^E})r_i^E(t) \\ &+ \frac{1}{2}\left(c - \beta^E\right) + \sqrt{\frac{\tau}{2}}\sigma_\xi^E \eta_i^E(t), \\ \tau\dot{V}_i^I(t) = &-\left(V_i^I(t) - V_{\text{rest}}^I\right) + \tau\sum_{j=1}^{N^E} J_{ij}^{IE} f_j^E(t) - \tau\sum_{\substack{j=1 \\ i\neq j}}^{N^I} J_{ij}^{II} f_j^I(t) - \beta^I(1 - \frac{\tau}{\tau_r^I})r_i^I(t) \\ &+ \frac{1}{2}\left(c - \beta^I\right) + \sqrt{\frac{\tau}{2}}\sigma_\xi^I \eta_i^I(t), \\ &\text{if } V_i^y(t^-) \geq \vartheta_i^y \rightarrow V_i^y(t^+) = V_i^{y,\text{reset}}, \\ &\vartheta_i^y = V_{\text{rest}}^y + \frac{1}{2}\left(\left\|\vec{w}_i^y\right\|^2 + c\right), \\ &V_i^{E,\text{reset}} = V_{\text{rest}}^E - \beta^E + \frac{1}{2}\left(c + \left\|\vec{w}_i^E\right\|^2\right), \\ &V_i^{I,\text{reset}} = V_{\text{rest}}^I - \beta^I + \frac{1}{2}\left(c - \left\|\vec{w}_i^I\right\|^2\right), \end{aligned} \tag{28}$$

where $\eta_i^E(t)$ and $\eta_i^I(t)$ are the independent Gaussian white noise processes defined in *Equation 13* above. We note that all terms on the right-hand side of *Equation 28* have the desired units of mV. The model in *Equation 28* is an efficient E-I spiking network with improved compatibility with neurobiology. We have expressed two new terms in the membrane potentials of E and I neurons, one dependent on the metabolic constant $\beta^y$ and one on the noise that we assumed in the condition for spiking

(see *Equation 12*). We will group these two terms to define an external current, a current that is well known in spiking models of neural dynamics (*Gerstner et al., 2014*).

## Efficient generalized leaky integrate-and-fire neuron model

Finally, we rewrite the model from *Equation 28* in a compact form in terms of transmembrane currents, and discuss their biological interpretation. The efficient coding with spikes is realized by the following model for the neuron $i$ of type $y \in \{E, I\}$:

$$\tau \dot{V}_i^E(t) = -\left(V_i^E(t) - V_{\text{rest}}^E\right) + R_m \left(I_i^{\text{syn},E}(t) - I_i^{\text{ad},E}(t) + I_i^{\text{ext},E}(t)\right),$$

$$\tau \dot{V}_i^I(t) = -\left(V_i^I(t) - V_{\text{rest}}^I\right) + R_m \left(I_i^{\text{syn},I}(t) - I_i^{\text{ad},I}(t) + I_i^{\text{ext},I}(t)\right),$$

$$\text{if } V_i^y(t^-) \geq \vartheta_i^y \to V_i^y(t^+) = V_i^{y,\text{reset}},$$

$$\vartheta_i^y = V_{\text{rest}}^y + \frac{1}{2}\left(\|\vec{w}_i^y\|^2 + c\right), \tag{29a}$$

$$V_i^{E,\text{reset}} = V_{\text{rest}}^E - \beta^E + \frac{1}{2}\left(c + \|\vec{w}_i^E\|^2\right),$$

$$V_i^{I,\text{reset}} = V_{\text{rest}}^I - \beta^I + \frac{1}{2}\left(c - \|\vec{w}_i^I\|^2\right),$$

with $R_m$ the current resistance. The leak current,

$$I_i^{\text{leak},y}(t) = -\frac{C_m}{\tau}\left(V_i^y(t) - V_{\text{rest}}^y\right), \qquad y \in \{E, I\}, \tag{29b}$$

with $\tau = R_m C_m$ and $C_m$ the capacitance of the neural membrane (*Burkitt, 2006*), arose by assuming the same time constant for the target signals $\vec{x}(t)$ and estimates $\hat{\vec{x}}^E(t)$ and $\hat{\vec{x}}^I(t)$ (see *Equations 8 and 18*). We see that the passive membrane time constant $\tau = \lambda^{-1}$ can be traced back to the time constant of the population read-out in *Equation 9*. The synaptic currents are defined as

$$I_i^{\text{syn},E}(t) = C_m \left((\vec{w}_i^E)^\top \vec{s}(t) - \sum_{j=1}^{N^I} J_{ij}^{EI} f_j^I(t)\right),$$

$$I_i^{\text{syn},I}(t) = C_m \left(\sum_{j=1}^{N^E} J_{ij}^{IE} f_j^E(t) - \sum_{\substack{j=1 \\ i \neq j}}^{N^I} J_{ij}^{II} f_j^I(t)\right), \tag{29c}$$

where we note the presence of a feedforward current to E neurons,

$$I_i^{\text{ff}}(t) = C_m (\vec{w}_i^E)^\top \vec{s}(t),$$

$$= C_m \sum_{k=1}^{M} w_{ki}^E s_k(t), \tag{29d}$$

which consist in a linear combination of the stimulus features $\vec{s}(t)$ weighted by the decoding weights $\vec{w}_i^E$. The stimulus features can be traced back to the definition of the target signals in *Equation 8*. This current emerges in E neurons, as a consequence of having the target signal $\vec{x}(t)$ in the loss function of the E population (see *Equations 10 and 11*). I neurons do not receive the feedforward current because their loss function does not contain the target signal.

The current providing within-neuron feedback triggered by each spike,

$$I_i^{\text{ad},y}(t) = C_m \beta^y \left(\frac{1}{\tau} - \frac{1}{\tau_r^y}\right) r_i^y(t), \tag{29e}$$

was recently recovered (*Koren and Panzeri, 2022*). This current has the kinetics of the single neuron readout $r_i^y(t)$ (i.e. low-pass filtered spike train). Its sign depends on the relation between the time constant of the population readout $\tau = \lambda^{-1}$ and single neuron readout $\tau_r^y = (\lambda_r^y)^{-1}$, because the metabolic constant $\beta^y$ is non-negative by definition (*Equation 10*). If the single neuron readout is slower than the population readout, $\tau_r^y > \tau$, within-neuron feedback is negative, and can thus be interpreted as spike-triggered *adaptation*. On the contrary, if the single neuron readout is faster than the population readout, $\tau_r^E < \tau$, the within-neuron feedback is positive and can thus be interpreted as

spike-triggered *facilitation*. In a special case where the time constant of the single neuron and population readout are assumed to be equal, within-neuron feedback vanishes.

Finally, we here derived the non-specific external current:

$$I_i^{\text{ext},y}(t) = C_m \left( \frac{c - \beta^y}{2} + \sigma^y \eta_i^y(t) \right), \qquad \sigma^y = \frac{\sigma_\xi^y}{\sqrt{2\tau}} \tag{29f}$$

that captures the ensemble of non-specific synaptic currents received by each single neuron. The non-specific current has a homogeneous mean across all neurons of the same cell type, and a neuron-specific fluctuation. The mean of the non-specific current can be traced back to the weighting of the metabolic cost over the encoding error in model objectives (*Equation 10*), while the fluctuation can be traced back to the noise strength that we assumed in the condition for spiking (*Equation 12*). The non-specific external current might arise because of synaptic inputs from other brain areas than the brain area that delivers feedforward projections to the E-I network we consider here, or it might result from synaptic activity of neurons that are part of the local network, but are not tuned to the feedforward input (*Zylberberg, 2018*).

We also recall the fast and slower time scales of single neuron activity:

$$f_i^y(t) = \sum_\alpha \delta(t - t_i^{y,\alpha}),$$
$$\dot{r}_i^y(t) = -(\tau_r^y)^{-1} r_i^y(t) + f_i^y(t), \tag{29g}$$

and the connectivity matrices

$$J_{ij}^{IE} = \left[ (\vec{w}_i^I)^\top \vec{w}_j^E \right]_+, \qquad J_{ij}^{II} = \left[ (\vec{w}_i^I)^\top \vec{w}_j^I \right]_+, i \neq j, \qquad J_{ij}^{EI} = \left[ (\vec{w}_i^E)^\top \vec{w}_j^I \right]_+. \tag{29h}$$

The structure of synaptic connectivity is fully determined by the similarity of tuning vectors of the presynaptic and the postsynaptic neurons ($\vec{w}_j^x$ and $\vec{w}_i^y$, respectively), while the distribution of synaptic connectivity weights is fully determined by the distribution of tuning parameters $w_{ki}^y$.

## Stimulus features

We define stimulus features $\vec{s}(t)$ as a set of $k = 1, \ldots, M$ independent Ornstein-Uhlenbeck processes with vanishing mean, standard deviation $\sigma^s$ and the correlation time $\tau_k^s$,

$$\tau_k^s \dot{s}_k(t) = -s_k(t) + \sqrt{2\tau_k^s} \sigma^s \eta_k(t). \tag{30}$$

If not mentioned otherwise, we use the following parameters, identical across stimulus features: $\sigma^s = 2 \, (\text{mV})^{1/2}$ and $\tau_k^s = \tau_s = 10$ ms. Variables $\eta_k(t)$ are independent Gaussian white noise processes with zero mean and covariance function $\langle \eta_k(t)\eta_l(t') \rangle = \delta_{kl}\delta(t - t')$. These variables should not be confused with the Gaussian white noises $\eta_k^y(t)$ in *Equation 28*.

## Parametrization of synaptic connectivity

In the efficient E-I model, synaptic weights $J_{ij}^{yx}$ are parametrized by tuning parameters $w_{ki}^y$ through *Equation 24*. The total number of synapses in the E-I, I-I, and I-E connectivity matrices (including silent synapses with zero synaptic weight) is $n_{\text{syn}} = 2N^E N^I + (N^I)^2$, while the number of tuning parameters is $n_w = M(N^E + N^I)$. Because the number of stimulus features $M$ is expected to be much smaller than the number of E or I neurons, the number of tuning parameters $n_w$ is much smaller than the number of synapses $n_{\text{syn}}$.

We can achieve a further substantial decrease in the number of free parameters by using a parametric distribution of tuning parameters $w_{ki}^y$. We set the tuning parameters following a normal distribution and found that excellent performance can be achieved with random draws of tuning parameters from the normal distribution, thus without searching for a specific set of tuning parameters. This drastically decreased the number of free parameters relative to synaptic weights to only a handful of parameters that determine the distributions of tuning parameters.

Given $M$ features, we sampled tuning parameters, $\vec{w}_i^y = [w_{1i}, \ldots, w_{Mi}]^\top$, with $i = 1, \ldots, N^y$, $y \in \{E, I\}$, as random points uniformly distributed on a $M$-dimensional sphere of radius $\sigma_w^y$. We obtained this by

sampling, for each neuron, a vector of $M$ i.i.d. standard Gaussian random variables, $\vec{\xi}_i^y = [\xi_{1i}^y, \ldots, \xi_{Mi}^y]^\top$, with $\xi_{ki}^y \sim \mathcal{N}(0,1)$, and normalizing the vector such as to have length equal to $\sigma_w^y$ (**Muller, 1959**),

$$\vec{w}_i^y = \sigma_w^y \frac{\vec{\xi}_i^y}{\|\vec{\xi}_i^y\|_2}, \qquad y \in \{E, I\}. \tag{31}$$

This ensures that the length of tuning vectors $\vec{w}_i^y$ in **Equation 31** is homogeneous across neurons of the same cell type, that is $\|\vec{w}_i^y\|_2 = \sigma_w^y$. Parameters $\sigma_w^E$ and $\sigma_w^I$ determine the heterogeneity (spread) of tuning parameters.

By combining **Equation 24** and **Equation 31**, we obtain the synaptic weights, $J_{ij}^{yx}$, as a function of the angle, $\alpha_{ij}^{xy}$, between the tuning vectors of presynaptic neurons, $\vec{w}_i^x$, and postsynaptic neurons, $\vec{w}_j^y$,

$$J_{ij}^{yx} = \sigma_w^y \sigma_w^x \left[ \cos \alpha_{ij}^{yx} \right]_+. \tag{32}$$

In the $M = 3$ dimensional case, we have that the distribution of the angle between two vectors is $p(\alpha_{ij}^{yx}) = \frac{1}{2} \sin(\alpha_{ij}^{yx})$, with $\alpha_{ij}^{yx} \in [0, \pi]$. Thus, the average strength of synaptic weights between the pre- and the postsynaptic population can be calculated as

$$\begin{aligned}
\langle J_{ij}^{yx} \rangle &= \frac{1}{2} \sigma_w^y \sigma_w^x \int_0^\pi d\alpha_{ij}^{yx} \sin(\alpha_{ij}^{yx}) \left[ \cos(\alpha_{ij}^{yx}) \right]_+ \\
&= \frac{1}{4} \sigma_w^y \sigma_w^x.
\end{aligned} \tag{33}$$

Thus, the upper bound for the synaptic weight between cell types $x$ and $y$ is simply

$$\max \left( J_{ij}^{yx} \right) = \sigma_w^y \sigma_w^x. \tag{34}$$

From the **Equation 33**, we have that the mean E-I connectivity is equal to the mean I-E connectivity, $\langle J_{ij}^{EI} \rangle = \langle J_{ij}^{IE} \rangle$. As we consider the ratio of the mean connectivity between I-I and E-I connections, we find that it is given by the following:

$$\frac{\langle J_{ij}^{II} \rangle}{\langle J_{ij}^{EI} \rangle} = \frac{\left( \sigma_w^I \right)^2}{\sigma_w^I \sigma_w^E} = \frac{\sigma_w^I}{\sigma_w^E}. \tag{35}$$

## Performance measures

### Average encoding error and average metabolic cost

The definition of the time-dependent loss functions (**Equation 10**) induces a natural choice for the performance measure: the mean squared error (MSE) between the targets and their estimators for each cell type. In the case of the E population, the time-dependent encoding error is captured by the variable $\epsilon^E(t)$ in the **Equation 11** and in case of I population it is captured by $\epsilon^I(t)$ defined in the same equation. We used the root MSE (RMSE), a standard measure for the performance of an estimator (**Gerstner et al., 2014**). For the cell type $y \in \{E, I\}$ in trial $q$, the RMSE is measured as

$$\mathrm{RMSE}^y = \sqrt{\langle \epsilon_q^y(t) \rangle_{t,q}}, \tag{36}$$

with $\langle z_q(t) \rangle_{t,q}$ denoting the time- and trial-average.

Following the definition of the time-dependent metabolic cost in the loss functions (**Equation 10**), we measured the average metabolic cost in a trial $q$ for the cell type $y \in \{E, I\}$ as

$$\mathrm{MC}^y = \sqrt{\langle \kappa_q^y(t) \rangle_{t,q}}, \tag{37}$$

with time-dependent metabolic cost $\kappa^y(t)$ as in model's objectives (**Equation 11**) and $\langle z_q(t) \rangle_{t,q}$ the time- and trial-average. The square root was taken to have the same scale as for the RMSE (see **Equation 36**).

## The bias of the estimator

The MSE can be decomposed into the bias and the variance of the estimator. The time-dependent bias of estimates $\hat{x}_k^y(t)$, $y \in \{E, I\}$, were evaluated for each time point over $q = 1, \ldots, Q$ trials. The time-dependent bias in input dimension $k = 1, \ldots, M$ is defined as

$$B_k^E(t) = \frac{1}{Q} \sum_{q=1}^{Q} [\hat{x}_{k,q}^E(t) - x_k(t)],$$

$$B_k^I(t) = \frac{1}{Q} \sum_{q=1}^{Q} [\hat{x}_{k,q}^I(t) - \langle \hat{x}_{k,q}^E(t) \rangle_q], \tag{38a}$$

with $\langle z_q(t) \rangle_q$ the trial-averaged realization at time $t$. To have an average measure of the encoding bias, we averaged the bias of estimators over time and over input dimensions:

$$B^y = \frac{1}{TM} \sum_{k=1}^{M} \int_0^T B_k^y(t) dt. \tag{38b}$$

The averaging over time and input dimensions is justified because $s_k(t)$ are independent realizations of the Ornstein-Uhlenbeck process (see *Equation 30*) with vanishing mean and with the same time constant, and variance across input dimensions.

## Criterion for determining optimal model parameters

The equations of the E-I spiking network in *Equation 29a*, *Equation 29b*, *Equation 29c*, *Equation 29d*, *Equation 29e*, *Equation 29f*, *Equation 29g*, *Equation 29h* (Materials and methods), derived from the instantaneous loss functions, give efficient coding solutions valid for any set of parameter values. However, to choose parameters values in simulated data in a principled way, we performed a numerical optimization of the performance function detailed below. Numerical optimization gave the set of optimal parameters listed in *Table 1*. When testing the efficient E-I model with simulations, we used the optimal parameters in *Table 1* and changed only the parameters plotted in the figure axes on a figure-by-figure basis.

To estimate the optimal set of parameters $\theta = \theta^*$, we performed a grid search on each parameter $\theta_i$ while keeping all other parameters fixed as specified in *Table 1*. While varying the parameters, we measured a weighted sum of the time- and trial-averaged encoding error and metabolic cost. For each cell type $y \in \{E, I\}$, we computed

$$\mathcal{L}_\theta^y = g_L \sqrt{\langle \epsilon_q^y(t \mid \theta) \rangle_{t,q}} + (1 - g_L) \sqrt{\langle \kappa_q^y(t \mid \theta) \rangle_{t,q}}, \tag{39a}$$

with $\langle z_q(t) \rangle_{t,q}$ the average over time and over trials and with $\epsilon^y(t)$ and $\kappa^y(t)$ as in model's objectives (*Equation 11*), where $g_L \in [0, 1]$ is a weighting factor.

To optimize the performance measure, we used a value of $g_L = 0.7$. The parameter $g_L$ in the *Equation 39a* regulates the relative importance of the average encoding error over the average metabolic cost. Since the performance measure in *Equation 39a* is closely related to the average over time and trials of the instantaneous loss function (*Equation 10*) where the parameter $\beta$ regulates the relative weight of instantaneous encoding error over the metabolic cost, setting $g_L$ is effectively achieved by setting $\beta$.

The optimal parameter set $\theta = \theta^*$ reported in *Table 1* is the parameter set that minimizes the sum of losses across E and I cell type

$$\theta^* = \arg\min_\theta \left( \mathcal{L}_\theta^E + \mathcal{L}_\theta^I \right). \tag{39b}$$

For visualization of the behavior of the average metabolic cost (*Equation 37*) and average loss (*Equation 39a*) across a range of a specific parameter $\theta_i$, we summed these measures across the E and I cell type and normalized them across the range of tested parameters.

The exact neural dynamics and performance of our model depends on the realizations of random variables which describe the the tuning parameters $w_{ki}^y$, the Gaussian noise in the non-specific currents

$\eta_i^y(t)$, and the initial conditions of the membrane potential $V_i^y(t=0)$, that were randomly drawn from a normal distribution in each simulation trial. To capture the performance of a 'typical' network, we iterated the performance measures across trials with different realizations of these random variables, and averaged the performance measures across trials. We typically used 100 simulation trials for each parameter value.

## Functional activity measures

### Tuning similarity

The pairwise tuning similarity was measured as the cosine similarity (*Luo et al., 2018*), defined as:

$$\Phi_{ij}^{yx} = \cos\alpha(\vec{w}_i^y, \vec{w}_j^x) = \frac{(\vec{w}_i^y)^\top \vec{w}_j^x}{\|\vec{w}_i^y\|_2 \|\vec{w}_j^x\|_2}, \qquad y, x \in \{E, I\}, \tag{40}$$

with $\|\vec{w}_i^y\|_2 = \sqrt{\sum_{k=1}^M (w_{ki}^y)^2}$ the length of the tuning vector in Euclidean space and $\alpha$ the angle between the tuning vectors $\vec{w}_j^x$ and $\vec{w}_i^y$.

### Cross-correlograms of spike timing

The time-dependent coordination of spike timing was measured with the cross-correlogram (CCG) of spike trains, corrected for stimulus-driven coincident spiking. The raw cross-correlogram (CCG) for neuron $i$ of cell type $y$ and neuron $j$ of cell type $x$ was measured as follows:

$$C_{ij}^{yx}(\tau) = \frac{1}{Q} \sum_{q=1}^Q \int_0^T f_{i,q}^y(t) f_{i,q}^y(t+\tau)\, dt, \tag{41a}$$

with $q = 1, \ldots, Q$ simulation trials with identical stimulus and $T$ the duration of the trial. We subtracted from the raw CCG the CCG of trial-invariant activity. To evaluate the trial-invariant cross-correlogram, we first computed the peri-stimulus time histogram (PSTH) for each neuron as follows:

$$P_i^y(t) = \frac{1}{Q} \sum_{q=0}^Q f_{i,q}^y(t). \tag{41b}$$

The trial-invariant CCG was then evaluated as the cross-correlation function of PSTHs between neurons $i$ and $j$,

$$S_{ij}^{yx}(\tau) = \int_0^T P_i^y(t) P_j^x(t+\tau)\, dt. \tag{41c}$$

Finally, the temporal coordination of spike timing was computed by subtracting the correction term from the raw CCG:

$$c_{ij}^{yx}(\tau) = C_{ij}^{yx}(\tau) - S_{ij}^{yx}(\tau). \tag{41d}$$

### Average imbalance of synaptic inputs

We considered time and trial-averaged synaptic inputs to each E and I neuron $i$ in trial $q$, evaluated as:

$$\bar{A}_{i,q}^{\mathrm{net},E} = \frac{1}{TC_m} \int_0^T I_{i,q}^{\mathrm{syn},E}(t)\, dt,$$

$$\bar{A}_{i,q}^{\mathrm{net},I} = \frac{1}{TC_m} \int_0^T I_{i,q}^{\mathrm{syn},I}(t)\, dt, \tag{42}$$

with synaptic currents to E neurons $I_{i,q}^{\mathrm{syn},E}(t)$ and to I neurons $I_{i,q}^{\mathrm{syn},I}(t)$ as in *Equation 29c*. Synaptic inputs were measured in units of mV. We reported trial-averages of the net synaptic inputs from the *Equation 42*.

## Instantaneous balance of synaptic inputs

We measured the instantaneous balance of synaptic inputs as the Pearson correlation of time-dependent synaptic inputs incoming to the neuron $i$. For those synaptic inputs that are defined as weighted delta-spikes (for which the Pearson correlation is not well defined; see *Equation 29c*), we convolved spikes with a synaptic filter $F(t) = \exp(-\frac{t}{\tau_{syn}})$,

$$
\begin{aligned}
A_{i,q}^{IE}(t) &= \sum_{j=1}^{N^E} J_{ij,q}^{IE} \int_0^t f_{j,q}^E(t-s)F(s)ds, \\
A_{i,q}^{II}(t) &= \sum_{\substack{j=1 \\ i \neq j}}^{N^I} J_{ij,q}^{II} \int_0^t f_{j,q}^I(t-s)F(s)ds, \\
A_{i,q}^{EI}(t) &= \sum_{j=1}^{N^I} J_{ij,q}^{EI} \int_0^t f_{j,q}^I(t-s)F(s)ds, \\
A_{i,q}^{\text{ff}}(t) &= C_m^{-1} I_{i,q}^{\text{ff}}(t),
\end{aligned}
\tag{43}
$$

where we used the expression for the feedforward synaptic current from the *Equation 29d*. Note that the feedforward synaptic current is already already low-pass filtered (see *Equation 30*). Using synaptic inputs from the *Equation 43*, we computed the Pearson correlation of synaptic inputs incoming to single E neurons, $\rho_{i,q}^E \left( A_{i,q}^{IE}(t), A_{i,q}^{II}(t) \right)$ for $i = 1, \ldots, N^E$, and to single I neurons, $\rho_{i,q}^I \left( A_{i,q}^{EI}(t), A_{i,q}^{\text{ff}}(t) \right)$ for $i = 1, \ldots, N^I$. The coefficients were then averaged across trials.

## Tuning curves and selectivity index

The selectivity index of a neuron captures the change in neuron's firing rate in response to a change in the stimulus. We first evaluated the tuning curve of each neuron by measuring the firing rate of the neuron $z_i^y(s_1), y \in \{E, I\}$, as a function of the amplitude of the stimulus feature $s_1$. The firing rate was evaluated from the network response to $M = 3$ stimulus features that were constant over time. We varied the first stimulus feature $s_1$ from strongly negative ($s_1 = -5$) to strongly positive values ($s_1 = s_{\max} = 5$), while the two other features were kept at an intermediate positive value ($s_2 = s_3 = 1.6$). Note that with all three features at such intermediate value ($s_1 = s_2 = s_3 = 1.6$), the average firing rate was about 8 Hz in E and 12 Hz in I neurons. To evaluate the tuning curve of a neuron, we measured its firing rate in 100 simulation trials of 1 s duration, for each value of the stimulus feature $s_1$.

To evaluate the sensitivity index, we normalized the tuning curve of the neuron with its maximal value,

$$
\tilde{z}_i^y(s_1) = \frac{z_i^y(s_1)}{\max(z_i^y(s_1))}
\tag{44a}
$$

We then computed the sensitivity index as the average absolute change of the normalized firing rate with the change in the stimulus:

$$
\alpha_i^y = \frac{1}{2s_{\max}} \int_{-s_{\max}}^{s_{\max}} \left| \frac{d\tilde{z}_i^y}{ds_1} \right| ds_1.
\tag{44b}
$$

## Perturbation experiments

### Perturbation of neural activity

Empirical studies (*Chettih and Harvey, 2019*; *Oldenburg et al., 2024*) suggested experiments with perturbation of neural activity that estimate functional connectivity in recurrently connected neural networks. Here, we detail the procedure on how we performed similar experiments on simulated neural networks. To evaluate the functional connectivity between pairs of neurons, we measured the effect of activation of a single E neuron ('target' neuron) on the activity of other neurons. We stimulated a randomly chosen E neuron with a depolarizing input, capturing the effect of photostimulation in empirical studies (*Chettih and Harvey, 2019*; *Oldenburg et al., 2024*), and measured the deviation of the firing rate from the baseline in all other neurons.

The time-dependent deviation of the firing rate from the baseline for neuron $i$ of type $y \in \{E, I\}$ was computed as $\Delta z_i^y(t) = z_i^y(t) - \bar{z}_i^y$, with $z_i^y(t) = (\tau_r^y)^{-1} r_i^y(t)$ the estimate of the instantaneous firing rate and $\bar{z}_i^y$ the average spontaneous firing rate of the neuron $i$. The target neuron received a constant depolarizing current during 50 ms and the effect of its activity on other neurons was measured during a time window of [0, 100] ms with respect to the onset of the stimulation. The functional connectivity between the target neuron and every other neuron in the network was then computed as the time average of the variable $\Delta z_i^y(t)$. To isolate the functional effect of recurrent connections on firing rate changes, we performed these experiments in a network without external stimuli, setting $s_k(t) = 0 \, \forall t, k$.

## Removal of connectivity structure

To better understand the effect of optimally structured recurrent connectivity (as given by the *Equation 24*) on network's activity and efficiency, we compared networks with and without the connectivity structure. To fully remove the connectivity structure, we randomly permuted, without repetition, recurrent connectivity weights between all neuronal pairs of all the three recurrent connectivity matrices. This was achieved by shuffling entries within each recurrent connectivity matrix. This procedure preserves all properties of the distribution of connectivity weights and only removes the connectivity structure. Shuffling of connections was iterated across 200 simulation trials, with each trial implementing a different random permutation of the connectivity. Dale's law is preserved by such manipulation.

To compare the performance of models with structured and unstructured connectivity (as reported on *Figure 4A*), we collected the low-pass filtered spiking activity in networks with and without connectivity structure. We used this neural activity to train a linear decoder with least squares method that minimizes the Euclidean distance between target signals and a linear readout of low-pass filtered spikes. The output of the training was a set of linear coefficients akin to decoding weights $w_{ki}^y$. We used these decoding weights estimated by the decoder to weight spikes in a held-out validation set. The performance was measured with root mean squared error (RMSE) between target signals and their estimates in the validation set. The training set comprised 70% of trials (140 trials), and the validation test comprised the remaining 30% of trials (60 trials).

To compare networks with and without connectivity structure about their metabolic cost, firing rate, variability of spiking and the E-I balance (*Figure 4B–G*), we performed these measures in networks with and without connectivity structure and plotted their distributions across 200 simulation trials. For the comparison of the metabolic cost (*Figure 4B*), we additionally matched the network with and without the connectivity structure about their mean net synaptic input to E and I neurons, to see if the difference in the metabolic cost between structured and unstructured networks persists after such matching. For the comparison of the coefficient of variation in structured and unstructured networks (*Figure 4E*), we used a constant stimulus instead of the OU stimulus, to exclude possible effects of time-dependent variations of the stimulus on the variability of spiking. Constant stimulus was homogeneous across all stimulus dimensions, $s_k(t) = 1.6, \ \forall k = 1, \ldots, M$. The amplitude of the constant stimulus was set such that the average firing rate in response to the constant stimulus matched the firing rate in response to the OU stimulus.

For the comparison of the voltage correlations and the effective connectivity between structured and unstructured networks (*Figure 4H–I*), we additionally permuted individual connectivity (sub) matrices. This gave four cases, namely, permuted E-I, I-I, I-E, and "all", with "all" meaning that all three recurrent connectivity matrices have been randomly permuted.

We also tested networks where the connectivity structure was not fully but only partially removed. There, we limited random permutation of synaptic weights to pairs of neurons that already had a connection in the structured network. By the *Equation 24*, connected neurons are those with positive tuning similarity, that is, neuronal pairs for which the following holds: $\Phi_{ij}^{yx} \geq 0$, with tuning similarity as in *Equation 40*. We compared partially unstructured networks with structured networks by plotting measures of neural activity in structured and partially unstructured networks across 200 simulation trials (*Figure 4—figure supplement 1B–E*).

## Perturbation of connectivity

To test the robustness of the model to random perturbations of synaptic weights (*Figure 4—figure supplement 1G–H*), we applied a random jitter to optimally efficient recurrent synaptic connectivity weights. The random jitter was proportional to the optimal synaptic weight, $\tilde{J}_{ij}^{yx} = J_{ij}^{yx}(1 + \sigma_J Z_{ij}^{yx})$,

where $\sigma_J$ is the strength of the perturbation and $Z_{ij}^{yx}$ are independent standard normal random variables. All three recurrent connectivity matrices (E-I, I-I, and I-E) were randomly perturbed at once.

## Computer simulations

We ran computer simulations with Matlab R2023b (Mathworks). The membrane equation for each neuron was integrated with Euler integration scheme with the time step of $dt = 0.02$ ms. The simulation of the E-I network with 400 E units and 100 I units for an equivalent of 1 s of neural activity lasted approximately 1.65 s on a laptop.

## Code availability

The complete computer code to reproduce the results can be downloaded anonymously from a public GitHub repository https://github.com/VeronikaKoren/efficient_EI and has the associated DOI: https://doi.org/10.5281/zenodo.14628524.

## Acknowledgements

VK and TS thank Tatiana Engel for her contribution to the discussion of results and for her comments on an earlier version of the manuscript.

## Additional information

### Funding

| Funder | Grant reference number | Author |
| --- | --- | --- |
| Technische Universität Berlin | Equal Opportunity Program | Veronika Koren |
| Technische Universität Berlin | Internal Research Funding | Tilo Schwalger |
| NIH BRAIN Initiative | U19 NS107464 | Stefano Panzeri |
| Simons Foundation Autism Research Initiative | 982347 | Stefano Panzeri |
| Marie Sklodowska-Curie Actions | 10.3030/101152984 | Simone Blanco Malerba |
| NIH BRAIN Initiative | R01 NS109961 | Stefano Panzeri |
| NIH BRAIN Initiative | R01 NS108410 | Stefano Panzeri |

The funders had no role in study design, data collection and interpretation, or the decision to submit the work for publication.

### Author contributions

Veronika Koren, Conceptualization, Data curation, Software, Formal analysis, Funding acquisition, Validation, Investigation, Visualization, Methodology, Writing – original draft, Writing – review and editing; Simone Blanco Malerba, Formal analysis, Funding acquisition, Validation, Writing – review and editing; Tilo Schwalger, Resources, Supervision, Funding acquisition, Validation, Writing – review and editing; Stefano Panzeri, Conceptualization, Resources, Supervision, Funding acquisition, Writing – original draft, Project administration, Writing – review and editing

### Author ORCIDs

Veronika Koren https://orcid.org/0000-0003-2920-2717
Simone Blanco Malerba http://orcid.org/0000-0002-4467-5988
Tilo Schwalger https://orcid.org/0000-0002-5422-3723
Stefano Panzeri https://orcid.org/0000-0003-1700-8909

Reviewer #1 (Public review): https://doi.org/10.7554/eLife.99545.4.sa1

Reviewer #2 (Public review): https://doi.org/10.7554/eLife.99545.4.sa2
Reviewer #3 (Public review): https://doi.org/10.7554/eLife.99545.4.sa3
Author response https://doi.org/10.7554/eLife.99545.4.sa4

## Additional files

### Supplementary files
MDAR checklist

### Data availability
The current manuscript is a computational study, and no data have been generated for this manuscript. The complete computer code for reproducing the results can be downloaded anonymously froma public GitHub repository: https://github.com/VeronikaKoren/efficient_EI, with a copy archived at *Koren, 2025* and with the associated DOI: https://doi.org/10.5281/zenodo.14628524.

The following dataset was generated:

| Author(s) | Year | Dataset title | Dataset URL | Database and Identifier |
| --- | --- | --- | --- | --- |
| Koren V | 2025 | VeronikaKoren/efficient_EI: third release | https://doi.org/10.5281/zenodo.14628524 | Zenodo, 10.5281/zenodo.14628524 |

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

## Appendix 1

### Derivation of the one cell type model

An efficient spiking model network with one cell type (1CT) has been developed previously (***Boerlin et al., 2013***), and properties of the 1CT model where the computation is assumed to be the leaky integration of inputs has been addressed in a number of previous studies (***Bourdoukan et al., 2012***; ***Barrett et al., 2016***; ***Koren and Denève, 2017***; ***Gutierrez and Denève, 2019***; ***Brendel et al., 2020***). Compared to the efficient E-I model, the 1CT model can be seen as a simplification, and can be treated similarly to the E-I model, which is what we demonstrate in this section.

As the name of the model suggests, all neurons in the 1CT model are of the same cell type, and we have $i = 1, \ldots, N$ such neurons. We can then use the definitions in ***Equations 6–9*** (now without the index $y$) and a loss function similar to the one in ***Koren and Denève, 2017***, but with only one (quadratic) regularizer

$$L^{1CT}(t) = \left\| \vec{x}(t) - \hat{\vec{x}}(t) \right\| + \beta^1 \sum_{i=1}^{N} r_i^2(t), \tag{45}$$

with $\beta^1 > 0$. The encoding error of the 1CT model minimizes the squared distance between the target signal $\vec{x}(t)$ and the estimate $\hat{\vec{x}}(t)$. As we apply the condition for spiking as for the E-I network (***Equation 12*** without the index $y$) and follow the same steps as for the E-I network, we get

$$\vec{w}_i^\top \left( \vec{x}(t) - \hat{\vec{x}}(t) \right) - \beta^1 r_i(t) > \frac{1}{2} \left( \|\vec{w}_i\|^2 + \beta^1 - \xi_i(t) \right), \tag{46}$$

with $\xi_i(t)$ the noise at the condition for spiking. Same as in the E-I model, we define the noise as an Ornstein-Uhlenbeck process with zero mean, obeying

$$\dot{\xi}_i(t) = -\lambda \xi_i(t) + \sqrt{2\lambda} \sigma_\xi \eta_i(t), \tag{47}$$

where $\eta_i$ is a Gaussian white noise and $\lambda = \tau^{-1}$ is the inverse time constant of the process.

We now define proxies of the membrane potential and the firing threshold as

$$u_i(t) := \vec{w}_i^\top \left( \vec{x}(t) - \hat{\vec{x}}(t) \right) - \beta^1 r_i(t),$$
$$\theta_i := \frac{1}{2} \left( \|\vec{w}_i\|^2 + \beta^1 - \xi_i(t) \right). \tag{48}$$

Differentiating the proxy of the membrane potential $u_i(t)$ and rewriting the model as an integrate-and-fire neuron, we get

$$\dot{u}_i(t) = -\frac{1}{\tau} u_i(t) + \vec{w}_i^\top \vec{s}(t) - \sum_{\substack{j=1 \\ j \neq i}}^{N} \vec{w}_i^\top \vec{w}_j f_j(t),$$
$$\text{if } u_i(t^-) \geq \theta_i \to u_i(t^+) = u_i^{\text{reset}},$$
$$\theta_i = \frac{1}{2} \left( \|\vec{w}_i\|_2^2 + \beta^1 - \xi_i(t) \right),$$
$$u_i^{\text{reset}} = \theta_i - \left( \|\vec{w}_i\|^2 + \beta^1 \right). \tag{49}$$

We now proceed in the same way as with the E-I model and define new variables

$$V_i(t) := u_i(t) + V_{\text{rest}} + \frac{1}{2} \left( c - \beta^1 + \xi_i(t) \right), \qquad V_{\text{rest}} < 0,$$
$$\vartheta_i := V_{\text{rest}} + \frac{1}{2} \left( \|\vec{w}_i\|^2 + c \right). \tag{50}$$

In these new variables, we can rewrite the membrane equation of the 1CT model as follows:

$$\tau \dot{V}_i(t) = -(V_i(t) - V_{\text{rest}}) + \tau \vec{w}_i^\top \vec{s}(t) - \tau \sum_{\substack{j=1 \\ j \neq i}}^{N} \vec{w}_i^\top \vec{w}_j f_j(t) + \frac{1}{2} \left( c - \beta^1 \right) + \sqrt{\frac{\tau}{2}} \sigma_\xi \eta_i(t). \tag{51}$$

Finally, we rewrite the model with a more compact notation of a leaky integrate-and-fire neuron model with transmembrane currents,

$$\tau \dot{V}_i(t) = - \left( V_i(t) - V_{\text{rest}} \right) + R_m \left( I_i^{\text{ff}}(t) + I_i^{\text{syn}}(t) + I_i^{\text{ext}}(t) \right),$$

$$\text{if } V_i(t^-) \geq \vartheta_i \rightarrow V_i(t^+) = V_i^{\text{reset}},$$

$$\vartheta_i = V_{\text{rest}} + \frac{1}{2} \left( \|\vec{w}_i\|_2^2 + c \right),$$ \hfill (52a)

$$V_i^{\text{reset}} = V_{\text{rest}} - \beta^1 + \frac{1}{2} \left( c - \|\vec{w}_i\|_2^2 \right),$$

with currents

$$I_i^{\text{ff}}(t) = C_m \left( \vec{w}_i^\top \vec{s}(t) \right),$$

$$I_i^{\text{syn}}(t) = C_m \left( \sum_{\substack{j=1 \\ j \neq i}}^{N} J_{ij} f_j(t) \right), \qquad J_{ij} = -\vec{w}_i^\top \vec{w}_j,$$ \hfill (52b)

$$I_i^{\text{ext}}(t) = C_m \left( \frac{c - \beta^1}{2} + \sigma^1 \eta_i(t) \right), \qquad \sigma^1 = \frac{\sigma_\xi}{\sqrt{2\tau}}.$$

Note that the model with one cell type does not obey Dale's law, since the same neuron sends to its postsynaptic targets excitatory and inhibitory currents, depending on the tuning similarity of the presynaptic and the postsynaptic neuron $\vec{w}_i$ and $\vec{w}_j$ (**Equation 52b**). In particular, if the pre- and postsynaptic neurons have similar selectivity ($\vec{w}_i^\top \vec{w}_j > 0$), the recurrent interaction is inhibitory, and if the neurons have different selectivity ($\vec{w}_i^\top \vec{w}_j < 0$), the interaction is excitatory. Simply put, neurons with similar selectivity inhibit each other while neurons with different selectivity excite each other (**Koren and Denève, 2017**).

Dale's law can be imposed to the 1CT model the same way as in the E-I model, by removing synaptic interactions between neurons with different selectivity with rectification of the connectivity matrix,

$$\tilde{J}_{ij} = -[\vec{w}_i^\top \vec{w}_j]_+.$$ \hfill (53)

However, this manipulation results in a network with only inhibitory recurrent synaptic interactions, and thus a network of only inhibitory neurons. Network with only inhibitory interactions is less relevant for the description of recurrently connected biological networks.

# Appendix 2

## Parameters of the E-I model without non-specific currents

Our analytical derivation in *Equation 25* suggested an efficient E-I model that is simpler with respect to the E-I model studied in this contribution, as it does not have non-specific synaptic currents. Optimal (computational) model parameters of such simpler model, listed above the double line are by definition identical to the full E-I model listed in *Table 1*. However, the model without non-specific synaptic currents differs from the full E-I model about the distance between the resting potential and the threshold. In the simpler model, this distance is lower compared to the full E-I model, and is not consistent with empirically measured distance, which is about 20 mV (*Constantinople and Bruno, 2013*).

**Appendix 2—table 1.** Table of optimal model parameters for the efficient E-I network without non-specific synaptic currents.

As in *Table 1*, for the E-I model without non-specific currents. The model is defined in *Equation 25*.

| Parameter | Notation | Value |
| --- | --- | --- |
| Number of E neurons | $N^E$ | 400 |
| Ratio of E to I neuron numbers | $N^E : N^I$ | 4:1 |
| Number of the input features | $M$ | 3 |
| Time constant of the population readout (E and I) | $\tau$ | 10ms |
| Time constant of the single neuron readout | $\tau_r^E = \tau_r^I$ | 10ms |
| Noise strength (non-specific current) | $\sigma$ | 5.0 mV |
| Heterogeneity factor of tuning parameters in E | $\sigma_w^E$ | 1.0 (mV)$^{1/2}$ |
| Ratio of mean E-I to I-I synaptic connectivity | mean E-I: mean I-I | 3:1 |
| Metabolic constant | $\beta$ | 14 mV |
| (Threshold constant) | $(c/2)$ | (0 mV) |
| Distance threshold to reset potential (E neurons) | $|\vartheta^E - V_{\text{rest}}^E|$ | 7.5 mV |
| Distance threshold to reset potential (I neurons) | $|\vartheta^I - V_{\text{rest}}^I|$ | 11.5 mV |
| Connection probability (recurrent synapses) | $p^{IE} = p^{II} = p^{EI}$ | 0.5 |
| Mean E-I synaptic weight (EPSP to I at max) | $\langle J_{ij}^{IE} \rangle$ | 0.75 mV |
| Mean I-E synaptic weight (IPSP to E at max) | $\langle J_{ij}^{EI} \rangle$ | 0.75 mV |
| Mean I-I synaptic weight (IPSP at max) | $\langle J_{ij}^{II} \rangle$ | 2.25 mV |

A simple way to increase the distance between the resting potential and the firing threshold is to introduce a constant that multiplies all mathematical terms in the *Equation 25*. While this allows to achieve biologically plausible values for the distance between the resting potential and the threshold, it leads to values of mean recurrent synaptic connectivity $\langle J_{ij}^{IE} \rangle$, $\langle J_{ij}^{II} \rangle$ and $\langle J_{ij}^{EI} \rangle$ that are stronger than typically reported in the empirical literature (*Campagnola et al., 2022*).

## Appendix 3

### Analysis of the one cell type model and comparison with the E-I model

We re-derived the 1CT model as a simplification of the E-I network (Appendix 1, *Figure 1—figure supplement 1A–B*), with objective function of the same form as $L^E$ and by allowing a single type of neurons sending both excitatory and inhibitory synaptic currents to their post-synaptic targets (*Figure 1—figure supplement 1C*). Similarly to the E-I model, also the 1CT model exhibits structured connectivity, with synaptic strength depending on the tuning similarity between the presynaptic and the postsynaptic neuron. Pairs of neurons with stronger tuning similarity (dissimilarity) have stronger mutual inhibition (excitation); see *Figure 1—figure supplement 1D*.

We compared the coding performance of the E-I model with that of a fully connected 1CT model. Both models received the same set of stimulus features and performed the same computation. In the 1CT model, tuning parameters were drawn from the same distribution as used for the E neurons in the E-I model. We used the same membrane time constant $\tau$ in both models, while the metabolic constants ($\beta$ of the E-I model and $\beta^1$ of the 1CT model) and the noise strength ($\sigma$ of the E-I model and $\sigma^1$ of the 1CT model) were chosen such as to optimize the average loss for each model (*Figure 6B* for E-I model, *Figure 1—figure supplement 1F–G* for 1CT model). Parameters of the 1CT model are listed in the Table below. A qualitative comparison of the E-I and the 1CT model showed that with optimal parameters, both models accurately tracked multiple target signals (*Figure 1G*, *Figure 1—figure supplement 1E*).

To compare the performance of the E-I and the 1CT models also quantitatively, we measured the average encoding error (RMSE), metabolic cost (MC) and loss of each model. The RMSE and the MC in the 1CT model were measured as in *Equation 36* and *Equation 37*, while the average loss of each model was evaluated as follows:

$$
\begin{aligned}
\mathcal{L}^{\text{1CT}} &= g_L \sqrt{\langle \epsilon_q^{\text{1CT}}(t) \rangle_{t,q}} + (1 - g_L) \sqrt{\langle \kappa_q^{\text{1CT}}(t) \rangle_{t,q}}, \\
\mathcal{L}^{\text{E-I}} &= g_L \sqrt{\langle \epsilon_q^E(t) \rangle_{t,q}} + (1 - g_L) \sqrt{\langle \kappa_q^E(t) \rangle_{t,q}}.
\end{aligned}
\tag{54}
$$

Unless mentioned otherwise, we weighted stronger the encoding error compared to the metabolic cost and used $g_L = 0.7$.

**Appendix 3—table 1.** Table of default model parameters for the efficient network with one cell type.
The parameters $N$, $M$, $\tau$ and $\sigma_w^1$ were chosen identical to the E-I network (see *Table 1* in the main text). Parameters $\sigma^1$ and $\beta^1$ were determined as values that maximize network efficiency (see section "Performance measures" in Methods).

| Parameter | Notation | Value |
|---|---|---|
| Number of E neurons | $N$ | 400 |
| Number of the input features | $M$ | 3 |
| Time constant of the single neuron and population readout | $\tau$ | 10ms |
| Noise strength | $\sigma^1$ | 1.8 mV |
| SD of tuning parameters | $\sigma_w^1$ | 1 (mV)$^{1/2}$ |
| Metabolic constant | $\beta^1$ | 11.4 mV |

