## [Editor Report · eLife Assessment]

This study offers a **valuable** treatment of how the population of excitatory and inhibitory neurons integrates principles of energy efficiency in their coding strategies. The **convincing** analysis provides a comprehensive characterisation of the model, highlighting the structured connectivity between excitatory and inhibitory neurons. The role of the many free parameters are discussed and studied in depth.

---

## [Referee Report · Reviewer #1 (Public review)]

Koren et al. derive and analyse a spiking network model optimised to represent external signals using the minimum number of spikes. Unlike most prior work using a similar setup, the network includes separate populations of excitatory and inhibitory neurons. The authors show that the optimised connectivity has a like-to-like structure, which leads to the experimentally observed phenomenon of feature competition. The authors also examine how various (hyper)parameters-such as adaptation timescale, the excitatory-to-inhibitory cell ratio, regularization strength, and background current-affect the model. These findings add biological realism to a specific implementation of efficient coding. They show that efficient coding explains, or at least is consistent with, multiple experimentally observed properties of excitatory and inhibitory neurons.

As discussed in the first round of reviews, the model's ability to replicate biological observations such as the 4:1 ratio of excitatory vs. inhibitory neurons hinges on somewhat arbitrary hyperparameter choices. Although this may limit the model's explanatory power, the authors have made significant efforts to explore how these parameters influence their model. It is an empirical question whether the uncovered relationships between, e.g., metabolic cost and the fraction of excitatory neurons are biologically relevant.

The revised manuscript is also more transparent about the model's limitations, such as the lack of excitatory-excitatory connectivity.

---

## [Referee Report · Reviewer #2 (Public review)]

Summary:

In this work, the authors present a biologically plausible, efficient E-I spiking network model and study various aspects of the model and its relation to experimental observations. This includes a derivation of the network into two (E-I) populations, the study of single-neuron perturbations and lateral-inhibition, the study of the effects of adaptation and metabolic cost, and considerations of optimal parameters. From this, they conclude that their work puts forth a plausible implementation of efficient coding that matches several experimental findings, including feature-specific inhibition, tight instantaneous balance, a 4 to 1 ratio of excitatory to inhibitory neurons, and a 3 to 1 ratio of I-I to E-I connectivity strength.

Strengths:

While many network implementations of efficient coding have been developed, such normative models are often abstract and lacking sufficient detail to compare directly to experiments. The intention of this work to produce a more plausible and efficient spiking model and compare it with experimental data is important and necessary in order to test these models. In rigorously deriving the model with real physical units, this work maps efficient spiking networks onto other more classical biophysical spiking neuron models. It also attempts to compare the model to recent single-neuron perturbation experiments, as well as some long-standing puzzles about neural circuits, such as the presence of separate excitatory and inhibitory neurons, the ratio of excitatory to inhibitory neurons, and E/I balance. One of the primary goals of this paper, to determine if these are merely biological constraints or come from some normative efficient coding objective, is also important. Lastly, though several of the observations have been reported and studied before, this work arguably studies them in more depth, which could be useful for comparing more directly to experiments.

Weaknesses:

This work is the latest among a line of research papers studying the properties of efficient spiking networks. Many of the characteristics and findings here have been discussed before, thereby limiting the new insights that this work can provide. Thus, the conclusions of this work should be considered and understood in the context of those previous works, as the authors state. Furthermore, the number of assumptions and free parameters in the model, though necessary to bring the model closer to biophysical reality, make it more difficult to understand and to draw clear conclusions from. As the authors state, many of the optimality claims depend on these free parameters, such as the dimensionality of the input signal (M=3), the relative weighting of encoding error and metabolic cost, and several others. This raises the possibility that it is not the case that the set of biophysical properties measured in the brain are accounted for by efficient coding, but rather that theories of efficient coding are flexible enough to be consistent with this regime. With this in mind, some of the conclusions made in the text may be overstated and should be considered in this light.

Conclusions, Impact, and additional context:

Notions of optimality are important for normative theories, but they are often studied in simple models with as few free parameters as possible. Biophysically detailed and mechanistic models, on the other hand, will often have many free parameters by their very nature, thereby muddying the connection to optimality. This tradeoff is an important concern in neuroscientific models. Previous efficient spiking models have often been criticized for their lack of biophysically-plausible characteristics, such as large synaptic weights, dense connectivity, and instantaneous communication. This work is an important contribution in showing that such networks can be modified to be much closer to biophysical reality without losing their essential properties. Though the model presented does suffer from complexity issues which raise questions about its connections to "optimal" efficient coding, the extensive study of various parameter dependencies offers a good characterization of the model and puts its conclusions in context.

---

## [Referee Report · Reviewer #3 (Public review)]

Summary:

In their paper the authors tackle three things at once in a theoretical model: how can spiking neural networks perform efficient coding, how can such networks limit the energy use at the same time, and how can this be done in a more biologically realistic way than previous work.

They start by working from a long-running theory on how networks operating in a precisely balanced state can perform efficient coding. First, they assume split networks of excitatory (E) and inhibitory (I) neurons. The E neurons have the task to represent some lower dimensional input signal, and the I neurons have the task to represent the signal represented by the E neurons. Additionally, the E and I populations should minimize an energy cost represented by the sum of all spikes. All this results in two loss functions for the E and I populations, and the networks are then derived by assuming E and I neurons should only spike if this improves their respective loss. This results in networks of spiking neurons that live in a balanced state, and can accurately represent the network inputs.

They then investigate in depth different aspects of the resulting networks, such as responses to perturbations, the effect of following Dale's law, spiking statistics, the excitation (E)/inhibition (I) balance, optimal E/I cell ratios, and others. Overall, they expand on previous work by taking a more biological angle on the theory and show the networks can operate in a biologically realistic regime.

Strengths:

* The authors take a much more biological angle on the efficient spiking networks theory than previous work, which is an essential contribution to the field

* They make a very extensive investigation of many aspects of the network in this context, and do so thoroughly

* They put sensible constraints on their networks, while still maintaining the good properties these networks should have

Weaknesses:

* One of the core goals of the paper is to make a more biophysically realistic network than previous work using similar optimization principles. One of the important things they consider is a split into E and I neurons. While this works fine, and they consider the coding consequences of this, it is not clear from an optimization perspective why the split into E and I neurons and following Dale's law would be beneficial. This would be out of scope for the current paper however.

* The theoretical advances in the paper are not all novel by themselves, as most of them (in particular the split into E and I neurons and the use of biophysical constants) had been achieved in previous models. However, the authors discuss these links thoroughly and do more in-depth follow-up experiments with the resulting model.

Assessment and context:

Overall, although much of the underlying theory is not necessarily new, the work provides an important addition to the field. The authors succeeded well in their goal of making the networks more biologically realistic, and incorporate aspects of energy efficiency. For computational neuroscientists this paper is a good example of how to build models that link well to experimental knowledge and constraints, while still being computationally and mathematically tractable. For experimental readers the model provides a clearer link of efficient coding spiking networks to known experimental constraints and provides a few predictions.

---

## [Author Response]

The following is the authors’ response to the previous reviews.

**Public Reviews:**

**Reviewer #1 (Public review):**
Koren et al. derive and analyse a spiking network model optimised to represent external signals using the minimum number of spikes. Unlike most prior work using a similar setup, the network includes separate populations of excitatory and inhibitory neurons. The authors show that the optimised connectivity has a like-to-like structure, which leads to the experimentally observed phenomenon of feature competition. The authors also examine how various (hyper)parameters-such as adaptation timescale, the excitatory-to-inhibitory cell ratio, regularization strength, and background current-affect the model. These findings add biological realism to a specific implementation of efficient coding. They show that efficient coding explains, or at least is consistent with, multiple experimentally observed properties of excitatory and inhibitory neurons.As discussed in the first round of reviews, the model's ability to replicate biological observations such as the 4:1 ratio of excitatory vs. inhibitory neurons hinges on somewhat arbitrary hyperparameter choices. Although this may limit the model's explanatory power, the authors have made significant efforts to explore how these parameters influence their model. It is an empirical question whether the uncovered relationships between, e.g., metabolic cost and the fraction of excitatory neurons are biologically relevant.The revised manuscript is also more transparent about the model's limitations, such as the lack of excitatory-excitatory connectivity. Further improvements could come from explicitly acknowledging additional discrepancies with biological data, such as the widely reported weak stimulus tuning of inhibitory neurons in the primary sensory cortex of untrained animals.

We thank the Reviewer for their insightful characterization of our paper and for further suggestions on how to improve it. We have now further improved the transparency about model’s limitations and we explicitly acknowledged the discrepancy with biological data about connection probability and about the selectivity of inhibitory neurons (pages 4 and 15).

**Reviewer #2 (Public review):**
Summary:In this work, the authors present a biologically plausible, efficient E-I spiking network model and study various aspects of the model and its relation to experimental observations. This includes a derivation of the network into two (E-I) populations, the study of single-neuron perturbations and lateral-inhibition, the study of the effects of adaptation and metabolic cost, and considerations of optimal parameters. From this, they conclude that their work puts forth a plausible implementation of efficient coding that matches several experimental findings, including feature-specific inhibition, tight instantaneous balance, a 4 to 1 ratio of excitatory to inhibitory neurons, and a 3 to 1 ratio of I-I to E-I connectivity strength.Strengths:While many network implementations of efficient coding have been developed, such normative models are often abstract and lacking sufficient detail to compare directly to experiments. The intention of this work to produce a more plausible and efficient spiking model and compare it with experimental data is important and necessary in order to test these models. In rigorously deriving the model with real physical units, this work maps efficient spiking networks onto other more classical biophysical spiking neuron models. It also attempts to compare the model to recent single-neuron perturbation experiments, as well as some long-standing puzzles about neural circuits, such as the presence of separate excitatory and inhibitory neurons, the ratio of excitatory to inhibitory neurons, and E/I balance. One of the primary goals of this paper, to determine if these are merely biological constraints or come from some normative efficient coding objective, is also important. Lastly, though several of the observations have been reported and studied before, this work arguably studies them in more depth, which could be useful for comparing more directly to experiments.Weaknesses:This work is the latest among a line of research papers studying the properties of efficient spiking networks. Many of the characteristics and findings here have been discussed before, thereby limiting the new insights that this work can provide. Thus, the conclusions of this work should be considered and understood in the context of those previous works, as the authors state. Furthermore, the number of assumptions and free parameters in the model, though necessary to bring the model closer to biophysical reality, make it more difficult to understand and to draw clear conclusions from. As the authors state, many of the optimality claims depend on these free parameters, such as the dimensionality of the input signal (M=3), the relative weighting of encoding error and metabolic cost, and several others. This raises the possibility that it is not the case that the set of biophysical properties measured in the brain are accounted for by efficient coding, but rather that theories of efficient coding are flexible enough to be consistent with this regime. With this in mind, some of the conclusions made in the text may be overstated and should be considered in this light.Conclusions, Impact, and additional context:Notions of optimality are important for normative theories, but they are often studied in simple models with as few free parameters as possible. Biophysically detailed and mechanistic models, on the other hand, will often have many free parameters by their very nature, thereby muddying the connection to optimality. This tradeoff is an important concern in neuroscientific models. Previous efficient spiking models have often been criticized for their lack of biophysically-plausible characteristics, such as large synaptic weights, dense connectivity, and instantaneous communication. This work is an important contribution in showing that such networks can be modified to be much closer to biophysical reality without losing their essential properties. Though the model presented does suffer from complexity issues which raise questions about its connections to "optimal" efficient coding, the extensive study of various parameter dependencies offers a good characterization of the model and puts its conclusions in context.

We thank the Reviewer for their thorough and accurate assessment of our paper.

**Reviewer #3 (Public review):**
Summary:In their paper the authors tackle three things at once in a theoretical model: how can spiking neural networks perform efficient coding, how can such networks limit the energy use at the same time, and how can this be done in a more biologically realistic way than previous work.They start by working from a long-running theory on how networks operating in a precisely balanced state can perform efficient coding. First, they assume split networks of excitatory (E) and inhibitory (I) neurons. The E neurons have the task to represent some lower dimensional input signal, and the I neurons have the task to represent the signal represented by the E neurons. Additionally, the E and I populations should minimize an energy cost represented by the sum of all spikes. All this results in two loss functions for the E and I populations, and the networks are then derived by assuming E and I neurons should only spike if this improves their respective loss. This results in networks of spiking neurons that live in a balanced state, and can accurately represent the network inputs.They then investigate in depth different aspects of the resulting networks, such as responses to perturbations, the effect of following Dale's law, spiking statistics, the excitation (E)/inhibition (I) balance, optimal E/I cell ratios, and others. Overall, they expand on previous work by taking a more biological angle on the theory and show the networks can operate in a biologically realistic regime.Strengths:* The authors take a much more biological angle on the efficient spiking networks theory than previous work, which is an essential contribution to the field* They make a very extensive investigation of many aspects of the network in this context, and do so thoroughly* They put sensible constraints on their networks, while still maintaining the good properties these networks should haveWeaknesses:* One of the core goals of the paper is to make a more biophysically realistic network than previous work using similar optimization principles. One of the important things they consider is a split into E and I neurons. While this works fine, and they consider the coding consequences of this, it is not clear from an optimization perspective why the split into E and I neurons and following Dale's law would be beneficial. This would be out of scope for the current paper however.* The theoretical advances in the paper are not all novel by themselves, as most of them (in particular the split into E and I neurons and the use of biophysical constants) had been achieved in previous models. However, the authors discuss these links thoroughly and do more in-depth follow-up experiments with the resulting model.Assessment and context:Overall, although much of the underlying theory is not necessarily new, the work provides an important addition to the field. The authors succeeded well in their goal of making the networks more biologically realistic, and incorporate aspects of energy efficiency. For computational neuroscientists this paper is a good example of how to build models that link well to experimental knowledge and constraints, while still being computationally and mathematically tractable. For experimental readers the model provides a clearer link of efficient coding spiking networks to known experimental constraints and provides a few predictions.

We thank the Reviewer for a positive assessment and for pointing out the merits of our work.

**Recommendations for the authors:**

**Reviewer #1 (Recommendations for the authors):**
The authors have addressed my previous concerns, and I agree that the manuscript has improved. However, I believe they could still do more to acknowledge two notable mismatches between the model and experimental data.(1) Stimulus selectivity of excitatory and inhibitory neuronsIn the model, excitatory and inhibitory neurons exhibit similar stimulus selectivity, which appears inconsistent with most experimental findings. The authors argue that whether inhibitory neurons are less selective remains an open question, citing three studies in support. However, only one of these studies (Ranyan) was conducted in primary sensory cortex and it is, to my knowledge, one of the few papers showing this (indeed, it's often cited as an exception). The other two studies (Kuan and Najafi) recorded from the parietal cortex of mice trained on decision making tasks, and therefore seem less relevant to the model.In contrast to the cited studies, the overwhelming majority of the work has found that inhibitory neurons in sensory cortex, in particular those expressing Parvalbumin, are less stimulus selective than excitatory cells. And this is indeed the prevailing view, as summarized by the review from Hu et al. (Science, 2014): "PV+ interneurons exhibit broader orientation tuning and weaker contrast specificity than pyramidal neurons." This view emerged from numerous classical studies, including Sohya et al. (J. Neurosci., 2007), Cardin (J. Neurosci., 2007), Nowak (Cereb. Cortex, 2008), Niell et al. (J. Neurosci., 2008), Liu (J. Neurosci., 2009), Kerlin (Neuron, 2010), Ma et al. (J. Neurosci., 2010), Hofer et al. (Nature Neurosci. 2011), and Atallah et al. (Neuron 2012). Weak inhibitory tuning has been confirmed by recent studies, such as Sanghavi & Kar (biorxiv 2023), Znamenskiy et al. (Neuron 2024), and Hong et al. (Nature, 2024).The authors should acknowledge this consensus and cite the conflicting evidence. Failing to do so is cherry picking from the literature. Since training can increase the stimulus selectivity of PV+ neurons to that of Pyr levels, also in primary visual cortex (Khan et al. Neuron 2018), a favourable interpretation of the model is that it represents a highly optimized, if not overtrained, state.

We have carefully considered the literature cited by the Reviewer. We agree with the interpretation that stimulus selectivity of inhibitory neurons in our model is higher than the stimulus selectivity of Parvalbumin-positive inhibitory neurons in the primary sensory cortex of naïve animals. We have edited the text in Discussion (page 14).

(2) Connection probabilityThe manuscript claims that "rectification sets the overall connection probability to 0.5, consistent with experimental results (Pala & Petersen; Campagnola et al.)." However, the cited studies, and others, report significantly lower probabilities, except for Pyr-PV (E-I connections in the model). For example, Campagnola et al. measured PV-Pyr connectivity at 34% in L2/3 and 20% in L5.It's perfectly acceptable that the model cannot replicate every detail of biological circuits. But it's important to be cautious when claiming consistency with experimental data.

Here as well, we agree with the Reviewer that the connection probability of 0.5 is consistent with reported connectivity of Pyr-PV neurons, but less so with reported connectivity of PV-Pyr neurons. We have now qualified our claim about compatibility of the connection probability in our model with empirical observations more precise (page 4).

**Reviewer #2 (Recommendations for the authors):**
I commend the authors for an extremely thorough and detailed rebuttal, and for all of the additional work put in to address the reviewer concerns. For the most part, I am satisfied with the current state of the manuscript.

We thank the Reviewer for recognizing our effort to address the first round of Reviews to our best ability.

Here are some small points still remaining that I think the authors should address:(1) Pg. 8, "We verified the robustness of the model to small deviations from the optimal synaptic weights" - while the authors now cite Calaim et al. 2022 in the discussion, its relevance to several of the results justify its inclusion in other places. Here is one place where the authors test something that was also studied in this previous paper.

The Reviewer is correct that Calaim et al. (eLife 2022) addressed the robustness of synaptic weights, and we now cited this study when describing our results on jiVering of synaptic connections (page 8).

(2) Pg. 9, "In our optimal E-I network we indeed found that optimal coding efficiency is achieved in absence of within-neuron feedback or with weak adaptation in both cell types" Pg. 10, "the absence of within-neuron feedback or the presence of weak and short-lasting spike-triggered adaptation in both E and I neurons are optimally efficient solutions" The authors seem to state that both weak adaptation and no adaptation at all are optimal. In contrast to the rest of the results presented, this is very vague and does not give a particular level of adaptation as being optimal. The authors should make this more clear.

We agree that the text about optimal level of adaptation was unclear. The optimal solution is no adaptation, while weak and short-lasting adaptation define a slightly suboptimal, yet still efficient, network state, as now stated on page 10.

(3) Pg. 13, "In summary our analysis suggests that optimal coding efficiency is achieved with four times more E neurons than I neurons and with mean I-I synaptic efficacy about 3 times stronger..." --- claims such as these are still too strong, in my opinion. It is rather the case that the particular ratio of E to I neurons and connections strengths can be made consistent with an optimally efficient regime.

We agree here as well. We have revised the text (page 13) to beVer explain our results.

(4) Pg. 14, "firing rates in the 1CT model were highly sensitive to variations in the metabolic constant" (Fig. 8I, as compared to Fig. 6C). This difference between the 1CT and E-I networks is striking, and I would suspect it is due to some idiosyncrasies in the difference between the two models (e.g., the relative amount of delay that it takes for lateral inhibition to take effect, or the fact that E-E connections have not been removed in this model). The authors should ideally back up this result with some justified explanation.

We agree with Reviewer that the delay for lateral inhibition in the E-I model is twice that of the 1CT model and that the E-I model gains stability from the lack of E-E connectivity. Furthermore, the tuning is stronger in I compared to E neurons in the E-I model, which contributes to making the E-I network inhibition-dominated (Fig. 1H). In contrast, the average excitation and inhibition in the 1CT model are of exactly the same magnitude. The property of being inhibition-dominated makes the E-I model more stable. We report these observations in the revised text (pages 14-15).

**Reviewer #3 (Recommendations for the authors):**
Overall my points were very well responded to and I removed most of my weaknesses.I appreciate the authors implementing my suggested analysis change for Figure 8, and I find the result very clear. I would further suggest they add a bit of text for the reader as to why this is done. For a new reader without much knowledge of these networks at first it seems the inhibitory population is very good at representation in fig 8G: so why is it not further considered in fig 8H?

We thank the reviewer for providing further suggestions. We now clarified in the text why only the excitatory population of the E-I model is considered in E-I vs 1 cell type model comparison (page 14).

Thanks for sharing the code. From a quick browse through it looks very manageable to implement for follow up work, although some more guidance for how to navigate the quite complicated codebase and how to reproduce specific paper results would be helpful.

We have also updated the code repository, where we have included more complete instructions on how to reproduce results of each figure. We renamed the folders with the computer code so that they point to a specific figure in the paper. The repository has been completed with the output of the numerical simulations we run, which allows immediate replot of all figures. We have deposited the repository at Zenodo to have the final version of the code associated with the DOI https://doi.org/10.5281/zenodo.14628524. This is mentioned in the section Code availability (page 17).